# Robust Distortion-free Watermarks for Language Models

**Rohith Kuditipudi**                                                          *rohithk@stanford.edu*
*Department of Computer Science*
*Stanford University*

**John Thickstun**                                                          *jthickstun@stanford.edu*
*Department of Computer Science*
*Stanford University*

**Tatsunori Hashimoto**                                                          *thashim@stanford.edu*
*Department of Computer Science*
*Stanford University*

**Percy Liang**                                                          *pliang@cs.stanford.edu*
*Department of Computer Science*
*Stanford University*

**Reviewed on OpenReview:** *https://openreview.net/forum?id=FpaCL1MO2C*

## Abstract

We propose a methodology for planting watermarks in text from an autoregressive language model that are robust to perturbations without changing the distribution over text up to a certain maximum generation budget. We generate watermarked text by mapping a sequence of random numbers—which we compute using a randomized watermark key—to a sample from the language model. To detect watermarked text, any party who knows the key can align the text to the random number sequence. We instantiate our watermark methodology with two sampling schemes: inverse transform sampling and exponential minimum sampling. We apply these watermarks to three language models—OPT-1.3B, LLaMA-7B and Alpaca-7B—to experimentally validate their statistical power and robustness to various paraphrasing attacks. Notably, for both the OPT-1.3B and LLaMA-7B models, we find we can reliably detect watermarked text ($p \leq 0.01$) from 35 tokens even after corrupting between 40-50% of the tokens via random edits (i.e., substitutions, insertions or deletions). For the Alpaca-7B model, we conduct a case study on the feasibility of watermarking responses to typical user instructions. Due to the lower entropy of the responses, detection is more difficult: around 25% of the responses—whose median length is around 100 tokens—are detectable with $p \leq 0.01$, and the watermark is also less robust to certain automated paraphrasing attacks we implement.[1]

## 1 Introduction

The ability of language models to mass produce human-like text creates an acute, renewed emphasis on the importance of provenance of generated content. For example, the website StackOverflow has banned users from posting answers using OpenAI's ChatGPT model to mitigate the spread of misinformation on the platform Vincent (2022). A reliable forensic tool for attributing text to a particular language model would empower individuals—such as platform moderators and teachers—to enact and enforce policies on language model usage; it would also better enable model providers to track the (mis)use of their models, e.g., to scrub synthetic text from the training data of future language models.

---

[1]We release all code publicly at `https://github.com/jthickstun/watermark`.

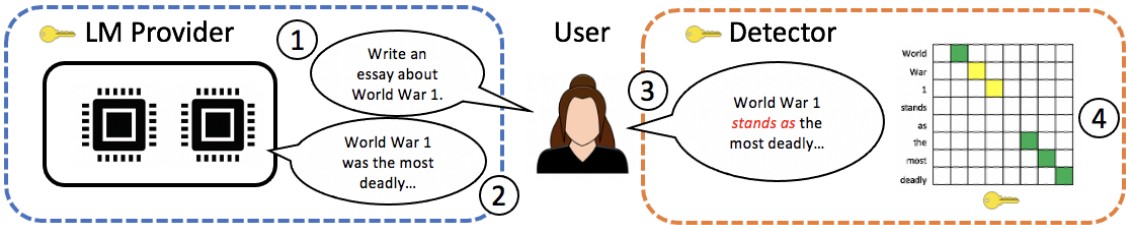

Figure 1: We define the following watermarking protocol between three parties: the LM provider, the user, the detector. The LM provider and the detector coordinate via a shared key, while the user is an untrusted third party. The protocol consists of four steps: 1) the user sends a prompt to the LM provider; 2) the LM provider generates watermarked text to the user; 3) the user edits the watermarked text (to avoid detection) and publishes the edited text; 4) the detector verifies which of the published text came from the LM provider.

To achieve provenance, a *watermark* is a signal embedded within some generated content—in our case, text from a language model—that encodes the source of the content. We consider a setting where a (untrusted) third party user queries a language model (LM) by sending prompts to a trusted provider (Figure 1): the LM provider generates text from their language model with a watermark so that a detector may later identify the source of the text if the user publishes it. The ideal watermark should satisfy at least the following three desiderata:

1. **distortion-free**—the watermark should preserve the original text distribution;

2. **agnostic**—it should be detectable without the language model and/or prompt;

3. **robust**—it should withstand perturbations of the watermarked text.

Existing watermarks either distort the model's sampling distribution, thus altering the API functionality Kirchenbauer et al. (2023); Aaronson (2023), or are not robust to editing or cropping the text Christ et al. (2023). Meanwhile, classical steganographic techniques for covertly encoding messages within samples of text from a language model are neither agnostic nor robust Ziegler et al. (2019). We develop the first watermarks for attributing text to a language model that achieve all three desiderata.

Our methodology consists of two components, which the LM provider and detector respectively use to execute the two steps of the protocol in Figure 1 under their control: a `generate` method that deterministically maps a sequence $\xi$ of random numbers encoded by a (secret) watermark key[2]—which we call the watermark key sequence—to a sample from the language model, and a `detect` method that aligns a putative watermarked text with the watermark key sequence using the shared key. Informally, our watermarks are *distortion-free* in the sense that—marginalizing over the watermark key sequence—each call to `generate` is equal in distribution to a sample from the original language model, i.e., $P(\textbf{text}) = \int_\xi \mathbf{1}\{\textbf{text} = \texttt{generate}(\xi, \textbf{prompt})\} \, d\nu(\xi)$ is equal to the original language model's sampling distribution.

The challenge of detecting watermarked text is that the detector cannot simply recompute `generate` and compare its output against the text since they do not necessarily know the prompt which produced the text: in practice, users often crop the prompt when publishing text from a language model. Our watermarks are *agnostic* in the sense that they are easily detectable with a suitable model-agnostic and prompt-agnostic test statistic $\phi$ such that $\phi(\texttt{generate}(\xi, \textbf{prompt}), \xi) \ll \phi(\textbf{text}, \xi)$ for any **text** that is independent of the watermark key sequence. The idea here is that the detector may use $\phi$ within `detect` to compute a $p$-value with respect to the null hypothesis that the text is independent of the watermark key sequence, i.e., that the text is not watermarked.

To ensure `detect` is *robust* to edits of the watermarked text, the core idea underpinning the design of each test statistic $\phi$ is to leverage techniques for robust sequence alignment to align a putative watermarked

---

[2]Whether the watermark key is secret or not (e.g., if the LM provider publishes the key to allow anyone to detect watermarked text) is an implementation choice that does not affect the main parts of our analysis.

text with the watermark key sequence; we quantify the quality of the alignment using an "alignment cost" specific to each watermark. The sequence alignment procedure ensures the watermark is detectable from even a small, corrupted block of watermarked text planted within some other larger text. Of course, a sufficiently motivated and/or sophisticated user can still evade detection by simply rewriting the text from scratch themselves (or, using another language model to generate the text); the point of a robust watermark is simply that the amount of effort and/or resources a user requires to produce text that evades watermark detection should be commensurate to what they would have expended had they not had access to the watermarked language model in the first place.

Whereas `generate` is a deterministic function, if our watermark produced the same text every time for each prompt it would not be very useful. We resolve this limitation by designing a wrapper `shift-generate` around `generate` that calls `generate` using a randomly chosen subsequence of $\xi$ instead of generating tokens from the same starting point each time. For the same reasons that `detect` is robust to editing and cropping watermarked text, calling `generate` in this fashion does not affect watermark detectability. In practice, the statistical power of our watermarks improves exponentially with respect to the length of the putative watermarked text and diminishes only linearly with the length of the random number sequence; thus, by increasing the length of the random number sequence, we can reduce the probability of reusing the same random subsequence while still ensuring our watermark has good statistical power (i.e., that it yields low $p$-values for watermarked text). So long as we do not reuse an element of the key sequence, successive calls to `shift-generate` will be jointly indistinguishable from regular calls to the language model.

To remark briefly on the work most closely related to ours, we contrast the distortion-free property of our watermarks with the hashing-based watermarks of Kirchenbauer et al. (2023) and Aaronson (2023) that bias the distribution of watermarked text towards certain $k$-grams by hashing a sliding window of the previous $k - 1$ tokens to determine the next token pseudorandomly. We give examples of prompts (e.g., "Give me a list of 20 movies.") for which the bias due to hashing is clearly noticeable in our experiments. Christ et al. (2023) propose a variation of hashing in which the window size changes based on the entropy of the generated tokens to avoid hash collisions with high probability. Their motivation is similar to ours in that they focus on preserving the original text distribution; however, like Kirchenbauer et al. (2023) and Aaronson (2023), using larger window sizes hurts robustness as an adversary can break the watermark by replacing a single token in each window. Our watermark is not only distortion-free but also robust to substantial corruption of the text, which is crucial in practice. We defer a more thorough discussion of related work to the next section (Section 1.1).

We describe the details of our methodology in Section 2, wherein we give two instantiations of watermarks—using inverse transform sampling and exponential minimum sampling—and provide analyses of their statistical power. We experimentally validate the power and robustness of our watermarks using the OPT-1.3B, LLaMA-7B and Alpaca-7B language models in Section 3. Across all models, we find the second instantiation using exponential minimum sampling to be the most powerful. For both the OPT-1.3B and LLaMA-7B models, using this watermark we can reliably detect watermarked text ($p \leq 0.01$) from 35 tokens even after corrupting between 40-50% of the tokens via random edits (i.e., substitutions, insertions or deletions); the watermark also remains detectable from 50 tokens even after paraphrasing the text by translating to French/Russian and back. For the Alpaca-7B model, we conduct a case study on the feasibility of watermarking responses to typical user instructions. Due to the lower entropy of the responses, detection is more difficult: around 25% of the responses—whose median length is around 100 tokens—are detectable with $p \leq 0.01$, and the watermark is also less robust to paraphrasing. We release code for implementing the watermark and reproducing the experiments in this paper, as well as additional supplementary material including an in-browser demo of the watermark detector.[3]

## 1.1 Related work

Text watermarking is a special case of linguistic steganography, in that the goal is to convey a hidden message—the watermark—within a passage of text. Existing approaches to linguistic steganography fall under two broad categories: *edit-based* methods that modify a pre-existing text, and *generative* methods that

---

[3]For assets and supplemental material, see: `https://github.com/jthickstun/watermark`.

construct a distribution over cover text (Ueoka et al., 2021). Crucially, in contrast to steganography, the literature on digital watermarking has historically foregrounded robustness to corruption as a key attribute of a good watermark (Katzenbeisser & Petitcolas, 2000; Atallah et al., 2001). In this light, a text watermark should be able to withstand some perturbations of the text, thus precluding the direct application of many existing techniques for linguistic steganography (Dai & Cai, 2019; Ziegler et al., 2019; Shen et al., 2020).

Older work on text watermarking considers editing a pre-existing text to include a watermark (Rizzo et al., 2019; Abdelnabi & Fritz, 2021; Yang et al., 2022); for a survey of edit-based watermarks, see Kamaruddin et al. (2018). In contrast, we are interested in generating watermarked text while preserving the distribution over the text from a language model. Work on generative watermarking is nascent, underwritten by recent advances in open-ended text generation (Brown et al., 2020). Pioneering work by Venugopal et al. (2011) proposed a generative watermark for the output of a machine translation system, biasing the system towards translations with particular features that can later be detected using a hypothesis test.

Our work is most closely related to Kirchenbauer et al. (2023), who watermark text from a language model by reweighting the token log-probabilities from the model at inference time as a function (i.e., hash) of the previous $k - 1$ tokens, where $k \in \mathbb{N}$ is a hyperparameter. In ongoing unpublished work concurrent to ours, Aaronson (2023) describes a technique for watermarking language models using exponential minimum sampling (a close relative of the Gumbel trick Papandreou & Yuille (2011)) to sample from the model, where the inputs to the sampling mechanism are also a hash of the previous $k - 1$ tokens. Neither watermark is distortion-free, and in fact we show in our experiments that the distortions manifest noticeably in practice (e.g., excessive repetition of certain tokens). Specifically, both Kirchenbauer et al. (2023) and Aaronson (2023) bias the distribution toward a subset of $k$-grams. Increasing $k$ makes the bias less noticeable but hurts the robustness of both watermarks: an adversary can break the signal from a particular token by replacing any one of the previous $k - 1$ tokens.

Also concurrent to our work, Christ et al. (2023) propose watermarking blocks of text from a language model by hashing each block to seed a sampler for the next block. Christ et al. (2023) vary their block sizes—which are analogous to the hyperparameter $k$ of Kirchenbauer et al. (2023) and Aaronson (2023)—as a function of the empirical entropy of the constituent tokens to avoid using the same seed twice with high probability. Their work is similar to ours in that they preserve the original language model's sampling distribution; however, the resulting watermark is not robust since in order to mitigate the distortion induced by hashing the block sizes must be sufficiently large to avoid hash collisions with high probability over all blocks and—similar to Kirchenbauer et al. (2023) and Aaronson (2023)—replacing any token in the previous block breaks the watermark in the next block. Whereas Christ et al. (2023)—who do not run experiments—choose their block sizes to be sufficiently large to minimize distortion, Kirchenbauer et al. (2023) and Aaronson (2023) recommend choosing $k$ to be a small constant in practice, which ensures a moderate amount of robustness by introducing some distortion. Finally, whereas our definition distortion-freeness implies exact equality in distribution of watermarked text to unwatermarked text for a single query to the language model, Christ et al. (2023) propose a definition of "undetectability" that implies approximate equality in distribution, i.e., approximate distortion-freeness, over multiple queries. Using `shift-generate`, we also achieve approximate distortion-freeness in the multiple query setting, though the runtime of our watermark detection procedure must grow with the number of queries; we discuss these trade-offs in more detail in Section 4.

An alternative approach for detecting synthetic text is to learn a classifier between synthetic and human text (Jawahar et al., 2020; Mitchell et al., 2023). A key advantage of such methods over watermarking is that they do not require coordination with the original producer of the text (i.e., the LM provider); however, their effectiveness is distribution dependent and they do not provide a priori (distribution-free) guarantees on the significance level of detection (i.e., Type I errors).

Finally, we note that our setting is different from the literature on planting watermarks in the training data of machine learning models, e.g., to infer the model's training set or otherwise influence the model's output (He et al., 2022a;b; Zhao et al., 2023). Such watermarks are not distortion-free by design, since the point is to plant some learnable signal in the training data that influences the behavior of models which train on the watermarked data.

## 2 Methodology and theoretical analysis

Let $\mathcal{V}$ be a discrete set, i.e., the *vocabulary*, and let $p \in \mathcal{V}^* \to \Delta(\mathcal{V})$ be an autoregressive *language model* which maps a string of arbitrary length to a distribution over the vocabulary, with $p(\cdot \mid x)$ denoting the distribution of the next token given the prefix $x \in \mathcal{V}^*$. Let $\Xi$ denote the space in which the elements of the watermark key sequence lie. Recall the main protocol (Figure 1) which defines our problem setting:

0. The LM provider shares a random watermark key sequence $\xi \in \Xi^*$ with the detector;

1. The user sends a prompt $x \in \mathcal{V}^*$ to the LM provider;

2. The LM provider generates text $Y \in \mathcal{V}^*$ by $Y = \texttt{generate}(x, \xi)$;

3. The user publishes text $\widetilde{Y} \in \mathcal{V}^*$, which may be either (i) (an edited version of) the generated text $Y$ or (ii) text independent of $Y$ (e.g., text that they wrote themselves);

4. The detector determines if $\widetilde{Y}$ is watermarked—i.e., if $\widetilde{Y}$ depends on the watermark key sequence—by computing a $p$-value $\widehat{p} = \texttt{detect}(\widetilde{Y}, \xi)$ with respect to the null hypothesis that $\widetilde{Y}$ is independent of $\xi$ (i.e., not watermarked).

### 2.1 Protocol details

In the protocol, the LM provider calls the $\texttt{generate}$ method (Algorithm 1) to autoregressively generate text from a language model using a *decoder* function $\Gamma : \Xi \times \Delta(\mathcal{V}) \to \mathcal{V}$ which maps an element $\xi_i$ of the watermark key and a distribution over the next token to a next token prediction. By design, over the randomness of $\xi_i$ the prediction should constitute a sample from the distribution, i.e., $\mathbb{P}(\Gamma(\xi_i, \mu) = y_i) = \mu(y_i)$.

**Definition 1.** A decoder $\Gamma : \Xi \times \Delta(\mathcal{V}) \to \mathcal{V}$ is *distortion-free* with respect to (the distribution of) a random variable $\xi \in \Xi$ if for any $\mu \in \Delta(\mathcal{V})$ and $y \in \mathcal{V}$ it satisfies $\mathbb{P}(\Gamma(\xi, \mu) = y) = \mu(y)$.

We relate Definition 1 to our informal definition of distortion-free text in the introduction through the following simple lemma. Essentially, so long as the watermark key sequence is long enough that we do not reuse any part of it to generate text, the only material difference between an LM provider using $\texttt{generate}$ versus sampling directly from the language model is that the sequence $\xi$ is an input to the method rather than resampled i.i.d. within the method for each call. We treat the language model $p$, the decoder $\Gamma$, and generation length $m$ as internal parameters of the $\texttt{generate}$ method.

**Lemma 2.1.** *Let $m, n \in \mathbb{N}$ with $n \geq m$. Let $\Gamma$ be distortion free with respect to a distribution $\nu \in \Delta(\Xi)$ and let $\{\xi_i\}_{i=1}^n \overset{i.i.d.}{\sim} \nu$. Let $Y = \texttt{generate}(\xi; m, p, \Gamma)$. Then $Y_i \sim p(\cdot \mid Y_{:i-1})$ for $i \in [m]$.*

*Proof.* As $n \geq m$, we have $\{\xi_i\}_{i=1}^m \overset{\text{i.i.d.}}{\sim} \nu$. The claim then follows immediately from applying Definition 1 to Line 2 of $\texttt{generate}$ for $i \in [m]$. $\qquad\square$

To simplify the remainder of the presentation, we do not pass a prompt as input to $\texttt{generate}$. As the language model $p$ is arbitrary and $\texttt{detect}$ is model-agnostic, this simplification is without loss of generality since $p$ itself may model the distribution of text from some base model given an arbitrary prompt. Also, unless stated otherwise, without loss of generality we let $\mathcal{V} = [N]$ throughout the paper, where $N \in \mathbb{N}$ is the vocabulary size.

The detector calls the $\texttt{detect}$ method (Algorithm 2) to compute—via a permutation test with $T$ resamples—a $p$-value with respect to a test statistic $\phi : \mathcal{V}^* \times \Xi^* \to \mathbb{R}$ for the null hypothesis that $\widetilde{Y}$ is not watermarked, i.e., that $\widetilde{Y}$ is independent of $\xi$. The output $\widehat{p}$ of $\texttt{detect}$ is a proper non-asymptotic $p$-value: if $\widetilde{Y}$ is not watermarked, then each $(\widetilde{Y}, \xi^{(t)})$ constitutes an independent, identically distributed copy of $(\widetilde{Y}, \xi)$ and therefore by symmetry $\widehat{p}$ is uniformly distributed over $\{1/(T+1), 2/(T+1), \ldots, 1\}$ for any (non-atomic)

---

**Algorithm 1:** Watermarked text generation (`generate`)

---

**Input** : watermark key sequence $\xi \in \Xi^n$
**Params:** generation length $m$, language model $p$, decoder $\Gamma$
**Output:** string $y \in \mathcal{V}^m$

**1 for** $i \in 1, \ldots, m$ **do**
**2** $\quad$ $y_i \leftarrow \Gamma(\xi_{i\%n}, p(\cdot \mid y_{:i-1}))$
**3 return** $y$

---

test statistic.[4] If $\phi$ returns a small $p$-value (e.g., 0.0001) then the text is likely watermarked; if the $p$-value is large (e.g., 0.25), then the text might not be.

---

**Algorithm 2:** Watermarked text detection (`detect`)

---

**Input** : string $y \in \mathcal{V}^*$, watermark key sequence $\xi \in \Xi^n$
**Params:** test statistic $\phi$; watermark key sequence distribution $\nu \in \Delta(\Xi^n)$; resample size $T$
**Output:** p-value $\widehat{p} \in [0, 1]$

**1 for** $t \in 1, \ldots, T$ **do**
**2** $\quad$ $\xi^{(t)} \sim \nu$
**3** $\quad$ $\phi_t \leftarrow \phi(y, \xi^{(t)})$
**4** $\widehat{p} \leftarrow \frac{1}{T+1} \left( 1 + \sum_{t=1}^{T} \mathbf{1}\{\phi_t \leq \phi(y, \xi)\} \right)$
**5 return** $\widehat{p}$

---

The goal then is to design the test statistic $\phi$ (Algorithm 3) such that $\widehat{p}$ will typically be small if $\widetilde{Y}$ is watermarked. In particular, the goal is to identify an alignment cost $d : (\mathcal{V} \times \Xi)^* \to \mathbb{R}$, which measures the quality of a match between a subsequence of the input text and a subsequence of the watermark key, and use this to define $\phi$ as the minimum cost alignment between length $k$ subsequences of the text and key.

This alignment-based detection strategy makes the watermark robust, since even if the user crops or otherwise corrupts $Y$, a single block of preserved watermarked text within some larger body of unwatermarked text will suffice to trigger a low $p$-value from `detect`. The actual form of the alignment cost will be specific to each watermark—in particular, it will depend on the nature of the decoder $\Gamma$ in `generate`. Our most robust watermarks incorporate a soft notion of edit distance (i.e., Levenshtein distance) into the computation of the alignment cost via dynamic programming, with runtime scaling quadratically in the block size. Thus, letting $m$ be the length of the input text $y$, $n$ be the length of the watermark key sequence $\xi$, and $k$ be the block size, the cost of computing the test statistic is $O(mnk^2)$.

---

**Algorithm 3:** Test statistic ($\phi$)

---

**Input** : string $y \in \mathcal{V}^*$, watermark key sequence $\xi \in \Xi^n$
**Params:** alignment cost $d$, block size $k$
**Output:** test statistic value $\phi(y, \xi) \in \mathbb{R}$

**1 for** $i \in 1, \ldots, \mathtt{len}(y) - k + 1$ **do**
**2** $\quad$ **for** $j \in 1, \ldots, n$ **do**
**3** $\quad\quad$ $y^i \leftarrow \{y_{i+\ell}\}_{\ell=0}^{k-1}$, $\xi^j \leftarrow \{\xi_{(j+\ell)\%n}\}_{\ell=0}^{k-1}$
**4** $\quad\quad$ $\widehat{d}_{i,j} \leftarrow d(y^i, \xi^j)$
**5 return** $\min_{i,j} \widehat{d}_{i,j}$

---

[4]By non-atomic, we mean for any $c \in \mathbb{R}$ that $\mathbb{P}(\phi(Y, \xi) = c) = 0$ so that almost surely we will not have to break ties (meaning, if $\phi(y, \xi) = \phi_t$) when computing $\widehat{p}$. In case of ties (i.e., if the test statistic is atomic), we can either modify `detect` to break ties uniformly at random, or simply report valid but conservative $p$-values by leaving `detect` as is.

To illustrate how the decoder and the alignment cost fit together, we give a simple example for the toy setting of a binary vocabulary.

**Example 1:** Consider a binary vocabulary $\mathcal{V} = \{0, 1\}$. To generate $Y \in \{0, 1\}^*$ from the model, the LM provider shares $\{\xi_i\}_{i=1}^n \overset{\text{i.i.d.}}{\sim} \text{Unif}([0, 1])$ with the detector and let $Y_i = 0$ if $\xi_i \leq p(0 \mid Y_{:i-1})$ and $Y_i = 1$ otherwise. In particular, defining the decoder $\Gamma$ by

$$\Gamma(\xi_i, \mu) := \begin{cases} 0 & \xi_i \leq \mu(0) \\ 1 & \xi_i > \mu(0), \end{cases}$$

let $Y = \texttt{generate}(\xi; m, p, \Gamma)$ for some $m \leq n$. Then $Y$ is a valid sample from the language model as $\mathbb{P}(\xi_i \leq p(0 \mid Y_{:i-1})) = p(0 \mid Y_{:i-1})$, and crucially $Y$ and $\xi$ are correlated (i.e., if $\xi_i$ is sufficiently close to zero then $Y_i = 0$, and likewise if $\xi_i$ is sufficiently close to one then $Y_i = 1$). Thus, we can define the alignment cost $d(y, \xi) = \|y - \xi\|_1$.

Assuming for the sake of this example that $n = m$ and the user does not corrupt the watermarked text from the LM provider, i.e., $\widetilde{Y} = Y$, the detector can run $\texttt{detect}$ to verify that $\widetilde{Y}$ is watermarked using the test statistic $\phi$ with alignment cost $d$ and block size $k = m$. The value of the test statistic will then be at most the $\ell_1$ norm of $\widetilde{Y} - \xi$. $\diamond$

## 2.2 Handling multiple queries

In the above example, the LM provider generates the same text each time from the watermark key sequence, which is not ideal in practice. One solution for avoiding reusing elements of the watermark key sequence across queries is to make $\texttt{generate}$ stateful, thus enabling the LM provider to generate a total of $\lfloor n/m \rfloor$ independent watermarked text samples of $m$ tokens each from the language model. Instead, to avoid persisting state, we provide a randomized wrapper $\texttt{shift-generate}$ (Algorithm 4) around $\texttt{generate}$ and modify the watermarking protocol from the start of the section to allow the LM provider to call the $\texttt{shift-generate}$ instead of $\texttt{generate}$ in the second step of the protocol. The wrapper $\texttt{shift-generate}$ randomly shifts the watermark key sequence before passing the shifted sequence to $\texttt{generate}$. Shifting the watermark key sequence does not affect the value of the test statistic in $\texttt{detect}$, since to compute the test statistic the detector anyways searches over all subsequences of the watermark key sequence to find the best match for each block of text. There are $n$ possible shifts, each of which may produce a distinct text; while in principle these $n$ texts will correlate with each other due to sharing elements of the watermark key sequence, in practice we find the effects of these correlations are not noticeable. The so-called birthday paradox Elfving et al. (1966) implies the LM provider can typically expect to call $\texttt{shift-generate}$ $\Omega(\sqrt{n})$ times, each time producing a different text, before reusing the same offset twice. In fact, the provider can expect call $\texttt{shift-generate}$ $\Omega(\sqrt{n/m})$ times before reusing a subsequence, in which case the constituent $\Omega(\sqrt{nm})$ tokens in these texts will be indistinguishable from regular samples from the language model.

In general, we can bound the distortion (i.e., statistical distance from regular samples) of $\texttt{shift-generate}$ in the multi-query setting by the probability of reusing an element of the watermark key sequence. Specifically, for $T$ queries and a maximum generation length of $m$ tokens per query, we will achieve negligible, i.e., $o(1)$ distortion, so long as $n = \omega(mT^2)$. Thus, similar to Christ et al. (2023) we can achieve approximate distortion-freeness in the multi-query setting. However, unlike Christ et al. (2023), to achieve approximate distortion-freeness in this setting the computational cost of our watermark detection procedure must grow with the target number of queries. In principle, this trade-off between the degree of distortion and the runtime of watermark detection means at least in an asymptotic sense that the latter effectively upper bounds the number of queries to the LM provider an attacker would require to learn information about the watermark key sequence. In practice, we expect the cost per token of queries to the LM provider will be significant enough to make such attacks expensive to implement.

---

**Algorithm 4:** Randomized watermarked text generation (`shift-generate`)

---

**Input** : watermark key sequence $\xi \in \Xi^n$
**Params:** generation length $m$, language model $p$, decoder $\Gamma$
**Output:** string $y \in \mathcal{V}^m$
**1** $\tau \sim \text{Unif}([n])$, $\xi' \leftarrow \{\xi_{(i+\tau)\%n}\}_{i=1}^m$
**2 return** `generate`$(\xi'; m, p, \Gamma)$

---

### 2.3 Terminology: watermark strategies and watermark potential

Henceforth, we use the term *watermarking strategy* to refer to a concrete instantiation of the `shift-generate`, `generate` and `detect` methods by specifying the internal parameters of both algorithms (i.e., the decoder $\Gamma$, the test statistic $\phi$ and the watermark key sequence distribution $\nu$). We give concrete watermarking strategies in the following sections (Sections 2.4 and 2.5). For each watermarking strategy, we show two main results: we prove the decoder is distortion-free and also obtain high probability upper bounds on the $p$-values of watermarked text—as a function of the length of the text and the watermark key sequence. We emphasize that only the former result (i.e., that the decoder is distortion-free) is critical to the validity of our main claims; we intend the latter collection of results to provide intuition for when we would expect the detector to have sufficient power and to anticipate the forthcoming experimental results in Section 3. The strength of the $p$-value upper bounds will depend on the observed token probabilities of (watermarked) text, through a quantity which we evocatively term the *watermark potential*.

**Definition 2.** (watermark potential) Define $\alpha : \mathcal{V}^* \to \mathbb{R}$ by

$$\alpha(y) := 1 - \frac{1}{\texttt{len}(y)} \sum_{i=1}^{\texttt{len}(y)} p(y_i \mid y_{:i-1}).$$

The watermark potential of text from a deterministic language model will always be zero, whereas for a high-entropy model it will approach one. The degree to which it is possible for the detector to reliably distinguish watermarked text from unwatermarked text necessarily depends on the watermark potential of the LM provider's language model. For example, if the language model is deterministic, then any distortion-free watermark will necessarily have zero statistical power. We formalize this intuition by establishing the following general lower bound on the detection accuracy of any watermarking strategy as a function of the watermark potential of the original language model. In particular, we lower bound the error of any classifier $h : \mathcal{V}^* \times \Xi^* \to \{-1, +1\}$ that tries to distinguish watermarked (positive label) versus nonwatermarked text (negative label) given some watermark key $\xi$ (we make no assumption on the distribution of $\xi$ except that it is independent of unwatermarked text by definition). We defer the proof of Lemma 2.2 to Appendix A.

**Lemma 2.2.** *Let $Y_i' \sim p(\cdot \mid Y_{:i-1}')$ for $i \in [m]$. Let $Y \stackrel{d}{=} Y'$ and let $\xi \in \Xi^*$ be a random variable that is independent of $Y'$. Let $h : \mathcal{V}^* \times \Xi^* \to \{-1, +1\}$ be a classifier. Let $c > 0$ and define the set $\mathcal{V}_c \subset \mathcal{V}^m$ by*

$$\mathcal{V}_c := \{y : p(y_i \mid y_{:i-1}) \geq \exp(-c/2) \text{ for all } i \in [m]\}.$$

*Then*

$$\mathbb{P}(h(Y, \xi) = -1) + \mathbb{P}(h(Y', \xi) = 1) \geq \mathbb{E}\left[\exp\left(-cm\alpha(Y)\right) \mathbf{1}\{Y \in \mathcal{V}_c\}\right].$$

Lemma 2.2 implies it is impossible to test between any watermarked and non-watermarked text (i.e., between $Y$ versus $Y'$) that are equal in distribution (i.e., distortion-free) if the text typically has low watermark potential, irrespective of the design of the watermark key; in particular, the sum of the Type I and II (resp., false positive/negative) error rates of $h$ will be close to one if the watermark potential is close to zero. The theorem is not tight: depending on the language model, its result may be vacuous for small values of $c$ (e.g., the constants which appear in our upper bounds) since only texts whose token likelihoods all exceed $\exp(-c/2)$ contribute to the lower bound. Also our upper bounds scale inverse exponentially with

the square of the watermark potential, which will always be smaller than the watermark potential itself since the watermark potential is bounded between zero and one.

The point of the forthcoming $p$-value upper bounds for the watermarking strategies in Sections 2.4 and 2.5 is to establish the existence of test statistics for each watermark such that the statistical power of the watermark improves exponentially with the length of the text and decays at most linearly with the length of the watermark key sequence. The test statistics we use to prove these upper bounds differ slightly from those we employ in our experiments: in the former case, we prioritize the simplicity of stating the bounds in terms of watermark potential, whereas in the latter case, we prioritize empirical performance.

## 2.4 Watermarking via inverse transform sampling

Inverse transform sampling is a general technique for sampling from a univariate distribution by taking the pushforward of a uniform random variable through its inverse cumulative distribution function (CDF). Crucially, the technique is valid irrespective of the ordering of the CDF, a property which we presently leverage to construct a watermarking strategy in which `generate` is distortion-free and also `detect` is agnostic. In particular, we implement `generate` with a decoder that maps a sequence of uniform random variables and permutations to tokens using inverse transform sampling. To detect watermarked text, the detector correlates the sequence of permuted indices of the tokens in the text with the sequence of uniform random variables to detect watermarked text. Meanwhile, for any nonwatermarked text, the sequence of permuted token indices will be i.i.d. uniform irrespective of the text itself and thus not correlate with the sequence of uniform random variables.

Formally, with $\Pi$ as the space of permutations over the vocabulary $[N]$, for $\xi = (u, \pi) \in [0, 1] \times \Pi =: \Xi$ and any distribution $\mu \in \Delta([N])$, define the decoder by

$$\Gamma(\xi, \mu) := \pi^{-1} \left( \min \left\{ \pi(i) : \mu(\{j : \pi(j) \leq \pi(i)\}) \geq u \right\} \right), \tag{1}$$

i.e., $\Gamma(\xi, \mu)$ is the token with the smallest index in the permutation $\pi$ such that CDF of $\mu$ with respect to $\pi$ is at least $u$. Generalizing the intuition from Example 1, we show this decoder is distortion-free in the following theorem.

**Theorem 1.** *Define $\Gamma$ by equation (1). Let $\pi \in \Pi$ be arbitrary and let $U \sim \text{Unif}([0, 1])$, with $\xi := (U, \pi)$. Then $\Gamma$ is distortion-free with respect to $\xi$.*

*Proof.* Recalling Definition 1, the result follows from showing for any $\mu \in \Delta([N])$ and $y \in [N]$ that $\mathbb{P}(\Gamma(\mu, \xi) = y) = \mu(y)$. To this end, by equation (1), we have $\Gamma(\mu, \xi) = y$ if and only if $U$ lies in the interval

$$[\mu(\{y' : \pi(y') < \pi(y)\}), \mu(\{y' : \pi(y') \leq \pi(y)\})).$$

As the width of this interval is exactly $\mu(y)$, the result follows immediately. $\square$

Having shown that the decoder is distortion-free, we now proceed to analyze the detectability of the watermark. For convenience, define the normalization $\eta : [N] \to [0, 1]$ by $\eta(i) := (i - 1)/(N - 1)$. Analogous to the toy example, the sequences $\{\eta(\pi_i(Y_i))\}_{i=1}^m$ and $U$ are correlated. Thus, for the sake of analysis, we define alignment cost $d : (\mathcal{V} \times \Xi)^* \to \mathbb{R}$ by

$$d(y, (u, \pi)) := - \sum_{i=1}^{\text{len}(y)} (u_i - 1/2) \cdot (\eta(\pi_i(y_i)) - 1/2), \tag{2}$$

i.e., the negative covariance (each $U_i$ and $\eta(\pi_i(Y_i))$ both have expectation $1/2$).

We exactly characterize in Lemma 2.3 the difference in the expected value of our alignment cost on some text assuming the text is watermarked (i.e., generated using the same key as the detector) versus not watermarked in terms of the watermark potential of the text (Definition 2). To state the result, we define the constant

$C_0 := \mathrm{Var}(\eta(\mathrm{Unif}([N])))$, where we abuse notation slightly to temporarily treat $\eta$ as a pushforward map over distributions.[5] We defer the proof of Lemma 2.3 to Appendix B.

**Lemma 2.3.** *Let $m, n \in \mathbb{N}$ with $n \geq m$, where $m$ is the generation length and $n$ is the watermark key length. Define the decoder $\Gamma$ by equation (1) and the alignment cost $d$ by equation (2). Let $\xi, \xi' \overset{i.i.d.}{\sim} \mathrm{Unif}(\Xi^n)$ with $Y = \mathtt{generate}(\xi; m, p, \Gamma)$. Then almost surely for all $i \in [m]$ and $j \in [n]$ we have*

$$\mathbb{E}[d(Y_i, \xi'_j) - d(Y_i, \xi_i) \mid Y] = C_0 \cdot (1 - p(Y_i \mid Y_{:i-1})) = C_0 \alpha(Y_{i-1:i}).$$

Summing the result of Lemma 2.3 over $i \in [m]$ implies for any $j \in [n]$ that

$$\mathbb{E}[d(Y, \xi'_{(j+1:j+m)\%n}) - d(Y, \xi_{1:m}) \mid Y] = C_0 m \alpha(Y).$$

Thus, we can upper bound the $p$-value output by $\mathtt{detect}$ in Lemma 2.4 using a standard concentration argument and taking a union bound over $j \in [n]$. We defer the proof of Lemma 2.4 to Appendix B. In fact, we actually prove a more general result for $k \leq m$ wherein we allow $\widetilde{Y}$ to be a subsequence of $Y$ which the user may choose adaptively. We defer this more general result to Appendix B as it is more cumbersome to state.

**Lemma 2.4.** *Let $m, n \in \mathbb{N}$ with $n \geq m$, where $m$ is the generation length and $n$ is the watermark key length. Define the decoder $\Gamma$ by equation (1), alignment cost $d$ by equation (2), and $\phi$ by Algorithm 3 with block size $k = m$. Let $\xi, \xi' \overset{i.i.d.}{\sim} \mathrm{Unif}(\Xi^n)$ with $Y = \mathtt{generate}(\xi; n, p, \Gamma)$ and $\widetilde{Y} = Y$. Then almost surely*

$$\mathbb{P}(\phi(\widetilde{Y}, \xi') \leq \phi(\widetilde{Y}, \xi) \mid \widetilde{Y}) \leq 2n \exp(-k C_0^2 \alpha(\widetilde{Y})^2 / 2).$$

Lemma 2.4 implies that with high probability the value of the test statistic on watermarked text with the correct key will be lower than with a resampled key. In particular, ignoring discretization errors due to the finite number of resamples $T$ in $\mathtt{detect}$, the lemma implies watermarked samples with watermark potential bounded away from zero (i.e., if the language model is not effectively deterministic) will have exponentially small expected $p$-values with respect to the length $m$ of the text. The bound grows only linearly with the length $n$ of the random number sequence, implying for moderately large $m$ (e.g., $m = 50$) an LM provider can generate plenty of distortion-free watermarked text (i.e., $n = 2^{\Omega(m)}$ total tokens) while still enabling detection of the watermark from snippets of $m$ tokens (e.g., 50 tokens typically amount to a couple sentences of text). Of course, recall the computational complexity of detection scales linearly with $n$, which in practice may be a more relevant limitation than the statistical power of the watermark.[6]

### 2.4.1 Robustness to substitutions, insertions and deletions

We show in Lemma 2.5 an analogous result to Lemma 2.4 holds even if an adversary corrupts the original watermarked text by substituting tokens. To state the lemma, we introduce a quantity $\widetilde{\alpha}$ which depends on both the corrupted and original watermarked text and accounts for the decrease in the expected value of the test statistic (which recall for the original text is equal up to a numerical constant to the watermark potential of the text) due to token substitutions. We defer the proof of Lemma 2.5 to Appendix B.

**Lemma 2.5.** *Let $m, n \in \mathbb{N}$ with $n \geq m$, where $m$ is the generation length and $n$ is the watermark key length. Define the decoder $\Gamma$ by equation (1), alignment cost $d$ by equation (2), and $\phi$ by Algorithm 3 with $k = m$. Let $\xi, \xi' \overset{i.i.d.}{\sim} \mathrm{Unif}(\Xi^n)$ with $Y = \mathtt{generate}(\xi; m, p, \Gamma)$ and let $\widetilde{Y} \in \mathcal{V}^m$ be conditionally independent of $\xi$ and $\xi'$ given $Y$. Define*

$$\widetilde{\alpha}(y, \widetilde{y}) := \frac{1}{\mathtt{len}(y)} \sum_{i=1}^{\mathtt{len}(y)} \mathbf{1}\{y_i = \widetilde{y}_i\}(1 - p(y_i \mid y_{:i-1})) - \mathbf{1}\{y_i \neq \widetilde{y}_i\}\frac{1}{N-1}.$$

*Then almost surely*

$$\mathbb{P}(\phi(\widetilde{Y}, \xi') \leq \phi(\widetilde{Y}, \xi) \mid Y, \widetilde{Y}) \leq 2n \exp(-k C_0^2 \widetilde{\alpha}(Y, \widetilde{Y})^2 / 2).$$

---

[5]Note that $C_0 = \mathrm{Var}(\mathrm{Unif}([0,1])) + o_N(1) = 1/12 + o_N(1)$.

[6]Note that both $\mathtt{detect}$ and the test statistic (Algorithm 3) are easily parallelizeable.

Lemma 2.5 implies that even if an adversary replaces the vast majority of tokens in a watermarked text, detection with low $p$-values will still be possible so long as the remaining tokens have watermark potential bounded away from zero. In particular, the permuted indices of the original tokens will still positively correlate with the corresponding uniform random variables from the watermark key sequence, while those of the substituted tokens will exhibit a small negative correlation scaling as $O(1/N)$.

To handle insertions and deletions, we can robustify our test statistic by incorporating a soft notion of edit distance into our original alignment cost. The parameter $\gamma$ in Definition 3 assigns a cost to each insertion and deletion operation when aligning the tokens $y$ with the sequence $\xi$, while the base alignment cost $d_0$ defines the quality of the alignment via a cost function over substitutions. In practice, we drop the minimizations over $y' \in \mathcal{V}$ and $\xi' \in \Xi$ in the second and third cases respectively of the definition; we include them here to make our subsequent theoretical analysis cleaner.

**Definition 3.** (Levenshtein cost) Let $\gamma \in \mathbb{R}$ and $d_0 : \mathcal{V} \times \Xi \to \mathbb{R}$. For $y \in \mathcal{V}^*$ and $\xi \in \Xi^*$, define the *Levenshtein cost* $d_\gamma : \mathcal{V}^* \times \Xi^* \to \mathbb{R}$ by

$$d_\gamma(y, \xi) := \min \begin{cases} d_\gamma(y_{2:}, \xi_{2:}) + d_0(y_1, \xi_1) \\ d_\gamma(y, \xi_{2:}) + \min_{y' \in \mathcal{V}} d_0(y', \xi_1) + \gamma \\ d_\gamma(y_{2:}, \xi) + \min_{\xi' \in \Xi} d_0(y_1, \xi') + \gamma, \end{cases}$$

with $d_\gamma(y, (u, \pi)) := \gamma \cdot \texttt{len}(y)$ if $\xi$ is empty and vice versa (as base cases).[7]

Redefining the test statistic $\phi$ using $d_\gamma$ as the alignment cost—using $d_0$ from equation (2)—ensures detect is robust not only to substituting tokens, but also inserting and deleting tokens from watermarked text, as we show in Lemma 2.6. We defer the proof of Lemma 2.6 to Appendix B. To state the lemma, we first recursively define a notion of edit distance between two strings. The definition is equivalent to the minimum number of insertion and/or deletion operations needed to transform one string into the other (see Lemma B.2).

**Definition 4.** (edit distance) For $y, \widetilde{y} \in \mathcal{V}^*$, define the *edit distance* by

$$d_{\text{edit}}(y, \widetilde{y}) := \begin{cases} d_{\text{edit}}(y_{2:}, \widetilde{y}_{2:}) & y_1 = \widetilde{y}_1 \\ 1 + \min\{d_{\text{edit}}(y_{2:}, \widetilde{y}), d_{\text{edit}}(y, \widetilde{y}_{2:})\} & y_1 \neq \widetilde{y}_1, \end{cases}$$

with $d_{\text{edit}}(y, \widetilde{y}) = \texttt{len}(y)$ if $\widetilde{y}$ is empty and vice versa.

**Lemma 2.6.** *Let $n, m \in \mathbb{N}$ with $n \geq m$, where $m$ is the generation length and $n$ is the watermark key length. Define the decoder $\Gamma$ by equation (1), alignment cost $d = d_\gamma$ with $d_0$ from equation (2) and $\gamma > 1/2$, and $\phi$ by Algorithm 3 using block size $k \leq m$ that divides evenly into $m$. Let $\xi, \xi' \overset{i.i.d.}{\sim} \text{Unif}(\Xi^n)$ with $Y = \texttt{generate}(\xi; m, p, \Gamma)$. Let $\widetilde{Y} \in \mathcal{V}^m$ be conditionally independent of $\xi$ and $\xi'$ given $Y$, with $d_{\text{edit}}(Y, \widetilde{Y}) \leq \varepsilon m$. Then almost surely*

$$\mathbb{P}(\phi(\widetilde{Y}, \xi') \leq \phi(\widetilde{Y}, \xi) \mid \widetilde{Y}, Y) \leq mn(2k)^{k/(4\gamma-1)} \exp(-kC_0^2(\alpha(Y) - \gamma\varepsilon)_+^2/2).$$

We prove the result by showing there must exist a length $k$ substring of the corrupted text $\widetilde{Y}$ within edit distance $k\varepsilon$ of a substring of $Y$ that the detector will be able to distinguish as watermarked. For fixed $k$, the set of strings within edit distance $\varepsilon k$ of an original block watermarked text blows up combinatorially with $\varepsilon$. To ensure we can detect the watermark, the result implies we must set $\gamma = \Omega(1/\varepsilon)$, which means our bound on the expected $p$-value is vacuous as soon as $\varepsilon = \Omega(1/\log k)$. Admittedly, our analysis is not tight; for example, as a preview of the experimental results to come, in practice we find smaller values of $\gamma$ (i.e., $\gamma < 1$) to perform significantly better. However, one takeaway from the result is that using a block size $k < m$, where here $m$ is the length of the input text, for detection can be an effective strategy when the user has substantially corrupted the text. The assumption that $k$ divides evenly into $m$ is an artifact of our analysis and not important in practice.

---

[7]For $y \in \mathcal{V}^*$ (resp., $\xi \in \Xi^*$), we let $y_{\texttt{len}(y)+1:}$ (resp., $\xi_{\texttt{len}(\xi)+1:}$) denote the empty string/sequence.

### 2.4.2  What we run in practice

In practice, to reduce overhead in both `generate` and `detect`, we use a single random permutation[8] instead of a full sequence, i.e., we let $\pi_i = \pi$ for all $i \in [n]$ for $\pi \sim \mathrm{Unif}(\pi)$. Recall Theorem 1 makes no assumption about the distribution of the permutations; thus, the watermark is still distortion-free. Also, for the test statistic, we find using

$$d(y, (u, \pi)) := \sum_{i=1}^{\mathtt{len}(y)} |u_i - \eta(\pi_i(y_i))| \tag{3}$$

as the alignment cost performs better empirically than the alignment cost in equation (2). To reiterate, the output of `detect` is a valid $p$-value irrespective of the test statistic we use.

Henceforth, we refer to this version of the watermarking strategy as `ITS`, and we refer to the corresponding Levenshtein version as `ITS-edit`, wherein we define the base alignment cost $d_0$ by equation (3) and use the following simplified notion of Levenshtein cost:

**Definition 5.** (simple Levenshtein cost) Let $\gamma \in \mathbb{R}$ and $d_0 : \mathcal{V} \times \Xi \to \mathbb{R}$. For $y \in \mathcal{V}^*$ and $\xi \in \Xi^*$, define the alignment cost function $d_\gamma : \mathcal{V}^* \times \Xi^* \to \mathbb{R}$ by

$$d_\gamma(y, \xi) := \min \begin{cases} d_\gamma(y_{2:}, \xi_{2:}) + d_0(y_1, \xi_1) \\ d_\gamma(y, \xi_{2:}) + \gamma \\ d_\gamma(y_{2:}, \xi) + \gamma, \end{cases}$$

with $d_\gamma(y, (u, \pi)) := \gamma \cdot \mathtt{len}(y)$ if $\xi$ is empty and vice versa (as base cases).[9]

In summary, for `ITS` we use the decoder from equation (1), the test statistic from Algorithm 3 with the alignment cost from equation (3), and the watermark key distribution as the uniform distribution over $[0,1]^n \times \Pi$, where recall $n$ is the length of the watermark key sequence. Meanwhile, `ITS-edit` differs from `ITS` only in that we define the test statistic using the Levenshtein cost from Definition 5 with the base cost again from equation (3).

### 2.5  Watermarking via exponential minimum sampling

Aaronson (2023) proposes mapping variables in $[0,1]^N$ to tokens in the vocabulary $[N]$ using exponential minimum sampling to generate watermarked text. Whereas Aaronson (2023) proposes the use of distortion-inducing hashes much like Kirchenbauer et al. (2023), we use exponential minimum sampling to implement the decoder in `generate`, which (after defining a suitable corresponding test statistic) enables an alternative distortion-free and robust watermarking strategy to inverse transform sampling. In particular, for $\xi \in [0,1]^N =: \Xi$ and $\mu \in \Delta([N])$, define the decoder by

$$\Gamma(\xi, \mu) := \underset{i \in [N]}{\arg\min}\ -\log(\xi_i)/\mu(i). \tag{4}$$

We show this decoder is distortion-free in Theorem 2, whose proof we defer to Appendix C.

**Theorem 2.** *Define the decoder $\Gamma$ by equation (4) and let $\xi \sim \mathrm{Unif}([0,1]^N)$. Then $\Gamma$ is distortion-free with respect to $\xi$.*

For the sake of analysis, we define the alignment cost as a slight variation of the proposal of Aaronson (2023) (see Section 2.5.2) by

$$d(y, \xi) := -\sum_{i=1}^{\mathtt{len}(y)} \log \xi_{i, y_i}, \tag{5}$$

---

[8]In principle, with a single random permutation the permuted token indices of both watermarked and nonwatermarked text are no longer conditionally independent of each other, and so the results of Lemmas 2.4, 2.5 and 2.6 no longer apply. However, in practice we observe no degradation in statistical power. Also, irrespective of the lemmas, the $p$-values from `detect` are still valid by construction.

[9]For $y \in \mathcal{V}^*$ (resp., $\xi \in \Xi^*$), we let $y_{\mathtt{len}(y)+1:}$ (resp., $\xi_{\mathtt{len}(\xi)+1:}$) denote the empty string/sequence.

again defining the test statistic $\phi$ by Algorithm 3. Similar to Lemma 2.3 for ITS, we exactly characterize the difference in the expected values of the alignment cost on watermarked versus non-watermarked text in terms of the watermark potential of the text. We defer the proof of Lemma 2.7 to Appendix C.

**Lemma 2.7.** *Let $n \in \mathbb{N}$. Define $\Gamma$ by equation* (4) *and $d$ by equation* (5). *Let $\xi, \xi' \overset{i.i.d.}{\sim} \mathrm{Unif}(\Xi^n)$ with $Y = \texttt{generate}(\xi; n, p, \Gamma)$. Then almost surely for all $i \in [n]$ we have*

$$\mathbb{E}[d(Y_i, \xi'_i) - d(Y_i, \xi_i) \mid Y] = 1 - p(Y_i \mid Y_{:i-1}) = \alpha(Y_{i-1:i}).$$

Summing the result of Lemma 2.7 over $i \in [m]$ implies for any $j \in [n]$ that

$$\mathbb{E}[d(Y, \xi'_{(j+1:j+m)\%n}) - d(Y, \xi_{1:m}) \mid Y] = m\alpha(Y).$$

Thus, defining the test statistic $\phi$ by Algorithm 3 with respect to the alignment cost $d$ from Eqn (5), we can again upper bound the $p$-value output by $\texttt{detect}$ in Lemma 2.8 using a standard concentration argument and taking a union bound over $j \in [n]$. We defer the proof of Lemma 2.8 to Appendix C. Once again, we actually prove a more general result that allows $\widetilde{Y}$ to be any length $k$ subsequence of $Y$.

**Lemma 2.8.** *Let $m, n \in \mathbb{N}$ with $n \geq m$. Define $\Gamma$ by equation* (4), *$d$ by equation* (5), *and $\phi$ by Algorithm 3 with $k = m$. Let $\xi, \xi' \overset{i.i.d.}{\sim} \mathrm{Unif}(\Xi^n)$ with $Y = \texttt{generate}(\xi; n, p, \Gamma)$ and $\widetilde{Y} = Y$. Then almost surely*

$$\mathbb{P}(\phi(\widetilde{Y}, \xi') \leq \phi(\widetilde{Y}, \xi) \mid \widetilde{Y}) \leq 2n \exp\left(-\min\{m\alpha(\widetilde{Y})^2/8, m\alpha(\widetilde{Y})/4\}\right).$$

### 2.5.1 Robustness to corruptions

Showing high probability $p$-value upper bounds for corruptions of watermarked text that hold almost surely given the corrupted text—i.e., analogues of Lemmas 2.5 and 2.6—is more difficult, primarily due to the fact that the summands in the alignment metric from equation (5) are no longer bounded and thus bounding the influence of each substitution and/or insertion operation on the test statistic requires more careful analysis. Of course, we could in principle tweak the alignment metric by truncating the summands in order to prove the analogous results; however, as the main intuitions would carry over from Lemmas 2.5 and 2.6 and the results are not critical to the main thrust of the paper, we do not carry this plan out.

### 2.5.2 What we run in practice

As in the case of ITS, in practice we find using a slight variation of the alignment cost in equation (5) performs better. Namely, following the prescription of Aaronson (2023), we modify the previous alignment cost to instead be

$$d(y, \xi) := \sum_{i=1}^{k} \log(1 - \xi_{i, y_i}). \tag{6}$$

Henceforth, we refer to this version of the watermarking strategy as $\texttt{EXP}$, and we refer to the corresponding Levenshtein version wherein we define the base alignment cost $d_0$ by equation (6) as $\texttt{EXP-edit}$.

In summary, for $\texttt{EXP}$ we use the decoder from equation (4), the test statistic from Algorithm 3 with the alignment cost from equation (6), and the watermark key distribution as the uniform distribution over $\Xi^n$, where recall $n$ is the length of the watermark key sequence and $\Xi = [0, 1]^N$. Meanwhile, $\texttt{EXP-edit}$ differs from $\texttt{EXP}$ only in that we define the test statistic using the Levenshtein cost from Definition 5 with the base cost again from equation (6).

## 3 Experimental results

We empirically validate the statistical power of our watermarking strategies (i.e., $\texttt{ITS}$, $\texttt{ITS-edit}$, $\texttt{EXP}$, and $\texttt{EXP-edit}$) via experiments with the OPT-1.3B (Zhang et al., 2022) and LLaMA-7B (Touvron et al., 2023)

models.[10] We run experiments using `generate` rather than `shift-generate`, mainly for the sake of reproducibility; recall however that this choice has no impact on the $p$-values we report. We test for all watermarks using a block size $k$ (in Algorithm 3) equal to the length $m$ of the text. Following the methodology of Kirchenbauer et al. (2023), we generate watermarked text continuations of prompts sampled from the news-like subset of the C4 dataset (Raffel et al., 2020). We vary the generation length $m$ (Experiment 1) and the random number sequence length $n$ (Experiment 2), and we report median $p$-values of watermarked text over 500 samples.[11]

We also evaluate robustness to four kinds of paraphrasing attacks: randomly substituting a fraction of the generated tokens with tokens chosen uniformly at random from the vocabulary (Experiment 3); randomly inserting a fraction of tokens among the generated tokens (Experiment 4); randomly deleting a fraction of the generated tokens (Experiment 5); using another language model to translate the text from English to French and back (Experiment 6). The first three attacks allow us to systematically vary the level of corruption, while the last attack is an example of an attack we might encounter in the wild. We defer the details of the translation procedures to Appendix D.2.

Finally, using the Alpaca-7B model and evaluation dataset Taori et al. (2023), we conduct a case-study on the feasibility of watermarking the responses of a performant instruction-tuned language model to user queries. We also show for certain kinds of instructions that hashing-based watermarks produce noticeably worse responses than our distortion-free watermarks, thus underlining the importance of the distortion-free property in practice.

In all our experiments—except for Experiment 2, where the control variable $n$ is a hyperparameter that is unique to our watermarks—we also replicate the watermark of Kirchenbauer et al. (2023) as a baseline, setting the greenlist fraction $\gamma = 0.25$ and varying the logit bias $\delta \in \{1.0, 2.0\}$. We respectively refer to these versions of their watermark as `KGW-1.0` and `KGW-2.0` after the first three authors' last names. We emphasize their watermark is not directly comparable to our watermarks as it is not distortion-free (e.g., Kirchenbauer et al. (2023) report that even the weakest version we employ with $\delta = 1.0$ and $\gamma = 0.25$ typically increases perplexity by 5–10%).

In their work, Kirchenbauer et al. (2023) report approximate $p$-values, which they obtain from computing the $z$-score of a certain test statistic. To ensure a fair comparison, we use `detect` (with $T = 5000$) to report $p$-values for all watermarks;[12] in the case of `KGW-1.0` and `KGW-2.0`, we run `detect` using the original inexact $p$-values they report as the test statistic. We report error bars for the median $p$-value based on a bootstrapped estimate of the standard deviation using 1000 resamples.

Instead of recomputing the test statistic $T$ times for each prompt—as we originally prescribe in `detect`—to save computation we simply sample $T$ prompts and compute the test statistic once for each ground-truth length $m$ completion; we then use the empirical distribution of these test statistics as the reference distribution within `detect`, which gives a proper $p$-value with respect to the null hypothesis that the text is an original completion from the dataset. For reference, we include the full pseudocode for this modified version of `detect` in Appendix D.3, and we also plot the full distributions of $p$-values for nonwatermarked generations (i.e., regular samples from the language models) to verify they are indeed roughly uniform over the interval $[0, 1]$.

We defer further details regarding our experimental protocol to Appendix D.

## 3.1 Varying text and watermark key length

We vary the length $m$ of watermarked text in Figure 2, fixing the watermark key length $n = 256$ for each of our watermarks and setting $\gamma = 0.4$ for `ITS-edit` and $\gamma = 0.0$ for `EXP-edit` (see Appendix D.4 for the details of tuning $\gamma$). Our ITS watermarks slightly outperform `KGW-1.0` while our EXP watermarks slightly

---

[10]We will also at times collectively refer to `ITS` and `ITS-edit` as the ITS watermarks and/or strategies and `EXP` and `EXP-edit` as the EXP watermarks and/or strategies.

[11]The median $p$-value corresponds to the significance level (i.e., Type I error rate) at which the power of our watermark detector is at least 0.5.

[12]This setting of $T$ means we never report $p$-values less than 1/5000 (i.e., 0.0002) in any of our experiments.

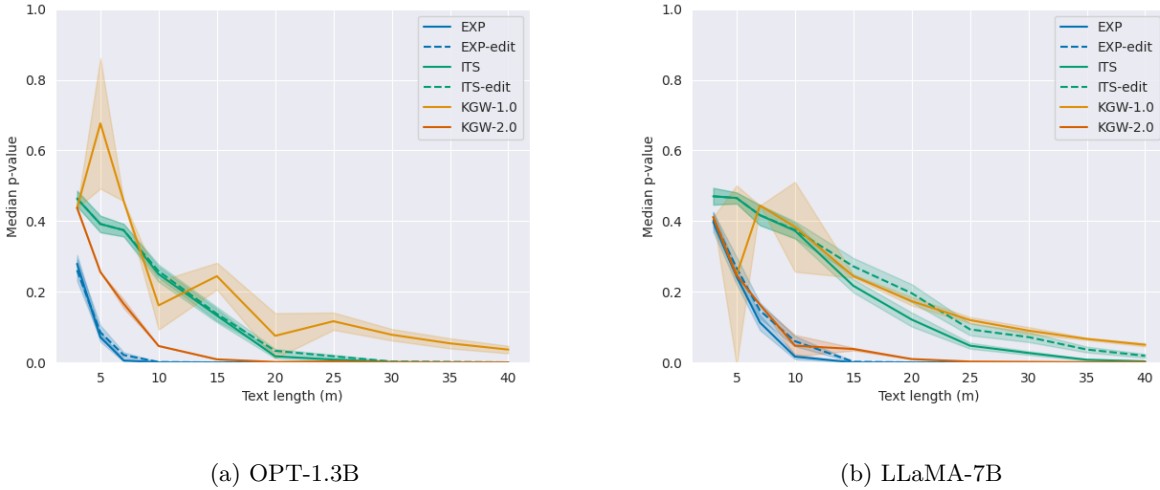

(a) OPT-1.3B

(b) LLaMA-7B

Figure 2: Median $p$-value of watermarked text relative to varying the text length $m$, for OPT-1.3B and LLaMA-7B models. Our watermark strategies are competitive with those of Kirchenbauer et al. (2023), despite the fact that they distort the text distribution to generate watermarked text whereas we do not.

outperform `KGW-2.0`, despite the fact that `KGW-1.0` and `KGW-2.0` both distort the text distribution. The EXP watermarks are notably more powerful than the ITS watermarks, requiring roughly two to three times fewer tokens to achieve a comparably low median $p$-value. One conceivable advantage of the ITS watermarks over the EXP watermarks is that they have comparatively less overhead: the watermark key for `EXP` and `EXP-edit` is a sequence of $n$ vectors in $[0,1]^N$, where recall $N$ is the size of the vocabulary, while for `ITS` and `ITS-edit` it is simply a sequence of $n$ numbers in $[0,1]$. All watermarking strategies perform worse on LLaMA-7B than OPT-1.3B, due to the fact that LLaMA-7B typically produces lower entropy text than OPT-1.3B. Due to the discrete nature of the test statistic of Kirchenbauer et al. (2023), i.e., the number of tokens in the text belonging to a "greenlist" versus a "redlist", the median $p$-values for the `KGW-1.0` and `KGW-2.0` watermarks are occasionally unstable, particularly for small values of $m$.

We vary the length $n$ of the watermark key sequence $\xi$ in Figure 3 for different lengths $m$ of watermarked text from the ITS and EXP watermarks respectively. Recall $n$ corresponds to the total number of tokens we can generate while maintaining our distortion-free guarantee. As our theory predicts, the $p$-values of watermarked text grow linearly with $n$. The rate of growth is fairly mild and decreases rapidly with $m$; even for $n = 4096$, which is larger than the maximum generation length of both the OPT-1.3B and LLaMA-7B models, slightly increasing the number of tokens (by 4–8 tokens in the case of EXP, and 10–20 tokens in the case of ITS) suffices to distinguish watermarked text with roughly the same statistical power as $n = 64$.

## 3.2 Robustness to corruption and paraphrasing

We now proceed to evaluate the robustness of our watermark strategies to various forms of corruption and paraphrasing. We focus on comparing our strongest watermarks (`EXP` and `EXP-edit`) against `KGW-2.0`, deferring results for all other watermarks to Appendix D.5. As larger $n$ increases the computational overhead of computing our test statistics and the effect of larger $n$ on statistical power is mild (as shown in Figure 3), we run all experiments with $n = 256$, which in any case is sufficiently large to ensure the watermarked text across all experiments is distortion-free. Decreasing the insertion/deletion penalty $\gamma$ improves robustness (at least up to a point) but hurts the statistical power of the `ITS-edit` and `EXP-edit` watermarks for larger $n$, since reducing the penalizer for edits effectively increases the number of candidate alignments under consideration. We run `ITS-edit` and `EXP-edit` with the same choices of $\gamma$ as in the previous section. We defer the details of tuning $\gamma$ to Appendix D.4.

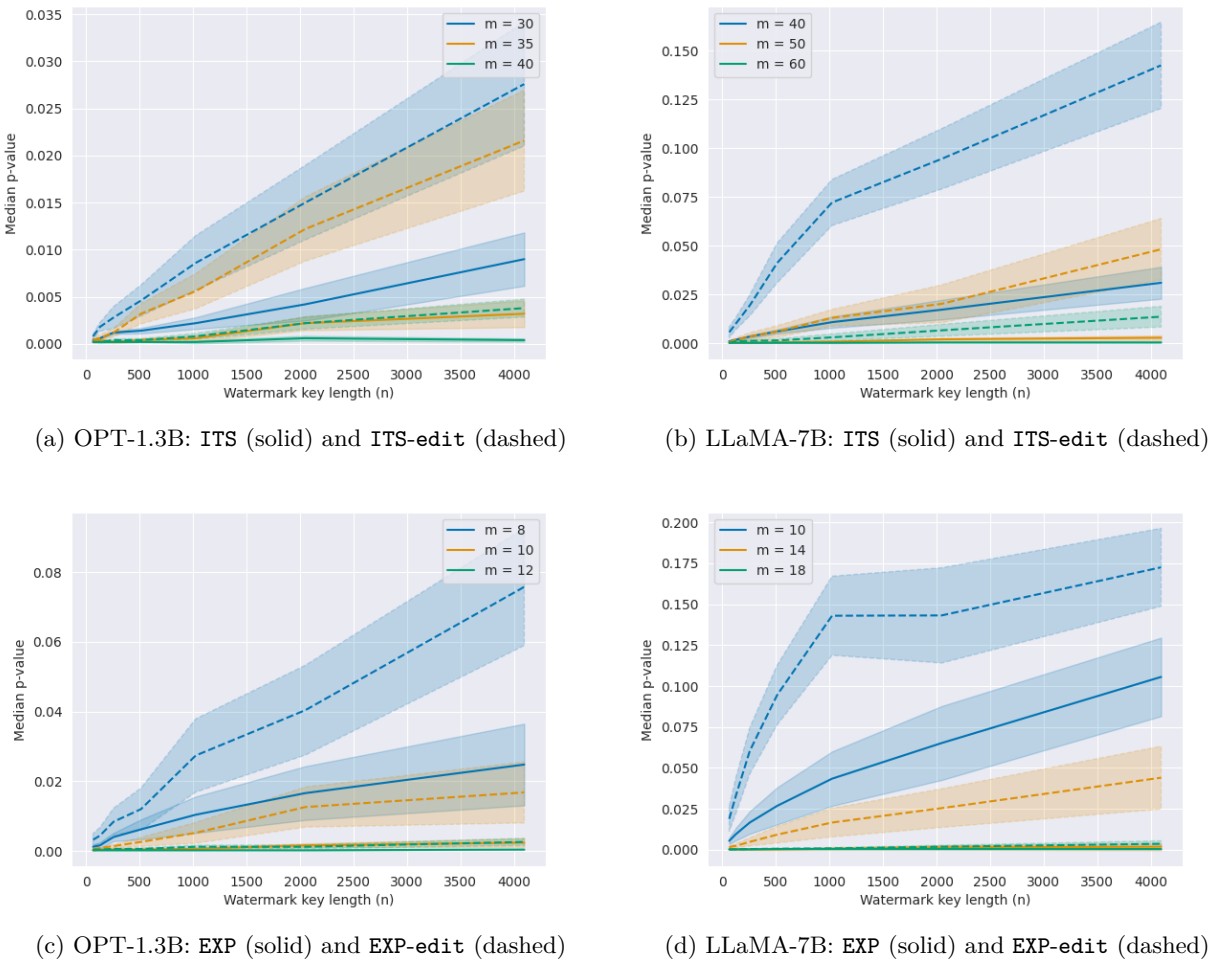

(a) OPT-1.3B: `ITS` (solid) and `ITS-edit` (dashed)

(b) LLaMA-7B: `ITS` (solid) and `ITS-edit` (dashed)

(c) OPT-1.3B: `EXP` (solid) and `EXP-edit` (dashed)

(d) LLaMA-7B: `EXP` (solid) and `EXP-edit` (dashed)

Figure 3: Median $p$-value of watermarked text for varying the watermark key length $n$. Across all watermarks for both the OPT-1.3B and LLaMA-7B models, the median $p$-values grow linearly with $n$ but decay rapidly with increasing text length $m$.

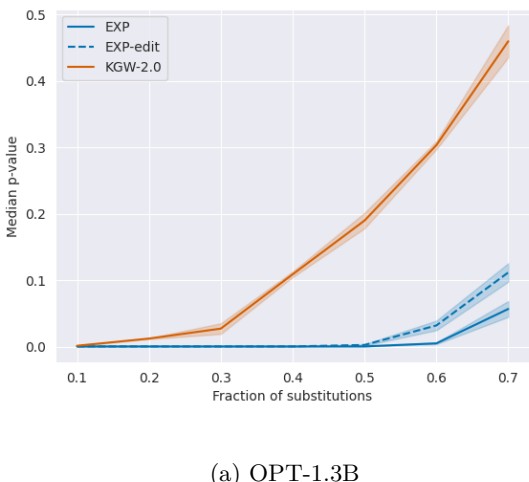

(a) OPT-1.3B

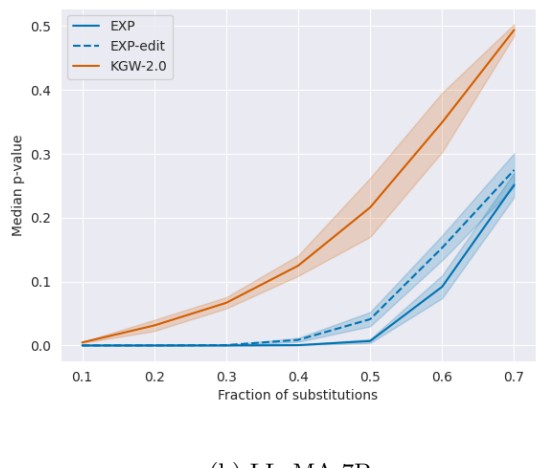

(b) LLaMA-7B

Figure 4: Median $p$-value of watermarked text relative to the fraction of substitution errors, for OPT-1.3B and LLaMA-7B models with $m = 35$. Both versions of the EXP watermark significantly outperform KGW-2.0, again despite KGW-2.0 distorting the text distribution.

We vary the fraction of substituted tokens in Figure 4, and we vary the fraction of inserted and deleted tokens in Figures 5 and 6 respectively. For the insertion experiment, we pass only the first $m$ tokens to the detector; similarly, for the deletion experiment, we initially generate more than $m$ watermarked tokens so that even after deleting a fraction thereof, there are at least $m$ tokens remaining. The EXP and EXP-edit watermarks are comparably robust to substitution errors, but the latter is far more robust to insertion and deletion errors.

We compare our watermarks against the most robust version of KGW-2.0, in the sense that we hash only the previous token to determine the next token distribution and thus bias the distribution towards some subset of bigrams. If instead we hash the previous $k$ tokens for $k > 1$, then substituting any one of the previous $k$ tokens will break the watermark signal in a particular token, and thus the statistical power of their watermark will be worse than what we report in our experiments.

Finally, in Figures 8 and 9 we implement a "roundtrip translation" attack, wherein we attempt to paraphrase watermarked texts of varying lengths by translating the (English) texts into another language (i.e., French and Russian respectively) and back again using a machine translation model (details in Appendix D.2). We include a representative example of the original and (re-)translated texts in Figure 7. Using Russian is a noticeably more effective attack than French: none of the watermarks aside from EXP-edit are able to reliably detect watermarked text with $p < 0.05$ irrespective of $m$. In fact, for Russian the power of both EXP and KGW-2.0 stagnates (or even diminishes) with increasing text length, perhaps due to the inherent ambiguity in translating longer texts.

In many cases, both using French and Russian, the roundtrip translation still preserves large chunks of the original text, which suffices for watermark detection even using EXP, which is substantially less robust to insertion and deletion errors than EXP-edit. Aside from inspecting a few examples, we did not verify that the roundtrip translations preserve the basic semantics of the original text; thus, it is possible our results provide an overly pessimistic view of the robustness of our watermarks to these attacks, since in practice users would presumably not publish such examples. It is also possible that using different machine translation models—or more generally, different forms of automated paraphrasing—might be far more effective in evading watermark detection than those we employed. We publish the full set of watermarked generations for each watermarking strategy, along with their (roundtrip) translations, as part of our code release.

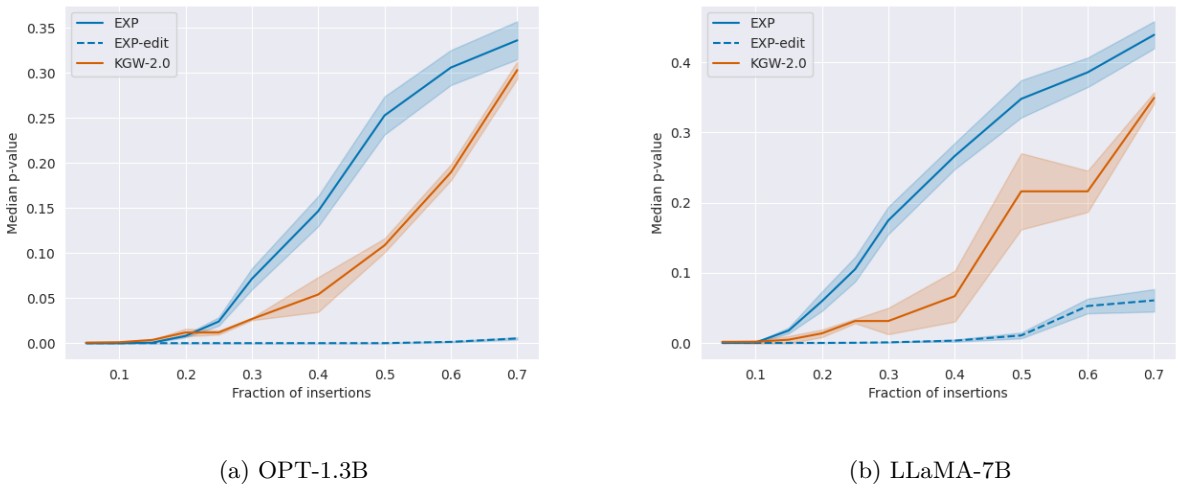

(a) OPT-1.3B

(b) LLaMA-7B

Figure 5: Median $p$-value of watermarked text relative to the fraction of insertion errors, for OPT-1.3B and LLaMA-7B models with $m = 35$. EXP-edit is by far the most robust.

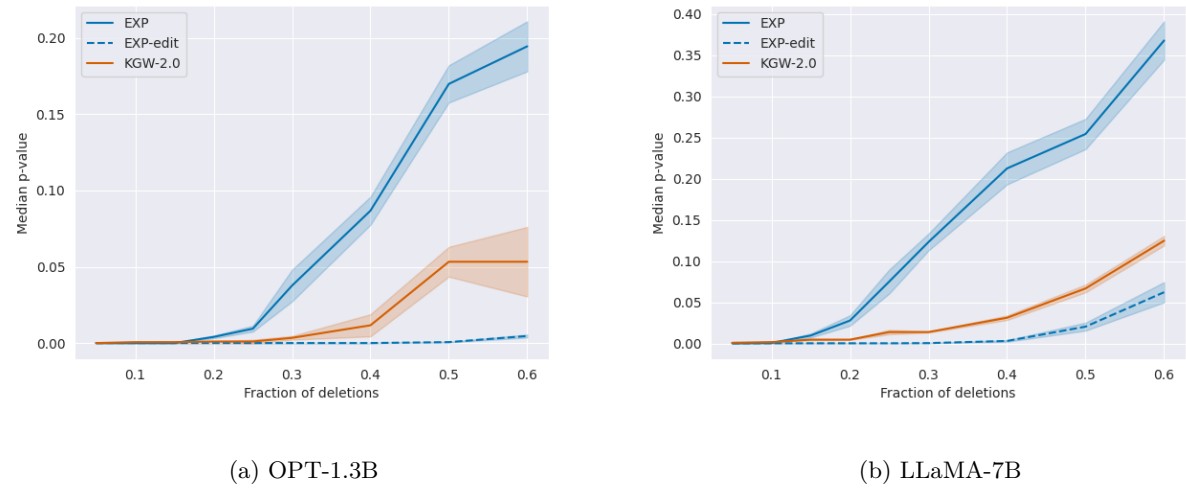

(a) OPT-1.3B

(b) LLaMA-7B

Figure 6: Median $p$-value of watermarked text relative to the fraction of deletion errors, for OPT-1.3B and LLaMA-7B models with $m = 35$. EXP-edit is again the most robust.

| | |
|---|---|
| original text | dependent on the private sector. That helps them preserve the advantages their industry has in government healthcare contracting, while decreasing to near extinction the government's ability to regulate the private sector's cost-driving practices. Hence, his proposal was not was not designed to prevent socialized medicine. It's designed to make it harder and more expensive than ever to get socialized medicine. That will help keep democrats out of power by making them seem soft on dependency and the social good. If you want to prevent socialized medicine, go for the Republicans' heads, not the bureaucrats' feet. In the meantime, though, the Republican plan is to redefine "govern |
| French translation | dépendant du secteur privé Cela les aide à préserver les avantages de leur industrie dans les contrats de soins de santé du gouvernement, tout en diminuant jusqu'à la quasi-extinction la capacité du gouvernement à réglementer les pratiques de conduite des coûts du secteur privé. Il est conçu pour rendre plus difficile et plus cher que jamais d'obtenir la médecine socialisée Cela aidera les démocrates hors du pouvoir en les faisant paraître doux sur la dépendance et le bien social. Si vous voulez empêcher la médecine socialisée, optez pour les têtes des républicains, pas les pieds des bureaucrates En attendant, cependant, le plan républicain est de redéfinir « le gouvernement ». |
| roundtrip translation | This helps them preserve the benefits of their industry in government health care contracts, while reducing the government's ability to regulate private sector cost-management practices to near extinction. It's designed to make it harder and more expensive than ever to get socialized medicine that will help keep Democrats out of power by making them look gentle on addiction and social good. If you want to prevent socialized medicine, opt for the heads of Republicans, not the feet of bureaucrats In the meantime, however, the Republican plan is to redefine "the government." |

Figure 7: An illustrative example of a roundtrip translation attack via French. Given the first 50 tokens of the roundtrip translation (highlighted in green, in addition to the closest matching snippets to these tokens from the original text), detect returns $\widehat{p} \leq 0.0002$.

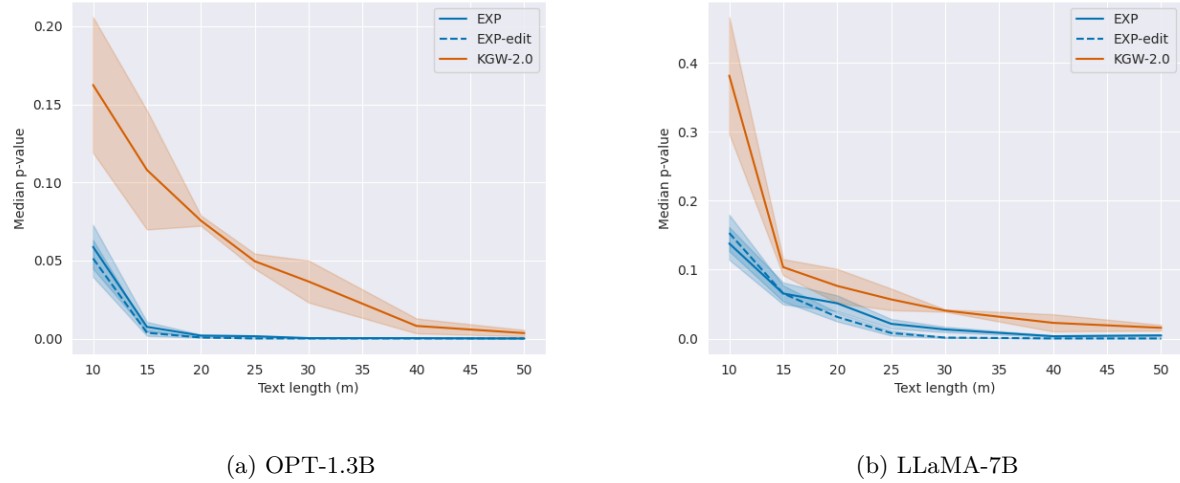

(a) OPT-1.3B          (b) LLaMA-7B

Figure 8: Median $p$-value of watermarked text relative to the text length, after roundtrip translation via French, for OPT-1.3B and LLaMA-7B models with $m = 35$. EXP performs comparably to EXP-edit, indicating that the roundtrip translation attack tends to preserve at least some snippets of the original text.

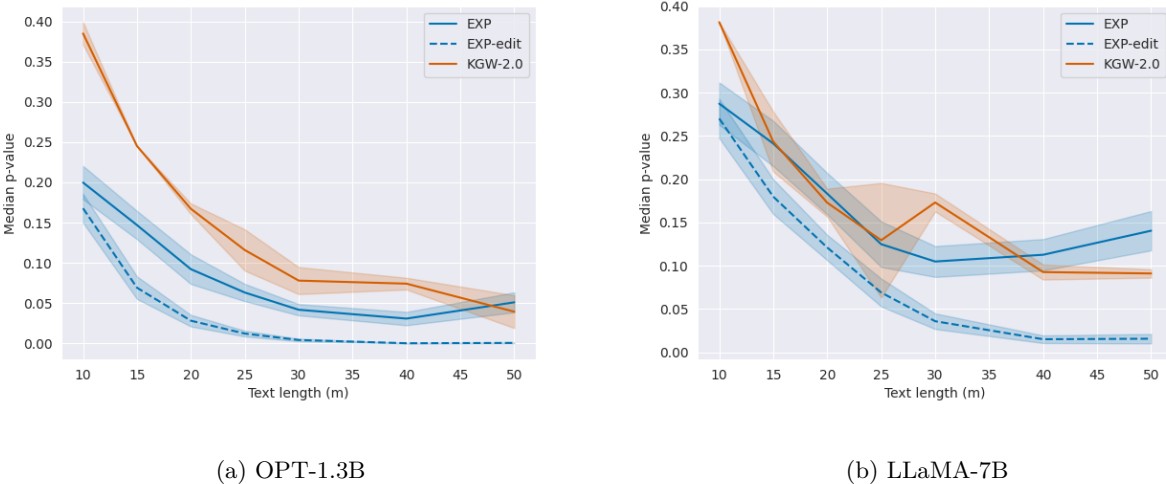

(a) OPT-1.3B                                        (b) LLaMA-7B

Figure 9: Median $p$-value of watermarked text relative to the text length, after roundtrip translation via Russian, for OPT-1.3B and LLaMA-7B models with $m = 35$. In contrast to French, `EXP-edit` noticeably outperforms `EXP`. Overall, the attack is noticeably more effective than using French.

### 3.3    Case study: instruction following

In the wild, most users interact with language models by prompting the model with instructions (e.g., "give me code for..."), and the most widely-used language models (e.g., ChatGPT) are specifically fine-tuned to follow such instructions. Thus, using the instruction fine-tuned Alpaca-7B model, we presently conduct a case study on the effectiveness of watermarking a performant instruction following model. In particular, we sample 200 instructions from the Alpaca-7B evaluation dataset and generate watermarked responses of at most 200 tokens for each. We then compute conditionally valid $p$-values for each response using the original version of `detect` with $T = 500$. We also replicate the roundtrip translation attack from Experiment 6. We publish the full set of watermarked generations for each method, along with their (roundtrip) translations, and the instruction prompts as part of our code release.

We plot the distribution of $p$-values for the `EXP-edit` and `KGW-2.0` watermarks in Figure 10, as well as the $p$-values versus the watermark potential of the watermarked text in Figure 11. In general, the Alpaca-7B responses have considerably lower per-token watermark potential than both the OPT-1.3B and LLaMA-7B models, and thus the statistical power of our watermark is worse despite the responses typically being longer than in the previous experiments (i.e., Experiments 1 and 6). In particular, based on the same random sample of 200 prompts (from the Alpaca evaluation set in the case of Alpaca-7B, and from the news-like subset of the C4 dataset in the cases of LLaMA-7B and OPT-1.3B), the average per-token watermark potential of text from Alpaca-7B is 0.28, compared to 0.59 for LLaMA-7B and 0.67 for OPT-1.3B. Unlike the previous experiments, `KGW-2.0` noticeably outperforms the `EXP-edit` watermark. Figure 11 indicates this difference in performance is largely due to the fact `KGW-2.0` distorts the distribution of the text and produces responses with noticeably larger watermark potential than regular responses from the model. For responses whose unnormalized watermark potential (i.e., watermark potential multiplied by the number of tokens in the response, to account for the varying lengths of the responses) exceeds roughly 60, both watermarks tend to yield $p$-values close to zero. Paraphrasing the responses via roundtrip translation attacks into both French and Russian degrades the statistical power of both watermarks, as we show in Figures 12 and 13.

Finally, recall the main distinguishing feature of our watermark compared to Kirchenbauer et al. (2023) and Aaronson (2023) is that we do not hash previous tokens to determine the distribution of the next token. To demonstrate the pitfalls of hashing, we implement a version of the watermark Aaronson (2023) proposes by modifying the `generate` method of `EXP` to obtain the vector $\xi_i \in [0,1]^N$ from seeding a random number generator using the previous $k$ tokens instead of using the watermark key; we call this version `EXP-hash`. We

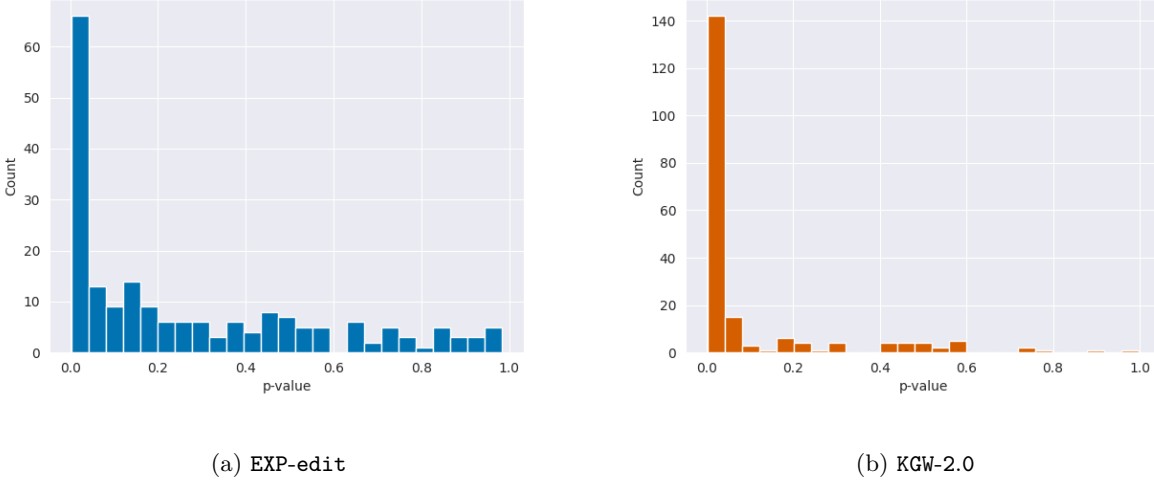

(a) `EXP-edit`

(b) `KGW-2.0`

Figure 10: Histogram of $p$-values of watermarked text from Alpaca-7B. `KGW-2.0` is noticeably better than `EXP-edit`, though again the results are not strictly comparable as `KGW-2.0` is not distortion-free.

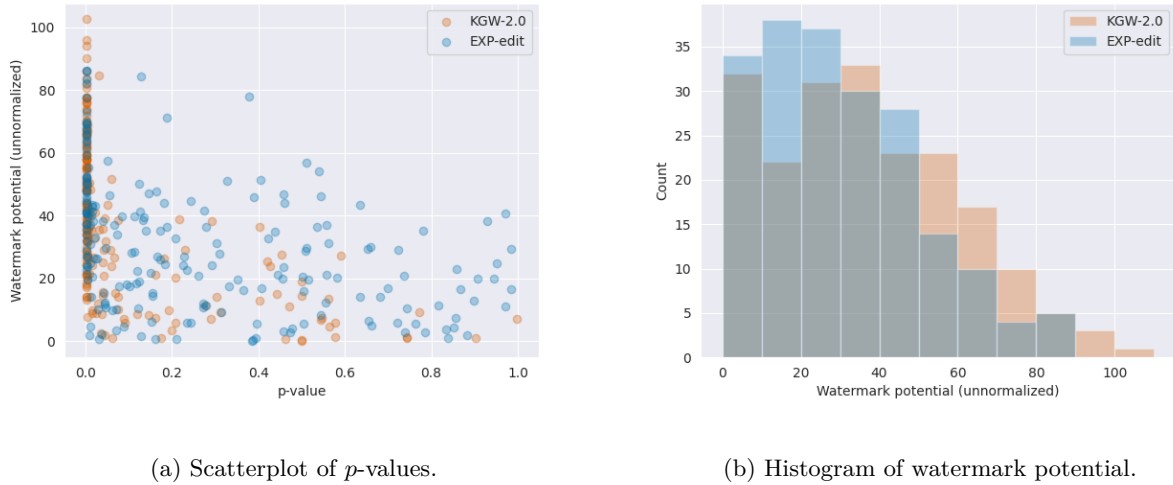

(a) Scatterplot of $p$-values.

(b) Histogram of watermark potential.

Figure 11: Watermark potential versus statistical power of `EXP-edit` versus `KGW-2.0`. `KGW-2.0` noticeably distorts the text distribution, tending to produce higher watermark potential text overall than the original language model (and consequently, `EXP-edit`).

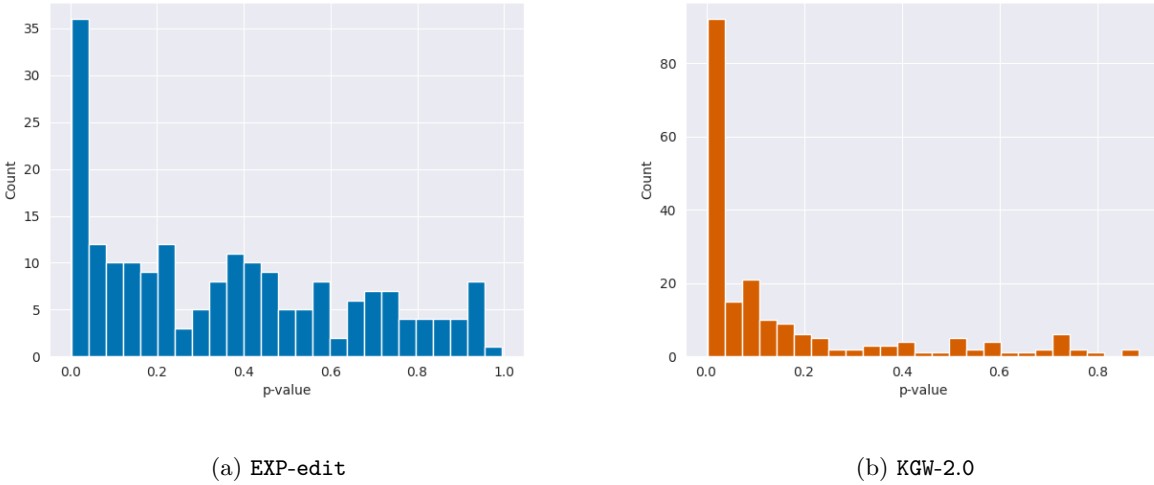

(a) `EXP-edit`            (b) `KGW-2.0`

Figure 12: Histogram of $p$-values of watermarked text after roundtrip translation via French. `KGW-2.0` outperforms `EXP-edit`, albeit by noticeably distorting the text distribution.

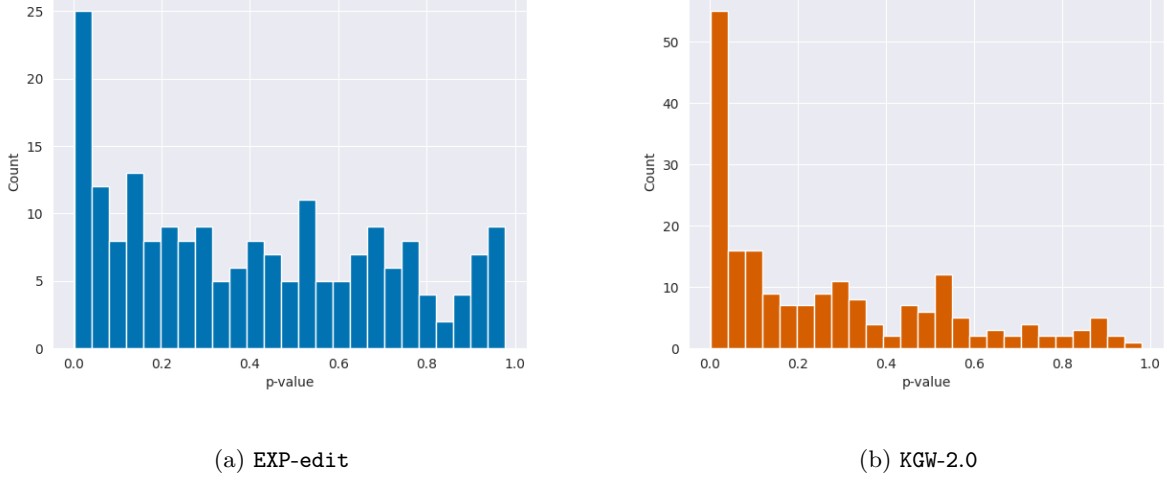

(a) `EXP-edit`            (b) `KGW-2.0`

Figure 13: Histogram of $p$-values of watermarked text after roundtrip translation via Russian. `KGW-2.0` again does significantly better than `EXP-edit`.

```
1. Watermarking Deep Learning Models: A Review
2. Understanding and Evaluating Watermarking Techniques for Neural Networks
3. Towards Unobtrusive and Invisible Watermarking for Neural Networks
4. Customizable Watermarking for Neural Networks
5. Reverse Engineering Watermarked Neural Networks
6. Understanding and Exploiting Watermarking Attacks on Neural Networks
7. Combatting Watermarking Attacks on Neural Networks
8. Customizable Watermarking for Neural Networks
9. Customizable Watermarking for Neural Networks
10. Customizable Watermarking for Neural Networks
11. Customizable Watermarking for Neural Networks
12. Customizable Watermarking for Neural Networks
13. Customizable Watermarking for Neural Networks
14. Customizable Watermarking for Neural Networks
15. Customizable Watermarking for Neural Networks
16. Customizable Watermarking for Neural Networks
17. Customizable Watermarking for Neural Networks
18. Customizable Watermarking for Neural Networks
19. Customizable Watermarking for Neural Networks
```

```
Here is the list of 20 ideas for the title of a paper on watermarking language models:

1. "Understanding Watermarking Techniques for Language Models"
2. "A Comprehensive Study on Watermarking for Language Models"
3. "Watermarking Techniques for Optimizing Language Models"
4. "A Survey of Watermarking Techniques for Language Models"
5. "Exploring the Potential of Watermarking for Language Models"
6. "Implementing Watermarking for Language Model Optimization"
7. "Watermarking Strategies for Enhancing Language Models"
8. "Investigating the Possibilities of Watermarking for Language Models"
9. "Advanced Watermarking Approaches for Language Models"
10. "Exploring the Use of Watermarking for Language Model Validation"
11. "Understanding the Benefits of Watermarking for Language Model Ensemble"
12. "Leveraging Watermarking Techniques for Enhanced Language Model Performance"
13. "Enhancing Language Models with Watermarking"
14. "Evaluating the Impact of Watermarking Techniques on Language Models"
15. "Analyzing the Feasibility of Watermarking for Language Model Comparison"
16. "Exploring the Possibilities of Watermarking for Optimizing Language Models"
17. "Exploiting Watermarking to Enhance Language Model Accuracy"
18. "Advantages of Using Watermarking for Validating Language Models"
19. "The Promise of Watermarking for Evaluating Language Model Performance"
```

(a) `EXP-hash`          (b) `EXP`

Figure 14: Example responses from Alpaca-7B to the prompt: "Give me 20 ideas for the title of a paper on watermarking language models." We generate (a) by hashing the previous token to determine the inputs to the `EXP` decoder, while (b) is a regular sample from our `EXP` strategy. Hashing causes the model to degenerate into repetition.

then prompt Alpaca-7B with requests for various kinds of lists. Because Alpaca-7B tends to separate items in lists by the same recurring token, e.g., a comma or a newline character, and because this recurring token determines the next token, for $k = 1$ the lists degenerate into repetition (Figure 14).[13]

From inspection, hashing with $k > 1$ substantially improves the quality of samples; however, even using $k = 4$ can sometimes produce noticeably repetitive text. We reiterate that while increasing $k$ may improve sample quality by making the distortions of watermarked text less noticeable, doing so harms the robustness of the watermark (e.g., replacing just 20% of the tokens would suffice to evade detection for $k = 4$). Moreover, using a more robust hash function does not avoid this trade-off between robustness and distortion-freeness, as there is a direct trade-off between the likelihood of a hash collision and the robustness of the hash. In addition to Figure 14, we include more examples (for both $k = 1$ and $k = 4$) and different prompts in Appendix D.5.5 and our code release.

## 4 Discussion

In this paper, we give the first distortion-free watermarking strategies for language models that are robust to editing and/or cropping. The key idea underpinning our approach is to leverage methods for robust sequence alignment to align a putative watermarked text to a watermark key sequence which the LM provider uses to generate watermarked text. The statistical power of our watermarks improves exponentially with respect to the length of the text and diminishes only linearly with respect to the length of the watermark key sequence.

The core assumption underlying watermarking is that the LM provider and the watermark detector coordinate by sharing information in advance, e.g., a watermark key. Indeed, the main inherent limitation of watermarking is that the detector must trust the LM provider to faithfully apply the watermark when generating text. A second limitation, which is not inherent to watermarking language models but does presently apply to all known watermarks, is that the LM provider cannot release the model weights, since then users could simply query the model directly instead of through the LM provider. Planting robust watermarks directly into the weights of a language model without degrading the quality of the model is an important direction for future work.

---

[13]The authors would like to pat themselves on the back by drawing the reader's attention to the fact that the title of this paper is not among those suggested by Alpaca-7B.

### 4.1   Trade-offs among watermarks

Hashing-based watermarks (Aaronson, 2023; Kirchenbauer et al., 2023; Christ et al., 2023) incur a direct trade-off between the degree of distortion versus robustness: larger hash windows reduce distortion but hurt robustness. We avoid sacrificing distortion-freeness for robustness by choosing to formulate watermark detection as a sequence alignment problem; however, this design choice introduces a new trade-off: the computational complexity of our watermark detection algorithms grows linearly with the length of the watermark key sequence. In contrast, the complexities of the watermark detection algorithms of both Christ et al. (2023) and also Aaronson (2023) and Kirchenbauer et al. (2023) depend (in essence) only on the length of the input text. Just as the watermark key length imposes a cap on both the total number of distortion-free watermarked tokens the LM provider may generate for a single query as well as the expected total they can generate across multiple queries before reusing a part of the key sequence, the the window size imposes a cap on the number of (distortion-free) tokens one can expect to generate using a hashing-based watermark without incurring a hash collision. Whether this apparent tension between detection complexity, robustness and (approximate) distortion-freeness is due to fundamental trade-offs is an interesting open question.

To illustrate how these trade-offs manifest in practice, suppose an LM provider responds with $m = 100$ tokens to a sequence of $T = 10$ user queries (which may be adaptively chosen). Recall from Section 2.2 that we must set the key length $n = \omega(mT^2)$ to achieve approximate distortion-freeness using our watermarks, where here $mT^2 = 10000$; for $n = 10000$, the runtimes[14] of our implementation of the test statistics for ITS, ITS-edit, EXP, EXP-edit are $0.004 \pm 0.0002$, $0.60 \pm 0.01$, $2.20 \pm 0.01$ and $3.21 \pm 0.01$ seconds respectively.[15]

In order to achieve approximate distortion-freeness using a hashing-based watermark in the same setting, the LM provider must set the window size $k$ to be sufficiently large as a function of $m$ and $T$. For example, Christ et al. (2023) argue that the probability of incurring the same sequence of tokens twice decays exponentially with the observed entropy of the token sequence (which will depend on the previous tokens); thus, they dynamically adjust the window size $k$ during generation to ensure the observed entropy of the constituent tokens in the window is sufficiently large such that the probability of a hash collision with any of the other windows is negligible across all $mT$ tokens. Specifically, letting $h$ denote the expected observed per-token entropy (i.e., log-probability), the typical window size for a hashing-based watermark would need to be at least $k \approx \frac{\log mT}{h}$ in order to ensure approximate distortion-freeness in our setting. Concretely, in the setting of Section 3.3, we have $h \approx 0.70$ for Alpaca-7B, in which case such a hashing-based watermark would not be robust to replacing more than roughly 10% of watermarked tokens.

### 4.2   Recommendations in practice and combining watermarks

We conclude with some salient recommendations for practitioners aiming to watermark their deployed language models. First, though in principle the length of the watermark key sequence $n$—which recall imposes a cap on the total number of distortion-free watermarked tokens the LM provider can generate—can grow (nearly) exponentially in the block size $k$ of the test statistic while still enabling watermark detection from as few as $k$ tokens, in practice we find that using a fairly small watermark key sequence (e.g., $n = 256$) does not noticeably affect the quality of watermarked text (i.e., even when generating more than $n$ tokens total) while allowing for fast detection and improved robustness. In settings where robustness is important (e.g., discouraging students from using a language model for homework assistance) we recommend practitioners use our EXP-edit watermark, as it is by far the most robust watermark of those we tested. Meanwhile, in settings where throughput of detection is important (e.g., scrubbing synthetic text from a large training corpus), we recommend practitioners use our ITS watermark: its detection is essentially an instance of maximum inner-product search, a problem for which there exist various fast implementations and indexing structures (e.g., via vector databases).

---

[14] We report average runtimes and the associated standard deviations across 5 calls on an Apple M2 Macbook Pro Laptop. We include benchmarking scripts with our code release.

[15] In principle, running detect requires recomputing the test statistic for each resampled watermark key in order to obtain an exact $p$-value. However, as we discuss in Section D and Appendix D.3, we can avoid this recomputation and still obtain approximate $p$-values with respect to some reference distribution of unwatermarked text, in which case we need only compute the single test statistic once (using the original watermark key) during watermark detection (Algorithm 5).

Finally, we remark that for certain hashing-based watermarks we can combine our watermark with the hashing-based watermark to generate watermarked text that is detectable using either of the two corresponding watermark detection algorithms. For example, we can use the hashing-based watermark of Kirchenbauer et al. (2023), which biases the distribution of the next token by upweighting certain logits over others, to determine the distribution of the next token and then use our watermarks to sample from this next token distribution. One can then later determine which detection procedure to run for a collection of putative watermarked text depending on whether throughput or robustness is a higher priority. Similarly, we can combine our watermarks with the watermark of Christ et al. (2023) by alternating tokens between the watermarks (and only hashing the alternate tokens). The resulting watermarked text will be approximately distortion-free since both watermarks are approximately distortion-free. One can choose between the two detection procedures to optimize precision versus recall in adversarial settings; in particular, the watermark of Christ et al. (2023) is hard to spoof (due to its multi-query undetectability guarantee) while our watermarks are harder to remove (due to our robustness guarantees). Exploring such combinations of watermarks with complementary strengths is an exciting direction for future work.

## Acknowledgement

We thank Saminul Haque, Gary Cheng and Padma Kuditipudi for pointing out errors in preliminary drafts of this work and for their helpful feedback in general. This work is supported by an Open Philanthropy Project Award (OpenPhil) and an NSF Frontier Award (NSF Grant no. 1805310).

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

## A  Proof of Lemma 2.2

*Proof.* To show the claim, we first lower bound the probability that $Y = Y'$. In particular,

$$
\begin{aligned}
\mathbb{P}(Y = Y') &= \sum_y \mathbb{P}(Y = y)\mathbb{P}(Y' = y) \\
&= \sum_y \mathbb{P}(Y = y) \prod_{i \in [m]} p(y_i \mid y_{:i-1}) \\
&= \sum_y \mathbb{P}(Y = y) \prod_{i \in [m]} (1 - (1 - p(y_i \mid y_{:i-1}))) \\
&\overset{(\star)}{\geq} \sum_{y \in \mathcal{V}_c} \mathbb{P}(Y = y) \exp\left(-c \sum_{i \in [m]} 1 - p(y_i \mid y_{:i-1})\right) \\
&\geq \mathbb{E}\left[\exp\left(-cm\alpha(Y)\right) \mathbf{1}\{Y \in \mathcal{V}^c\}\right],
\end{aligned}
$$

where $(\star)$ follows from $\exp(-cx) \leq 1 - x$ for $0 \leq x \leq 1 - \exp(-c/2)$. It then follows immediately that we can bound the total variation distance between the joint distributions of $(Y, \xi)$ and $(Y', \xi)$ by

$$
\begin{aligned}
D_{TV}((Y, \xi)\|(Y', \xi)) &\leq \mathbb{P}((Y, \xi) \neq (Y', \xi)) \\
&\leq 1 - \mathbb{E}\left[\exp\left(-cm\alpha(Y)\right) \mathbf{1}\{Y \in \mathcal{V}^c\}\right].
\end{aligned}
$$

Observe for any event $A$ that

$$
D_{TV}((Y, \xi)\|(Y', \xi)) \geq \mathbb{P}((Y, \xi) \in A) - \mathbb{P}((Y', \xi) \in A),
$$

and thus, combining the previous two displays, we have

$$
\begin{aligned}
\mathbb{P}((Y, \xi) \in A) + \mathbb{P}((Y', \xi) \notin A) &\geq \mathbb{P}((Y, \xi) \in A) + \mathbb{P}((Y, \xi) \notin A) - D_{TV}((Y, \xi)\|(Y', \xi)) \\
&\geq \mathbb{E}\left[\exp\left(-cm\alpha(Y)\right) \mathbf{1}\{Y \in \mathcal{V}^c\}\right].
\end{aligned}
$$

The desired result thus follows from letting $A$ be the event that $h$ predicts $-1$. $\square$

## B  Analysis of inverse transform sampling

We first introduce the following supporting lemma. Recall $C_0 = \mathrm{Var}(\eta(\mathrm{Unif}([N])))$ for $\eta(i) = (i-1)/(N-1)$.

**Lemma B.1.** *Let* $\mu \in \Delta([N])$. *Let* $(U, \pi) \sim \text{Unif}([0,1]) \times \text{Unif}(\Pi)$ *and* $Y = \Gamma((U, \pi), \mu)$. *Then* $\frac{1}{C_0} \text{Cov}(U, \eta(\pi(Y)) \mid Y) = 1 - \mu(Y)$ *almost surely.*

*Proof.* We first characterize the conditional distribution of $\pi$ given $Y$ and the conditional distribution of $U$ given both $\pi$ and $Y$, where recall $\pi$ and $Y$ are discrete. Applying Bayes' formula and Theorem 1, we have

$$\mathbb{P}(\pi \mid Y) = \frac{\mathbb{P}(Y \mid \pi)\mathbb{P}(\pi)}{\mathbb{P}(Y)} \stackrel{(\star)}{=} \frac{\mu(Y)\mathbb{P}(\pi)}{\mathbb{P}(Y)} = \mathbb{P}(\pi). \tag{7}$$

Also, defining the interval

$$I(Y, \pi) := [\mu(\{y : \pi(y) < \pi(Y)\}), \mu(\{y : \pi(y) \leq \pi(Y)\})],$$

for any interval $I \subset [0, 1]$ we have

$$\mathbb{P}(U \in I \mid Y, \pi) \stackrel{(a)}{=} \frac{\mathbb{P}(Y \mid U \in I, \pi)\mathbb{P}(U \in I)\mathbb{P}(\pi)}{\mu(Y)\mathbb{P}(\pi)} \stackrel{(b)}{=} \frac{|I \cap I(Y, \pi)|}{\mu(Y)} \stackrel{(c)}{=} \frac{|I \cap I(Y, \pi)|}{|I(Y, \pi)|}, \tag{8}$$

where $(a)$ follows from Bayes' formula and the independence of $U$ and $\pi$; $(b)$ follows from the definition (1) of the decoder $\Gamma$; and $(c)$ follows from $I(Y, \pi) \subset [0, 1]$ having width equal to $\mu(Y)$. The displays (7) and (8) respectively imply $\pi \mid Y \sim \text{Unif}(\Pi)$ and $U \mid \pi, Y \sim \text{Unif}(I(Y, \pi))$, from which it follows that

$$\mathbb{E}[U \mid Y, \pi(Y)] = \mathbb{E}\left[\mu(\{y : \pi(y) < \pi(Y)\}) + \frac{|I(Y, \pi)|}{2} \,\middle|\, Y, \pi(Y)\right]$$

$$= \frac{(\pi(Y) - 1)(1 - \mu(Y))}{n - 1} + \frac{\mu(Y)}{2}$$

$$= 1/2 + (\eta(\pi(Y)) - 1/2)(1 - \mu(Y)).$$

By symmetry, we have $\mathbb{E}[U] = \mathbb{E}[\eta(\pi(Y))] = 1/2$, the former because $\mathbb{P}(Y \mid U) = \mathbb{P}(Y \mid 1 - U)$ for any $U$ and the latter because recall $\pi \mid Y$ is uniform over $\Pi$. Thus, marginalizing the preceding display over $\pi(Y)$ gives

$$\text{Cov}(U, \eta(\pi(Y)) \mid Y) = \mathbb{E}[(U - 1/2)(\eta(\pi(Y)) - 1/2) \mid Y]$$

$$= (1 - \mu(Y))\text{Var}(\eta(\pi(Y)) \mid Y),$$

from which the desired result follows immediately from recalling $\pi(Y) \mid Y \sim \text{Unif}([N])$ and the definition of the constant $C_0$. □

## B.1 Proof of Lemma 2.3

*Proof.* Recall by definition

$$d(Y_i, \xi_i) = -(U_i - 1/2) \cdot (\eta(\pi_i(Y_i)) - 1/2),$$

where (as in the proof of Lemma B.1) we have $\mathbb{E}[U_i \mid Y] = \mathbb{E}[\eta(\pi_i(Y_i)) \mid Y] = 1/2$. Lemma B.1 thus implies $\mathbb{E}[d(Y_i, \xi_i) \mid Y] = -C_0 \cdot (1 - p(Y_i \mid Y_{:i-1}))$, while trivially $\mathbb{E}[d(Y_i, \xi'_j) \mid Y] = 0$ as $Y$ and $\xi'$ are independent. The result follows immediately. □

## B.2 Proof of Lemma 2.4

We prove the following more general result, from which Lemma 2.4 follows as a corollary.

**Lemma B.2.** *Let* $m, n \in \mathbb{N}$ *with* $n \geq m$, *where* $m$ *is the generation length and* $n$ *is the watermark key length. Define the decoder* $\Gamma$ *by equation* (1), *alignment score* $d$ *by equation* (2), *and* $\phi$ *by Algorithm 3 with block size* $k \leq m$. *Let* $\xi, \xi' \stackrel{i.i.d.}{\sim} \text{Unif}(\Xi^n)$ *with* $Y = \texttt{generate}(\xi; n, p, \Gamma)$. *Let* $\widetilde{Y}$ *be a substring of* $Y$ *of length at least* $k$ *that is conditionally independent of* $\xi$ *and* $\xi'$ *given* $Y$, *i.e.,* $\widetilde{Y} = Y_{\tau+1:\tau+\ell}$ *for* $\ell \geq k$. *Then for* $\widehat{\alpha} := 1 - \frac{1}{k}\sum_{i=\tau+1}^{\tau+k} p(Y_i \mid Y_{:i-1})$, *almost surely*

$$\mathbb{P}(\phi(\widetilde{Y}, \xi') \leq \phi(\widetilde{Y}, \xi) \mid \widetilde{Y}, Y) \leq 2n \exp\left(-kC_0^2\widehat{\alpha}^2/2\right).$$

*Proof.* Recall by definition

$$d(y, (u, \pi)) = -\sum_{i=1}^{\texttt{len}(y)} (u_i - 1/2) \cdot (\eta(\pi_i(y_i)) - 1/2), \tag{9}$$

Lemma 2.3 and the conditional independence of $\tau$ and $\xi$ given $Y$ imply for any $j \in [n]$ that

$$\mathbb{E}[d(\widetilde{Y}_{1:k}, \xi'_{(j+1:j+k)\%n}) \mid Y, \widetilde{Y}] - \mathbb{E}[d(\widetilde{Y}_{1:k}, \xi_{\tau+1:\tau+k}) \mid Y, \widetilde{Y}] = kC_0\widehat{\alpha}.$$

Each summand in equation (9) lies between $-1/4$ and $1/4$, and also $(U_i, \pi_i)$ is conditionally independent of $U_{-i}$ and $\pi_{-i}$ given $Y$. Thus, Hoeffding's inequality (Wainwright, 2019, Proposition 2.5) implies for $j \in [n]$ that

$$\mathbb{P}\left(d(\widetilde{Y}, \xi'_{(j+1:j+k)\%n}) \leq d(\widetilde{Y}, \xi_{\tau+1:\tau+k}) \mid Y, \widetilde{Y}\right)$$
$$\leq \mathbb{P}\left(d(\widetilde{Y}, \xi_{1:m}) - \mathbb{E}[d(\widetilde{Y}, \xi_{1:m})] \geq kC_0\widehat{\alpha}/2 \mid Y, \widetilde{Y}\right)$$
$$+ \mathbb{P}\left(\mathbb{E}[d(\widetilde{Y}, \xi'_{j+1:j+m})] - d(\widetilde{Y}, \xi'_{j+1:j+m}) \geq kC_0\widehat{\alpha}/2 \mid Y, \widetilde{Y}\right)$$
$$\leq 2\exp\left(-mC_0^2\widehat{\alpha}^2/2\right).$$

Recalling the definition of the test statistic $\phi$ via Algorithm 3, the main claim then follows from taking a union bound over all $j \in [n]$. $\qquad\square$

## B.3  Proof of Lemma 2.5

*Proof.* We begin with the following observation for a single token.

**Observation B.1.** *Let $P \in \Delta([N])$. Let $(U, \pi) \sim \mathrm{Unif}([0, 1]) \times \mathrm{Unif}(\Pi)$ and $Y = \Gamma((U, \pi), P)$. Let $\widetilde{Y} \in [N]$ be conditionally independent of $(U, \pi)$ given $Y$. If $\widetilde{Y} \neq Y$, then almost surely*

$$\mathrm{Cov}(U, \eta(\pi(\widetilde{Y})) \mid Y, \widetilde{Y}) = -\frac{1}{N-1}\mathrm{Cov}(U, \eta(\pi(Y)) \mid Y, \widetilde{Y}).$$

*Proof of Observation B.1.* Observe the conditional distribution of $\pi(\widetilde{Y})$ given $Y$ is uniform over $[N]\setminus\{\pi(Y)\}$. Let $X$ be a random variable that is equal to $\eta(\pi(Y))$ with probability $1/N$ and otherwise equal to $\eta(\pi(\widetilde{Y}))$. Observe $X$ is independent of $Y$ and thus also $U$ by assumption—in particular, $(N-1)X+1 \mid Y \sim \mathrm{Unif}([N])$ irrespective of the value of $Y$. The claim thus follows from rearranging terms in the equality

$$0 = \mathrm{Cov}(U, X \mid Y, \widetilde{Y}) = \frac{1}{N}\mathrm{Cov}(U, \eta(\pi(Y)) \mid Y, \widetilde{Y}) + \frac{N-1}{N}\mathrm{Cov}(U, \eta(\pi(\widetilde{Y})) \mid Y, \widetilde{Y}).$$

$\qquad\square$

Lemma 2.3 and Observation B.1 together imply for any $j \in [n]$ that

$$\mathbb{E}[d(\widetilde{Y}, \xi'_{j+1:j+m}) \mid \widetilde{Y}, Y] - \mathbb{E}[d(\widetilde{Y}, \xi_{1:m}) \mid \widetilde{Y}, Y] = mC_0\widetilde{\alpha}(Y, \widetilde{Y}),$$

i.e., by adding the two results together using Observation B.1 to account for the influence of each substituted token on the expectation. Using the same concentration argument as in the proof of Theorem 2.4, we then have

$$\mathbb{P}\left(d(\widetilde{Y}, \xi'_{j+1:j+m}) \leq d(\widetilde{Y}, \xi_{1:m}) \mid \widetilde{Y}, Y\right)$$
$$\leq \mathbb{P}\left(d(\widetilde{Y}, \xi_{1:m}) - \mathbb{E}[d(\widetilde{Y}, \xi_{1:m})] \geq m\widetilde{\alpha}(Y, \widetilde{Y})/2 \mid \widetilde{Y}, Y\right)$$
$$+ \mathbb{P}\left(\mathbb{E}[d(\widetilde{Y}, \xi'_{j+1:j+m})] - d(\widetilde{Y}, \xi'_{j+1:j+m}) \geq m\widetilde{\alpha}(Y, \widetilde{Y})/2 \mid \widetilde{Y}, Y\right)$$
$$\leq 2\exp\left(-mC_0^2\widetilde{\alpha}(Y, \widetilde{Y})^2/2\right).$$

Recalling the definition of the test statistic $\phi$ via Algorithm 3, the main claim then follows from taking a union bound over all $j \in [n]$ and recalling $k = m$ by assumption. $\qquad\square$

### B.4 Proof of Lemma 2.6

*Proof.* We begin with the following useful facts about edit distance. Throughout, let $\mathcal{S}(y)$ denote the set of substrings of a string $y \in \mathcal{V}^*$, including the empty string.

**Observation B.2.** *Let $y, \widetilde{y} \in \mathcal{V}^*$. Then $d_{\text{edit}}(y, \widetilde{y})$ is the length of the smallest sequence of insertion and/or deletion operations to obtain $\widetilde{y}$ from $y$.*

*Proof of Observation B.2.* We proceed via induction on the sum $\texttt{len}(y) + \texttt{len}(\widetilde{y})$. The base case where $y$ and $\widetilde{y}$ are both empty is trivial. Now suppose the claim holds all strings whose lengths sum to at most $\texttt{len}(y) + \texttt{len}(\widetilde{y}) - 1$. Recalling the definition of $d_{\text{edit}}$ (Definition 4), there are three cases.

First, suppose $d_{\text{edit}}(y, \widetilde{y}) = d_{\text{edit}}(y_{2:}, \widetilde{y}_{2:})$. Then by induction there exists a sequence of $d_{\text{edit}}(y, \widetilde{y})$ insertion and/or deletion operations to obtain $\widetilde{y}_{2:}$ from $y_{2:}$. Because $y_1 = \widetilde{y}_1$, the same sequence suffices to obtain $\widetilde{y}$ from $y$ and thus the claim follows.

Second, suppose $d_{\text{edit}}(y, \widetilde{y}) = 1 + d_{\text{edit}}(y_{2:}, \widetilde{y})$. Again by induction, there exists a sequence of $d_{\text{edit}}(y, \widetilde{y}) - 1$ insertion and/or deletion operations to obtain $\widetilde{y}$ from $y_{2:}$. It follows immediately (i.e., by first deleting $y_1$) there exists a sequence of $d_{\text{edit}}(y, \widetilde{y})$ such operations to obtain $\widetilde{y}$ from $y$, and so the claim holds.

The third case follows by symmetry with the second case. $\square$

**Observation B.3.** *Let $y, \widetilde{y} \in \mathcal{V}^*$. Then for any $\tau < \texttt{len}(y)$, we have*

$$d_{\text{edit}}(y, \widetilde{y}) \geq \min_{y' \in \mathcal{S}(\widetilde{y})} d_{\text{edit}}(y_{:\tau}, y') + \min_{y' \in \mathcal{S}(\widetilde{y})} d_{\text{edit}}(y_{\tau+1:}, y').$$

*Proof of Observation B.3.* Observation B.2 implies there exists a sequence of $d_{\text{edit}}(y, \widetilde{y})$ insertion and/or deletion operations to obtain $\widetilde{y}$ from $y$. We may partition this sequence of operations into sequences based respectively on whether they occur on $y_{:\tau}$ or $y_{\tau+1:}$. Let $\widetilde{y}_{\text{pre}}$ be the result of performing the first sequence of operations on $y_{:\tau}$ and let $\widetilde{y}_{\text{suf}}$ be the result of performing the second sequence of operations on $y_{\tau+1:}$. Then $\widetilde{y}$ is the concatenation of $\widetilde{y}_{\text{pre}}$ and $\widetilde{y}_{\text{suf}}$, and so the claim follows from the fact that

$$\begin{aligned}
d_{\text{edit}}(y, \widetilde{y}) &= d_{\text{edit}}(y_{:\tau}, \widetilde{y}_{\text{pre}}) + d_{\text{edit}}(y_{\tau+1:}, \widetilde{y}_{\text{suf}}) \\
&\geq \min_{y' \in \mathcal{S}(\widetilde{y})} d_{\text{edit}}(y_{:\tau}, y') + \min_{y' \in \mathcal{S}(\widetilde{y})} d_{\text{edit}}(y_{\tau+1:}, y').
\end{aligned}$$

$\square$

**Observation B.4.** *Let $y, \widetilde{y} \in \mathcal{V}^*$ and $\xi \in \Xi^*$. Then $d_\gamma(y, \xi) \leq \gamma d_{\text{edit}}(y, \widetilde{y}) + d_\gamma(\widetilde{y}, \xi)$.*

*Proof of Observation B.4.* The case $d_{\text{edit}}(y, \widetilde{y}) = 0$ is trivial as we then have $y = \widetilde{y}$. Now suppose $d_{\text{edit}}(y, \widetilde{y}) = 1$, and let $i$ be the first index such that $y_i \neq \widetilde{y}_i$. Then, unrolling the recursive definition of $d_\gamma(\widetilde{y}_{i:}, \xi_{j:})$, there must exist $c \in \mathbb{R}$ and an index $j$ such that both $d_\gamma(\widetilde{y}, \xi) = c + d_\gamma(\widetilde{y}_{i:}, \xi_{j:})$ and $d_\gamma(y, \xi) \leq c + d_\gamma(y_{i:}, \xi_{j:})$. Moreover, from the definition of edit distance, either $y_{i+1:} = \widetilde{y}_{i:}$ or vice versa.

We claim $d_\gamma(y_{i:}, \xi_{j:}) \leq d_\gamma(\widetilde{y}_{i:}, \xi_{j:}) + \gamma$. If $y_{i+1:} = \widetilde{y}_{i:}$, then the claim obtains as

$$\begin{aligned}
d_\gamma(y_{i:}, \xi_{j:}) &\leq d_\gamma(y_{i+1:}, \xi_{j:}) + \min_{\xi' \in \Xi} d_0(y_i, \xi') + \gamma \\
&\overset{(\star)}{\leq} d_\gamma(y_{i+1:}, \xi_{j:}) + \gamma \\
&= d_\gamma(\widetilde{y}_{i:}, \xi_{j:}) + \gamma,
\end{aligned}$$

with $(\star)$ following from the fact that $d_0(y_i, \xi') = 0$ for $\xi' = (1/2, \pi)$ irrespective of $y_i$ and $\pi$.

Otherwise, if $y_{i:} = \widetilde{y}_{i+1:}$, then from unrolling the recursive definition of $d_\gamma(\widetilde{y}_{i:}, \xi_{j:})$ there must exist some index $j' \geq j$ such that either

$$d_\gamma(\widetilde{y}_{i:}, \xi_{j:}) = d_\gamma(\widetilde{y}_{i+1:}, \xi_{j':}) + \gamma + \min_{\xi' \in \Xi} d_0(\widetilde{y}_i, \xi') + \sum_{j \leq \ell < j'} \gamma + \min_{y' \in \mathcal{V}} d_0(y', \xi_\ell)$$

or

$$d_\gamma(\widetilde{y}_{i:}, \xi_{j:}) = d_\gamma(\widetilde{y}_{i+1:}, \xi_{j'+1:}) + d_0(\widetilde{y}_i, \xi_{j'}) + \sum_{j \leq \ell < j'} \gamma + \min_{y' \in \mathcal{V}} d_0(y', \xi_\ell).$$

In the first case, we have $\gamma + \min_{\xi' \in \Xi} d_0(\widetilde{y}_i, \xi') > 0$ since $\gamma > 1/2$ by assumption, and so the claim follows as

$$d_\gamma(y_{i:}, \xi_{j:}) \leq d_\gamma(y_{i:}, \xi_{j':}) + \sum_{j \leq \ell < j'} \gamma + \min_{y' \in \mathcal{V}} d_0(y', \xi_\ell)$$

$$= d_\gamma(\widetilde{y}_{i+1}, \xi_{j':}) + \sum_{j \leq \ell < j'} \gamma + \min_{y' \in \mathcal{V}} d_0(y', \xi_\ell)$$

$$< d_\gamma(\widetilde{y}_{i:}, \xi_{j:}).$$

In the second case, we have $d_0(\widetilde{y}_j)$ the claim follows as

$$d_\gamma(y_{i:}, \xi_{j:}) \leq d_\gamma(y_{i:}, \xi_{j'+1:}) + \sum_{j \leq \ell < j'+1} \gamma + \min_{y' \in \mathcal{V}} d_0(y', \xi_\ell)$$

$$= d_\gamma(\widetilde{y}_{i+1}, \xi_{j'+1:}) + \sum_{j \leq \ell < j'+1} \gamma + \min_{y' \in \mathcal{V}} d_0(y', \xi_\ell)$$

$$\leq d_\gamma(\widetilde{y}_{i:}, \xi_{j:}) + \gamma.$$

Thus, assuming $d_{\text{edit}}(y, \widetilde{y}) \leq 1$, we have shown $d_\gamma(y_{i:}, \xi_{j:}) \leq d_\gamma(\widetilde{y}_{i:}, \xi_{j:}) + \gamma$, from which it follows that $d_\gamma(y, \xi) \leq d_\gamma(\widetilde{y}, \xi) + \gamma$. The general result follows immediately by applying Observation B.2 and summing the bound for a single edit over the (smallest) sequence of edits to obtain $\widetilde{y}$ from $y$. $\qquad\square$

Proceeding with the main proof, define for convenience the quantity

$$\widehat{\alpha}_\tau := \frac{1}{k} \sum_{i=1}^{k} p(Y_{\tau+i} \mid Y_{:\tau+i-1}).$$

Observe

$$\alpha(Y) = \frac{k}{m} \sum_{\tau=0}^{m/k-1} \widehat{\alpha}_{k\tau}, \tag{10}$$

while Observation B.3 together with our assumption that $d_{\text{edit}}(Y, \widetilde{Y}) \leq \varepsilon m$ implies

$$\frac{k}{m} \sum_{\tau=0}^{m/k-1} \min_{Y' \in \mathcal{S}(\widetilde{Y})} d_{\text{edit}}(Y_{k\tau+1:k\tau+k}, Y') \leq k\varepsilon. \tag{11}$$

The displays (10) and (11) together imply there exists an index $\tau$ and $Y' \in \mathcal{S}(\widetilde{Y})$ such that $\widehat{\alpha}_\tau - \frac{1}{k} \min_{Y' \in \mathcal{S}(\widetilde{Y})} d_{\text{edit}}(Y_{\tau+1:\tau+k}, Y') \geq \alpha(Y) - \varepsilon$. Reusing the same concentration argument as in the proof of Theorem 2.4, for $t \geq 0$ we have

$$\mathbb{P}\left(d_0(Y_{\tau+1:\tau+k}, \xi_{\tau+1:\tau+k}) \geq -k\left(C_0\widehat{\alpha}_\tau + t\right) \mid Y\right) \leq \exp\left(-2kt^2\right),$$

and thus from Observation B.4 it follows that

$$\mathbb{P}\left(d_\gamma(Y', \xi_{\tau+1:\tau+k}) \geq -k\left(C_0\alpha(Y) - \gamma\varepsilon + t\right) \mid \widetilde{Y}, Y\right) \leq \exp\left(-2kt^2\right).$$

Letting $t = (C_0\alpha - \gamma\varepsilon)/2$ and recalling the definition of the test statistic, we have

$$\mathbb{P}\left(\phi(\widetilde{Y}, \xi) \geq -k\left(C_0\alpha(Y) - \gamma\varepsilon\right)/2 \mid \widetilde{Y}, Y\right) \leq \exp\left(-k(C_0\alpha(Y) - \gamma\varepsilon)_+^2/2\right). \tag{12}$$

All that remains to bound the probability of $\phi(\widetilde{Y}, \xi')$ exceeding the threshold from the above display. To this end, define the set-valued map $\mathcal{N}_\beta(y) := \{y' : d_{\text{edit}}(y, y') \leq \beta/(4\gamma - 1)\}$. Then we make the following observation.

**Observation B.5.** *For any $y \in \mathcal{V}^*$ and $\xi \in \Xi^*$, there exists $y' \in \mathcal{N}_{\mathtt{len}(\xi)}(y)$ such that*

$$d_\gamma(y, \xi) = \gamma \cdot d_{\mathrm{edit}}(y, y') + d_0(y', \xi).$$

*Proof.* We proceed via induction. The base case where $y$ and $\xi$ both have length 1 follows trivially by taking $y' = y$; in particular, $\gamma > 1/2$ implies $d(y, \xi) \leq \gamma + \min_{y'} d(y', \xi)$ and likewise $d(y, \xi) \leq \gamma + \min_{\xi'} d(y, \xi')$. Now suppose the result holds so long as $\mathtt{len}(y) + \mathtt{len}(\xi) \leq n - 1$. We claim that the result must then also hold if the lengths sum to at most $n$.

We prove this inductive claim by considering three exhaustive cases. First, suppose that $d_\gamma(y, \xi) = d_\gamma(y_{2:}, \xi_{2:}) + d(y_1, \xi_1)$. By our induction hypothesis, there exists $\hat{y} \in \mathcal{N}_{\mathtt{len}(\xi)-1}(y_{2:})$ such that $d_\gamma(y_{2:}, \xi_{2:}) = \gamma \cdot d_{\mathrm{edit}}(y_{2:}, \hat{y}) + d(\hat{y}, \xi_{2:})$. The desired result then obtains with $y'$ as the concatenation of $y_1$ and $\hat{y}$. Second, suppose $d_\gamma(y, \xi) = d_\gamma(y, \xi_{2:}) + \min_{\xi' \in \Xi} d(y_1, \xi') + \gamma$. By our induction hypothesis, there exists $\hat{y} \in \mathcal{N}_{\mathtt{len}(\xi)=1}(y)$ such that $d_\gamma(y_{2:}, \xi) = \gamma \cdot d_{\mathrm{edit}}(y_{2:}, \hat{y}) + d(\hat{y}, \xi_{2:})$. The result obtains with $y' = \hat{y}$. Finally, suppose $d_\gamma(y, \xi) = d_\gamma(y_{2:}, \xi) + d(y'', \xi_1) + \gamma$ for some $y'' \in \mathcal{V}$. By our induction hypothesis, there exists $\hat{y} \in \mathcal{N}_{\mathtt{len}(\xi)-1}(y)$ such that $d_\gamma(y_{2:}, \xi) = \gamma \cdot d_{\mathrm{edit}}(y_{2:}, \hat{y}) + d(\hat{y}, \xi)$. The result then obtains by concatenating $y''$ with $\hat{y}$. $\qquad\square$

Let $\mathcal{I}_j := \{(j+i)\%n\}_{i=1}^k$. For any $0 \leq i \leq \mathtt{len}(\widetilde{Y}) - k$ and $j \in [n]$, Observations B.4 and B.5 together imply that

$$d_\gamma(\widetilde{Y}_{i+1:i+k}, \xi'_{\mathcal{I}_j}) = \min_{y \in \mathcal{N}_k(\widetilde{Y}_{i+1:i+k})} \gamma \cdot d_{\mathrm{edit}}(\widetilde{Y}_{i+1:i+k}, y) + d_0(y, \xi'_{\mathcal{I}_j}) \tag{13}$$

$$\stackrel{(\star)}{=} \min_{y \in \mathcal{N}_{k/4(\gamma-1)}(\widetilde{Y}_{i+1:i+k})} \gamma \cdot d_{\mathrm{edit}}(y, \widetilde{Y}_{i+1:i+k}) + d_0(y, \xi'_{\mathcal{I}_j}), \tag{14}$$

where $(\star)$ follows from the fact that $d_{\mathrm{edit}}(\widetilde{Y}_{i+1:i+k}, y) > k/4(\gamma - 1)$ implies

$$\gamma \cdot d_{\mathrm{edit}}(\widetilde{Y}_{i+1:i+k}, y) + d_0(y, \xi'_{\mathcal{I}_j}) \geq k/4 > d_0(\widetilde{Y}_{i+1:i+k}, \xi'_{\mathcal{I}_j}),$$

and therefore the minimizer in equation (13) must be an element of $\mathcal{N}_{k/4(\gamma-1)}(\widetilde{Y}_{i+1:i+k})$.

By construction, $\mathcal{N}_\beta(y)$ consists of the set of strings obtainable from $y$ by a sequence of at most $\beta$ insertion and/or deletion operations. Now define another set-valued map $\mathcal{N}_{\beta,-}(y)$ as the restriction of $\mathcal{N}_\beta(y)$ such that we may only insert a particular token into $y$ (which token is immaterial). As the specific identity of each token we insert into $y$ can only influence the value of $d_\gamma$ by $\pm 1/2$, for any $\beta$ it follows that

$$\min_{y \in \mathcal{N}_\beta(\widetilde{Y}_{i+1:i+k})} \gamma \cdot d_{\mathrm{edit}}(y, \widetilde{Y}_{i+1:i+k}) + d_0(y, \xi'_{\mathcal{I}_j}) \geq \min_{y \in \mathcal{N}_{\beta,-}(\widetilde{Y})} d_0(y, \xi'_{\mathcal{I}_j}),$$

and so, letting $\beta = k/4(\gamma - 1)$, from equation (14) we have

$$d_\gamma(\widetilde{Y}_{i+1:i+k}, \xi'_{\mathcal{I}_j}) \geq \min_{y \in \mathcal{N}_{\beta,-}(\widetilde{Y}_{i+1:i+k})} d_0(y, \xi'_{\mathcal{I}_j})$$

Let $\widetilde{Y}(i, \ell)$ denote the $\ell$-th element of $\mathcal{N}_{\beta,-}(\widetilde{Y}_{i+1:i+k})$ for some $\widetilde{Y}$-measurable indexing. From the independence of $\widetilde{Y}$ and $\xi'$, we have $\mathbb{E}[d_0(\widetilde{Y}(i, \ell), \xi_{\mathcal{I}_j}) \mid \widetilde{Y}] = 0$ for any $\ell$ and $j$. The cardinality of $\mathcal{N}_{\beta,-}(\widetilde{Y}_{i+1:i+k})$ is equal to the number of possible combinations of locations for $\beta$ insertion and/or deletion operations on $\widetilde{Y}$, of which there are at most $(k + \beta)^\beta \leq (2k)^\beta$. Thus, applying the same concentration argument as in the proof of Theorem 2.4 and taking a union bound over all $i \leq m - k$, $j \leq n$ and $\ell \leq (2k)^\beta$, we have

$$\mathbb{P}(\phi(\widetilde{Y}, \xi') \leq -\alpha(Y)/2 + \gamma\varepsilon \mid \widetilde{Y}, Y) \leq mn(2k)^{k/(4\gamma-1)} \exp(-kC_0^2(\alpha(Y) - \gamma\varepsilon)_+^2/2). \tag{15}$$

Combining the displays (12) and (15) via another union bound gives the desired result. $\qquad\square$

## C   Analysis of exponential minimum sampling

To prove the main theorems, we introduce the following supporting lemma. The result is well known and we restate it here only for completeness.

**Lemma C.1.** *Let $\mu \in \Delta([N])$ and $\xi \sim \text{Unif}([0,1]^N)$. Then for any $y \in [N]$ we have*

$$\mathbb{P}(\Gamma(\xi,\mu) = y, -\log(\xi_y)/\mu(y) \geq t) = \mu(y)\exp(-t).$$

*Proof.* Suppose $\mu(y) > 0$ as otherwise the claim is trivial. Recalling $\xi_i \overset{\text{i.i.d.}}{\sim} \text{Unif}([0,1])$, for any $\lambda > 0$ we have $-\lambda \log \xi_i \overset{\text{i.i.d.}}{\sim} \text{Exp}(\lambda)$, i.e.,

$$\mathbb{P}(-\lambda \log \xi_i \geq t) = \mathbb{P}(\xi_i \leq \exp(-\lambda t)) = \exp(-\lambda t).$$

Thus, the claim follows as

$$
\begin{aligned}
&\mathbb{P}(\Gamma(\xi,\mu) = y, -\log(\xi_y)/\mu(y) \geq t) \\
&= \mathbb{P}(y = \arg\min_i -\log(\xi_i)/\mu(i), -\log(\xi_y)/\mu(y) \geq t) \\
&\overset{(\star)}{=} \int_{u \geq t} \mu(y)\exp(-\mu(y)u) \cdot \Pi_{i \in \text{supp}(\mu), i \neq y}\mathbb{P}(-\log(\xi_i)/\mu(i) > u) \\
&= \int_{u \geq t} \mu(y)\exp(-\mu(y)u) \cdot \Pi_{i \in \text{supp}(\mu), i \neq y}\exp(-\mu(i)u) \\
&= \mu(y)\int_{u \geq t} \Pi_{i \in \text{supp}(\mu)}\exp(-\mu(i)u) \\
&= \mu(y)\int_{u \geq t}\exp(-u) \\
&= \mu(y)\exp(-t),
\end{aligned}
$$

where in $(\star)$ we use the fact that the density of $-\log(\xi_y)/\mu(y)$ at $u$ is $\mu(y)\exp(-\mu(y)u)$.   $\square$

### C.1   Proof of Theorem 2

*Proof.* The result follows immediately from integrating the result of Lemma C.1 over $t \geq 0$.   $\square$

### C.2   Proof of Lemma 2.7

*Proof.* Lemma C.1 implies $-\log(\xi_i)/p(Y_i \mid Y_{:i-1}) \mid Y \sim \text{Exp}(1)$, and thus $\mathbb{E}[-\log(\xi_i) \mid Y] = p(Y_i \mid Y_{:i-1})$. Meanwhile, as $\xi'_i \sim \text{Unif}([0,1])$ independently of $Y$, we have

$$\mathbb{P}(-\log \xi'_i \geq t \mid Y) = \mathbb{P}(\xi'_i \leq \exp(-t)) = \exp(-t),$$

implying $-\log(\xi'_i) \mid Y \sim \text{Exp}(1)$ and so $\mathbb{E}[-\log(\xi'_i) \mid Y] = 1$. The result follows immediately, recalling $\alpha(Y_{i-1:i}) = 1 - p(Y_i \mid Y_{i-1})$ by definition.   $\square$

### C.3   Proof of Lemma 2.8

We prove the following general result, from which Lemma 2.8 follows as a corollary.

**Lemma C.2.** *Let $m, n \in \mathbb{N}$ with $n \geq m$, where $m$ is the generation length and $n$ is the watermark key length. Define the decoder $\Gamma$ by equation (4), alignment score $d$ by equation (5), and $\phi$ by Algorithm 3 with block size $k \leq m$. Let $\xi, \xi' \overset{\text{i.i.d.}}{\sim} \text{Unif}(\Xi^n)$ with $Y = \texttt{generate}(\xi; n, p, \Gamma)$. Let $\widetilde{Y}$ be a substring of $Y$ of length at least $k$ that is conditionally independent of $\xi$ and $\xi'$ given $Y$, i.e., $\widetilde{Y} = Y_{\tau+1:\tau+\ell}$ for $\ell \geq k$. Then for $\widehat{\alpha} := 1 - \frac{1}{k}\sum_{i=\tau+1}^{\tau+k} p(Y_i \mid Y_{:i-1})$, almost surely*

$$\mathbb{P}(\phi(\widetilde{Y},\xi') \leq \phi(\widetilde{Y},\xi) \mid \widetilde{Y}, Y) \leq 2n\exp\left(-\min\{k\widehat{\alpha}^2/8, k\widehat{\alpha}/4\}\right).$$

*Proof.* Recall by definition

$$d(y, \xi) = -\sum_{i=1}^{\texttt{len}(y)} \log \xi_{i,y_i}.$$

Lemma 2.7 and the conditional independence of $\tau$ and $\xi$ given $Y$ imply for any $j \in [n]$ that

$$\mathbb{E}[d(\widetilde{Y}, \xi'_{(j+1:j+k)\%n}) \mid \widetilde{Y}, Y] - \mathbb{E}[d(\widetilde{Y}, \xi_{\tau+1:\tau+k}) \mid \widetilde{Y}, Y] = k\widehat{\alpha}.$$

From Lemma C.1, we have $-\log \xi_{\tau+i,\widetilde{Y}_i} \mid \widetilde{Y}, Y \sim \mathrm{Exp}(\gamma_i)$ for some $\gamma_i \leq 1$ for all $i \in [m]$. Also, from the independence of $\widetilde{Y}$ and $\xi'$, we have $-\log \xi'_{j,\widetilde{Y}_i} \mid \widetilde{Y}, Y \sim \mathrm{Exp}(1)$ for all $i \in [m]$ and $j \in [n]$. The following observation thus implies $-\log \xi_{i,\widetilde{Y}_i} \mid \widetilde{Y}, Y$ and $-\log \xi'_{j,\widetilde{Y}_i} \mid \widetilde{Y}, Y$ are both $(2,2)$-subexponential random variables.

**Observation C.1.** *Let* $X \sim \mathrm{Exp}(1)$. *Then $X$ is a $(2,2)$ subexponential random variable.*

*Proof of Observation C.1.* For $t < 1/2$, we have

$$\mathbb{E}[e^{t(X-\mathbb{E}[X])}] = \int_0^\infty e^{t(x-1)} e^{-x} \, dx$$
$$\overset{(a)}{=} \frac{e^{-t}}{1-t}$$
$$\overset{(b)}{\leq} (1 - t + t^2)(1 + t + 2t^2)$$
$$\overset{(c)}{\leq} (1 + 2t^2)$$
$$\leq e^{2t^2},$$

where (a) follows from the fact that $t < 1$ (otherwise, the integral would not be finite); (b) follows from Taylor expanding $e^{-t}$ and $1/(1-t)$ and applying the fact that $t < 1/2$ to bound the higher-order terms; and (c) again follows from $t < 1/2$. The claim follows immediately. $\square$

Thus, using the fact that $\xi_i$ is conditionally independent of $\xi_{-i}$ given $Y$, a standard Chernoff bound (Wainwright, 2019, Proposition 2.9) implies for each $j \in [n]$ that

$$\mathbb{P}\left(d(\widetilde{Y}, \xi'_{j+1:j+k}) \leq d(\widetilde{Y}, \xi_{\tau+1:\tau+k}) \mid \widetilde{Y}, Y\right)$$
$$\leq \mathbb{P}\left(d(\widetilde{Y}, \xi_{1:m}) - \mathbb{E}[d(\widetilde{Y}, \xi_{1:m})] \geq k\widehat{\alpha}/2 \mid \widetilde{Y}, Y\right)$$
$$+ \mathbb{P}\left(\mathbb{E}[d(\widetilde{Y}, \xi'_{j+1:j+m})] - d(\widetilde{Y}, \xi'_{j+1:j+m}) \geq k\widehat{\alpha}/2 \mid \widetilde{Y}, Y\right)$$
$$\leq 2\exp\left(-\min\{k\widehat{\alpha}^2/8, k\widehat{\alpha}/4\}\right).$$

Recalling the definition of the test statistic $\phi$ via Algorithm 3, the main claim then follows from taking a union bound over all $j \in [n]$. $\square$

# D   Details of experiments

## D.1   Experimental protocol

In Experiments 1-6, for each watermark we first generate a sequence tokens, decode the tokens into text (i.e., a string) using the appropriate tokenizer for the language model, and then encode the text back into tokens before running `detect`. Each generation is coditioned on a prompt; we obtain the prompts by sampling

documents from the news-like subset of the C4 dataset and truncating the last $m$ tokens. We enforce a minimum prompt size of 50 tokens in all experiments; we skip over any document that is not long enough. The retokenization is not always equal to the original tokens generated by the model;[16] in order to ensure `detect` always receives at least $m$ tokens, we pad its input with special pad tokens (specific to each model's tokenizer). We also initially generate a number of buffer tokens beyond $m$, so in most cases the padding is unnecessary. We set the number of buffer tokens to be 20 in every experiment except for Experiment 5, where we set it to be 100 in order to ensure that even after deleting tokens there are typically still at least $m$ tokens remaining. We always truncate the number of tokens given to `detect` to be at most $m$, irrespective of the number of buffer tokens.

## D.2    Roundtrip translation

In Experiment 6, we perform round-trip translations from English to French and from English to Russian using the OPUS-MT collection of translation models (Tiedemann & Thottingal, 2020; Tiedemann et al., 2022). Specifically, we use the versions of these models hosted on the HuggingfaceHub,[17] associated with the identifiers:

- `Helsinki-NLP/opus-mt-tc-big-en-fr` - English to French,

- `Helsinki-NLP/opus-mt-tc-big-fr-en` - French to English,

- `Helsinki-NLP/opus-mt-en-ru` - English to Russian,

- `Helsinki-NLP/opus-mt-ru-en` - Russian to English.

## D.3    Computing p-values

As we mention previously, to save computation we modify `detect` to use a fixed reference distribution to compute $p$-values. For the sake of concreteness, we give the full pseudocode for the modified version of `detect` in Algorithm 5; in Experiments 1-6, we compute $p$-values using Algorithm 6 to construct the reference distribution using the news-like subset of the C4 dataset as the text distribution.

---

**Algorithm 5:** Watermarked text detection with fixed reference distribution

**Input**    : string $y \in \mathcal{V}^*$, seed sequence $\xi \in \Xi^n$
**Params:** test statistic $\phi$; reference distribution $\{\phi_t\}_{t=1}^T$
**Output:** p-value $\widehat{p} \in [0, 1]$
1  $\widehat{p} \leftarrow \frac{1}{T} \sum_{t=1}^T \mathbf{1}\{\phi(y, \xi) < \phi_t\}$
2  **return** $\widehat{p}$

---

As a sanity check, we include histograms of the $p$-values we compute for nonwatermarked text for each method to verify that they are roughly uniformly distributed on the interval $[0, 1]$ (setting $m = 50$ and sampling prompts from the news-like subset of the C4 dataset, as in Experiment 1). In the cases of `KGW-1.0` and `KGW-2.0`, the distribution is not quite uniform due to the discrete nature of their test statistics.

---

[16]Byte-pair tokenizations of text (used by both the OPT and LLaMA) are not unique, due to the fact that they augment a base vocabulary (e.g., characters) with extra tokens to represent common substrings.
[17]https://huggingface.co/

---

**Algorithm 6:** Reference distribution construction

**Input** : resample size $T \in \mathbb{N}$, text length $m \in \mathbb{N}$, watermark key sequence distribution $\nu \in \Delta(\Xi^n)$
**Params:** test statistic $\phi$; text distribution $P$; minimum prompt length $m_0$
**Output:** reference distribution $\{\phi_t\}_{t=1}^{T} \in \mathbb{R}^T$

**1** $t \leftarrow 1$
**2 while** $t \leq T$ **do**
**3**     $Y \sim P$
**4**     **if** `len`$(Y) \leq m_0 + m$ **then**
**5**        continue
**6**     $\xi^{(t)} \sim \nu$
**7**     $\phi_t \leftarrow \phi(Y_{-m:}, \xi^{(t)})$
**8**     $t \leftarrow t + 1$
**9 return** $\{\phi_t\}_{t=1}^{T}$

---

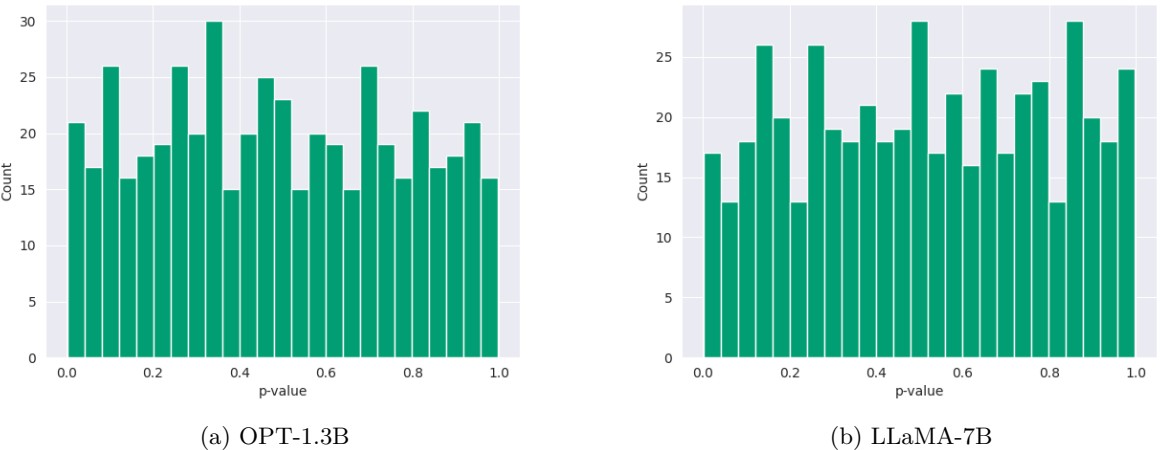

(a) OPT-1.3B             (b) LLaMA-7B

Figure 15: Distribution of $p$-values for nonwatermarked text using `ITS` detector.

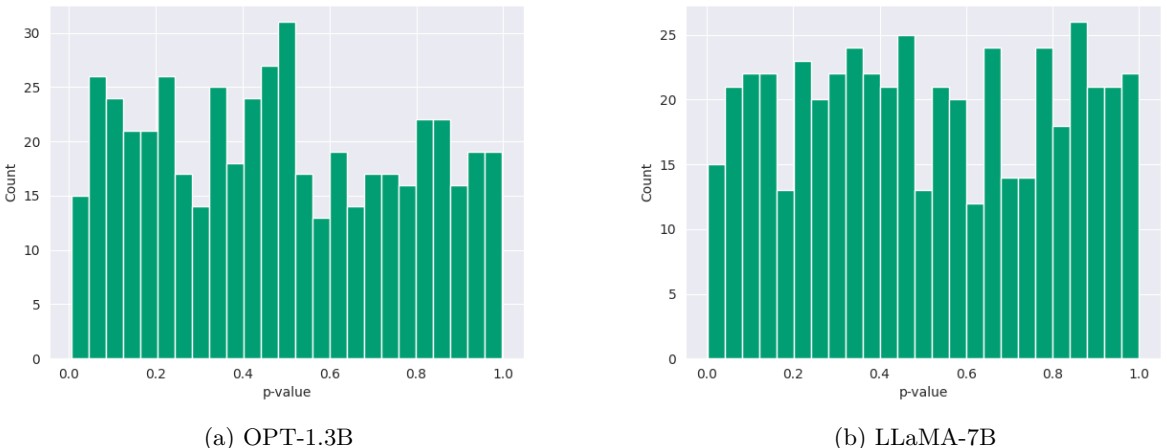

(a) OPT-1.3B             (b) LLaMA-7B

Figure 16: Distribution of $p$-values for nonwatermarked text using `ITS-edit` detector.

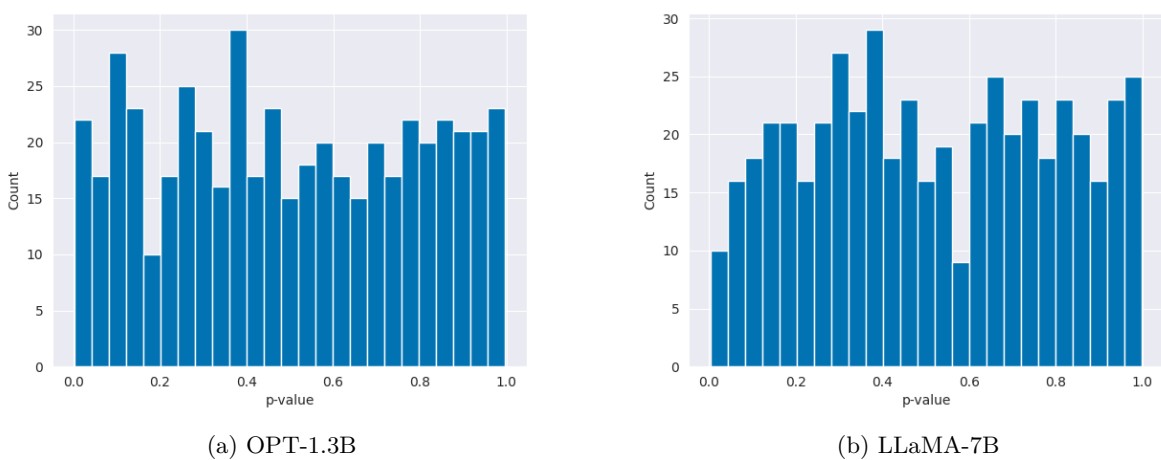

(a) OPT-1.3B                    (b) LLaMA-7B

Figure 17: Distribution of $p$-values for nonwatermarked text using EXP detector.

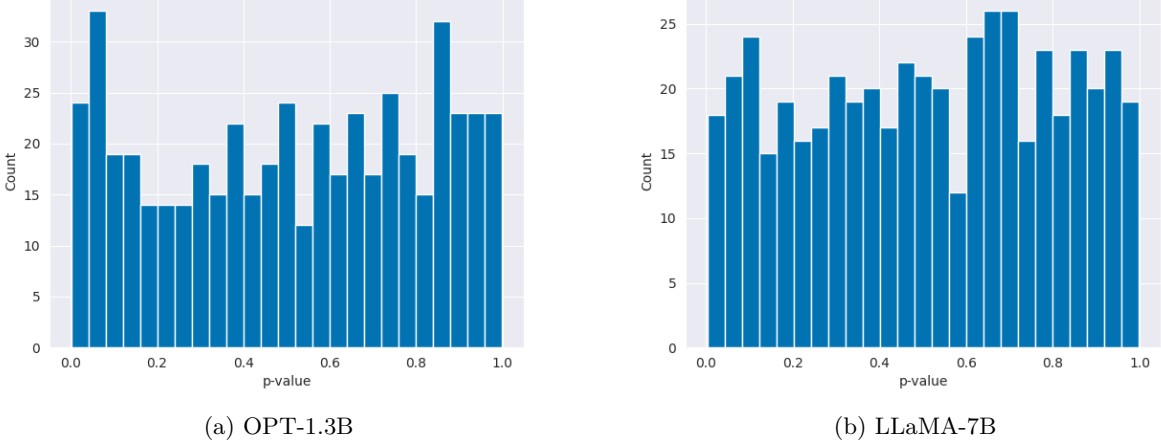

(a) OPT-1.3B                    (b) LLaMA-7B

Figure 18: Distribution of $p$-values for nonwatermarked text using EXP-edit detector.

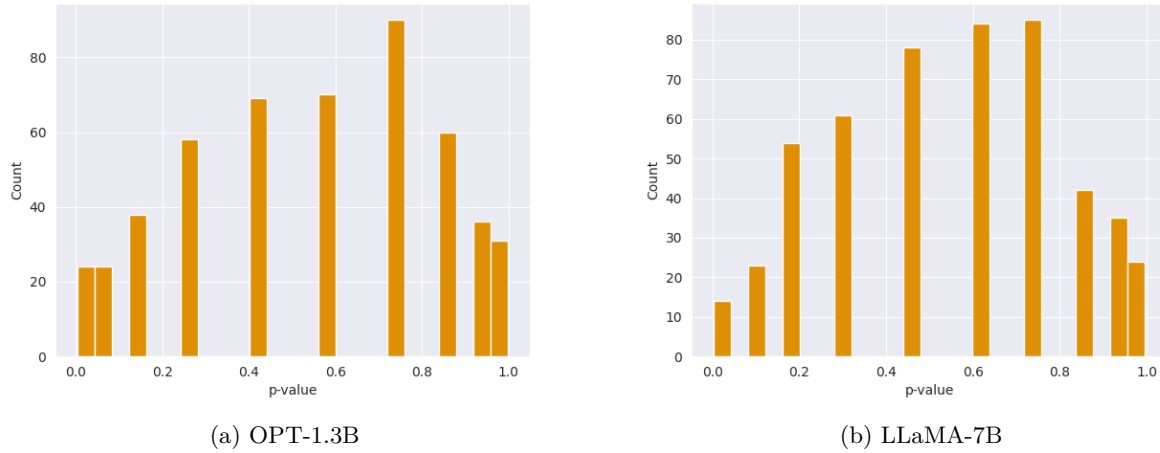

(a) OPT-1.3B

(b) LLaMA-7B

Figure 19: Distribution of $p$-values for nonwatermarked text using `KGW-1.0` detector.

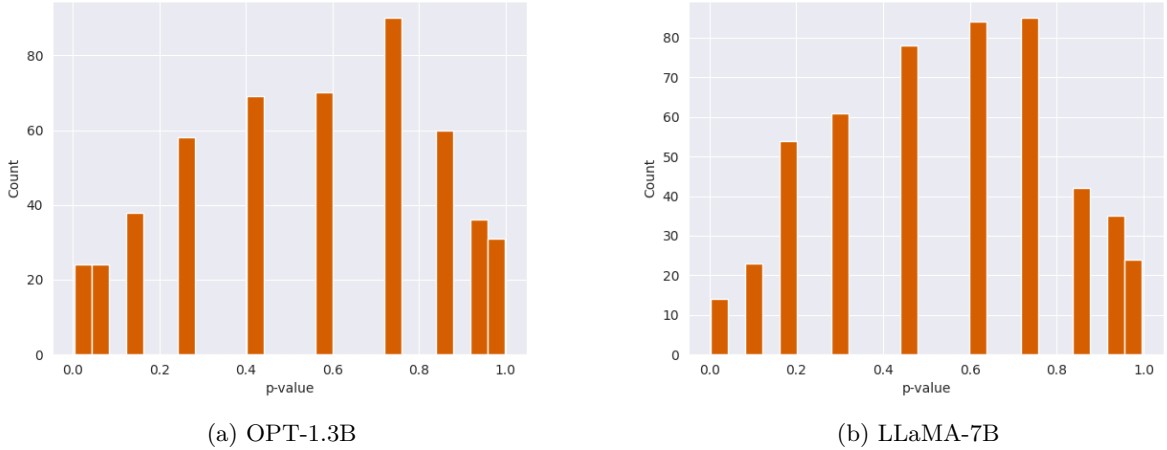

(a) OPT-1.3B

(b) LLaMA-7B

Figure 20: Distribution of $p$-values for nonwatermarked text using `KGW-2.0` detector.

### D.4 Hyperparameter tuning

There are two hyperparameters involved in computing each of our watermark test statistics (i.e., Algorithm 3), the block size $k$ and the alignment score $d$. We do not tune the block size $k$ for our experiments, instead simply letting $k = m$, i.e., the text length, and the alignment score is also fixed for each of our watermarks, except for the hyperparameter $\gamma$ in both `ITS-edit` and `EXP-edit`. Smaller values of $\gamma$ (at least to a certain point) tend to make these watermarks more robust to insertion and deletion errors, as Figure 21 illustrates, but also hurts their statistical power for large values of $n$, i.e., the watermark key length, as Figure 22 illustrates. We set $\gamma = 0.4$ for `ITS-edit` and $\gamma = 0.0$ for `EXP-edit` to balance these two competing desiderata.

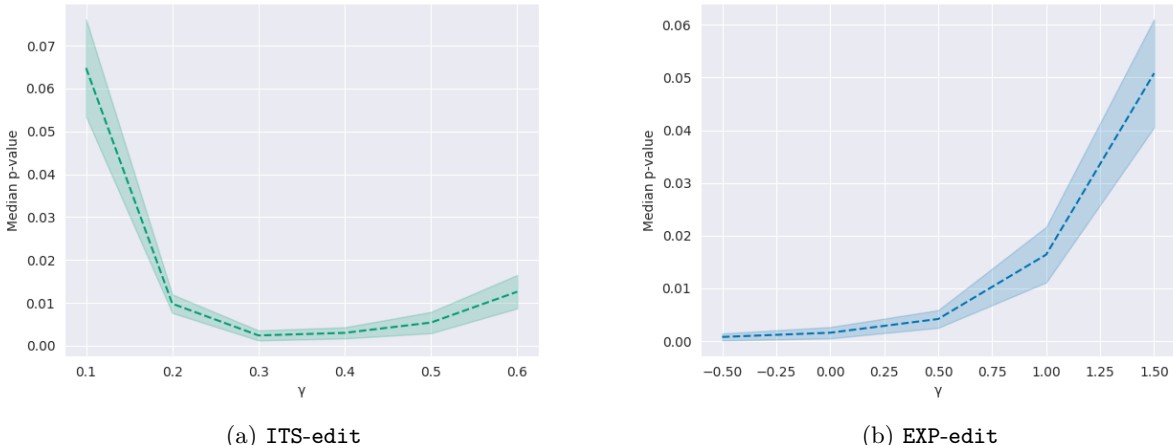

(a) ITS-edit          (b) EXP-edit

Figure 21: Median $p$-value of watermarked text for varying $\gamma$, with OPT-1.3B models and $m = 70$ for ITS-edit and $m = 35$ for EXP-edit, after corrupting the text with random insertions (fraction of inserted tokens is 0.1 for ITS-edit and 0.6 for EXP-edit).

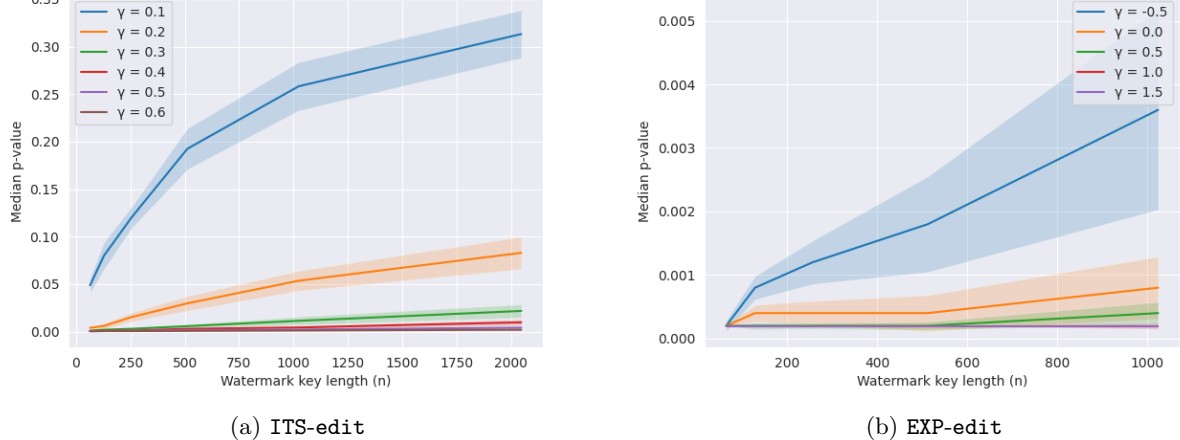

(a) ITS-edit          (b) EXP-edit

Figure 22: Median $p$-value of watermarked text, varying $\gamma$ and $n$, with OPT-1.3B model and $m = 40$ for ITS-edit and $m = 10$ for EXP-edit.

## D.5 Deferred results

### D.5.1 Experiment 3

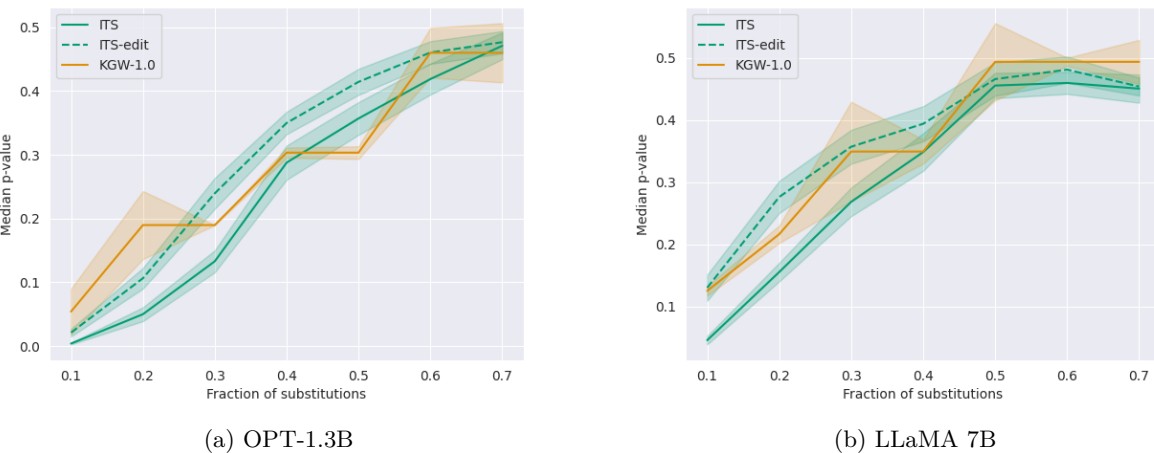

(a) OPT-1.3B

(b) LLaMA 7B

Figure 23: Median $p$-value of watermarked text relative to the fraction of substitution errors, for OPT-1.3B and LLaMA 7B models with $m = 35$.

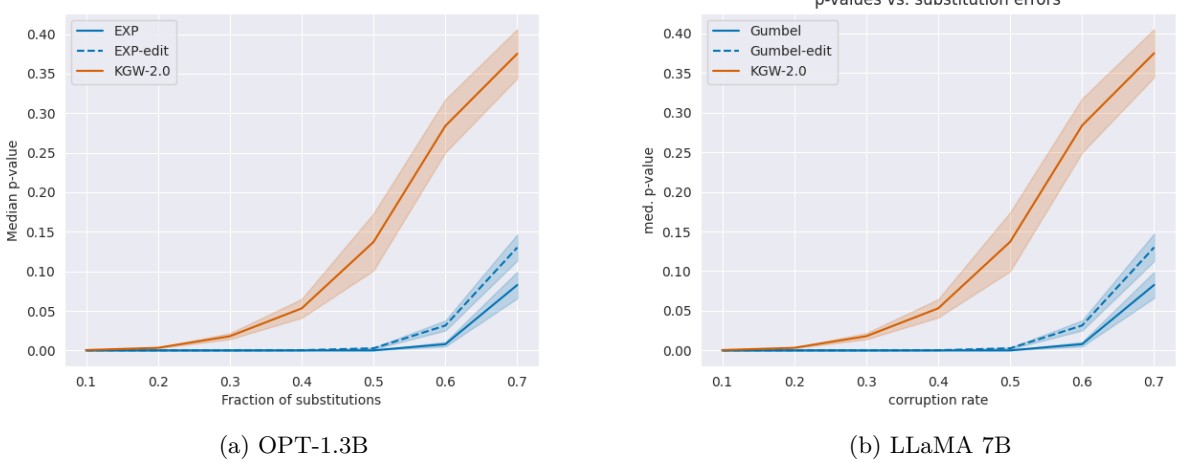

(a) OPT-1.3B

(b) LLaMA 7B

Figure 24: Median $p$-value of watermarked text relative to the fraction of substitution errors, for OPT-1.3B and LLaMA 7B models with $m = 70$.

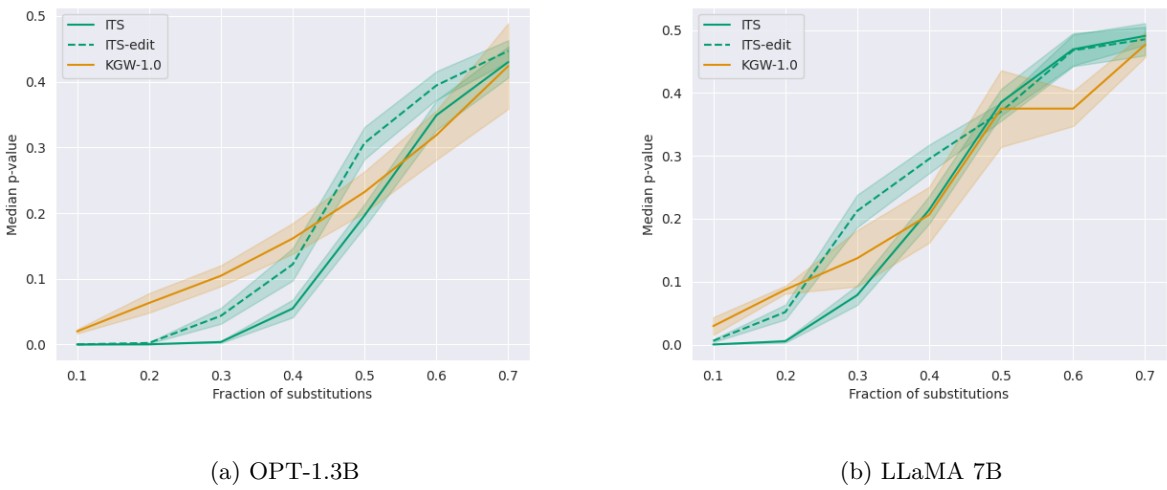

(a) OPT-1.3B                                      (b) LLaMA 7B

Figure 25: Median $p$-value of watermarked text relative to the fraction of substitution errors, for OPT-1.3B and LLaMA 7B models with $m = 70$.

### D.5.2    Experiment 4

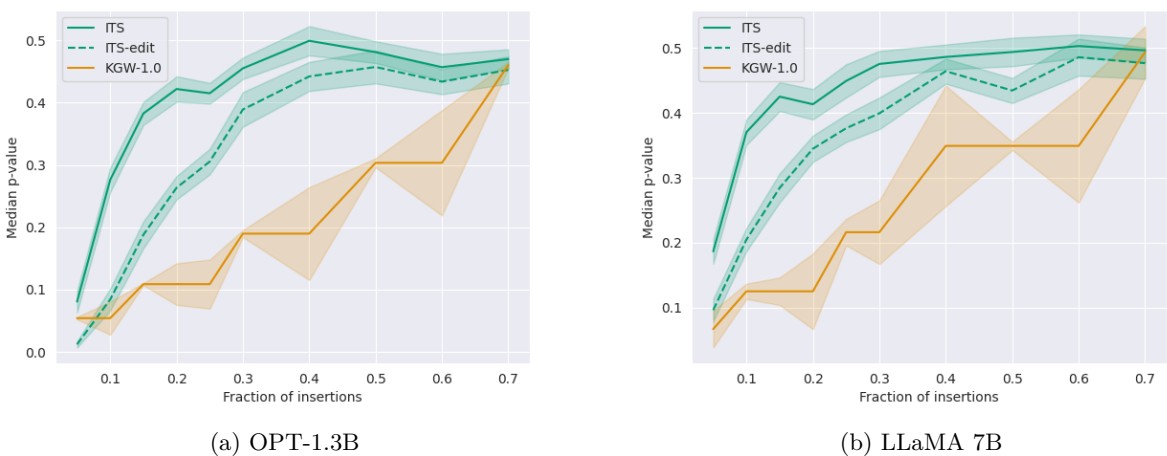

(a) OPT-1.3B                                      (b) LLaMA 7B

Figure 26: Median $p$-value of watermarked text relative to the fraction of insertion errors, for OPT-1.3B and LLaMA 7B models with $m = 35$.

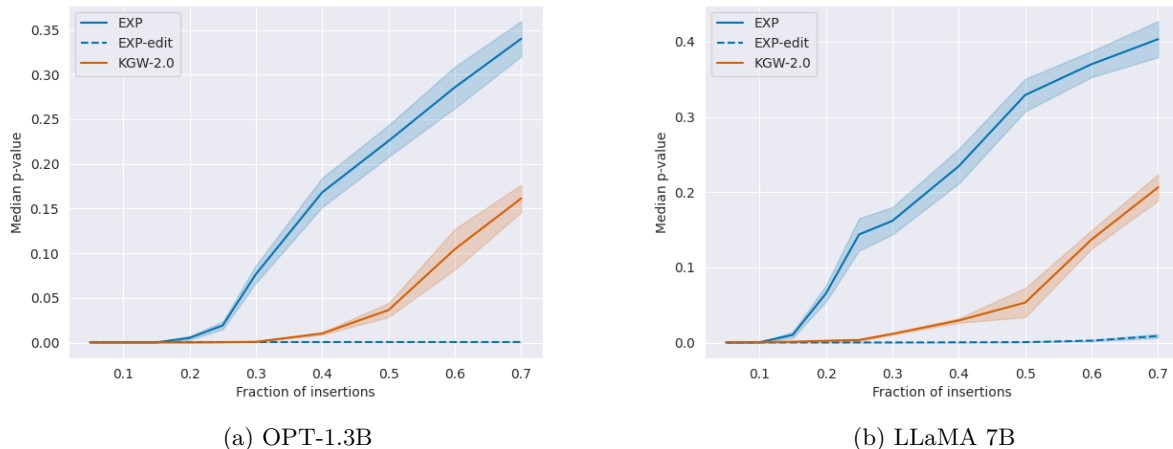

(a) OPT-1.3B          (b) LLaMA 7B

Figure 27: Median $p$-value of watermarked text relative to the fraction of insertion errors, for OPT-1.3B and LLaMA 7B models with $m = 70$.

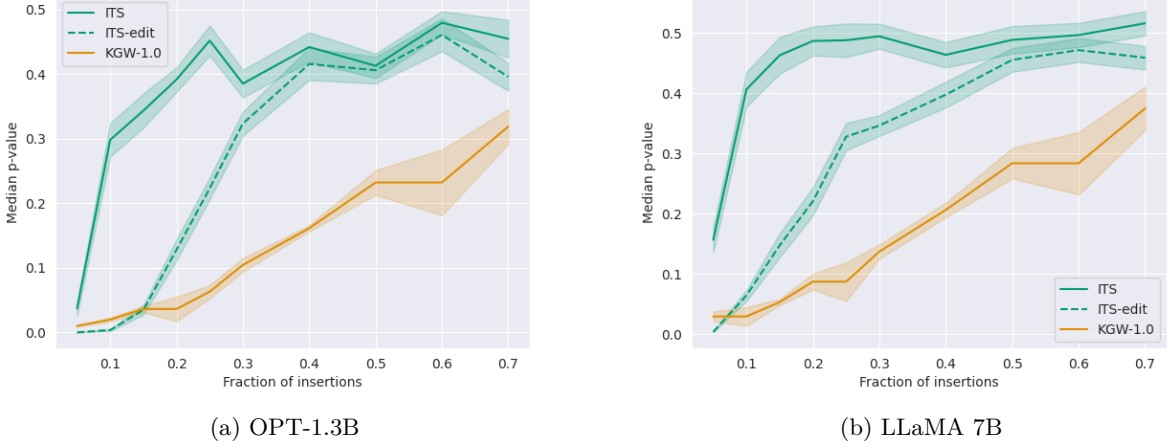

(a) OPT-1.3B          (b) LLaMA 7B

Figure 28: Median $p$-value of watermarked text relative to the fraction of insertion errors, for OPT-1.3B and LLaMA 7B models with $m = 70$.

### D.5.3 Experiment 5

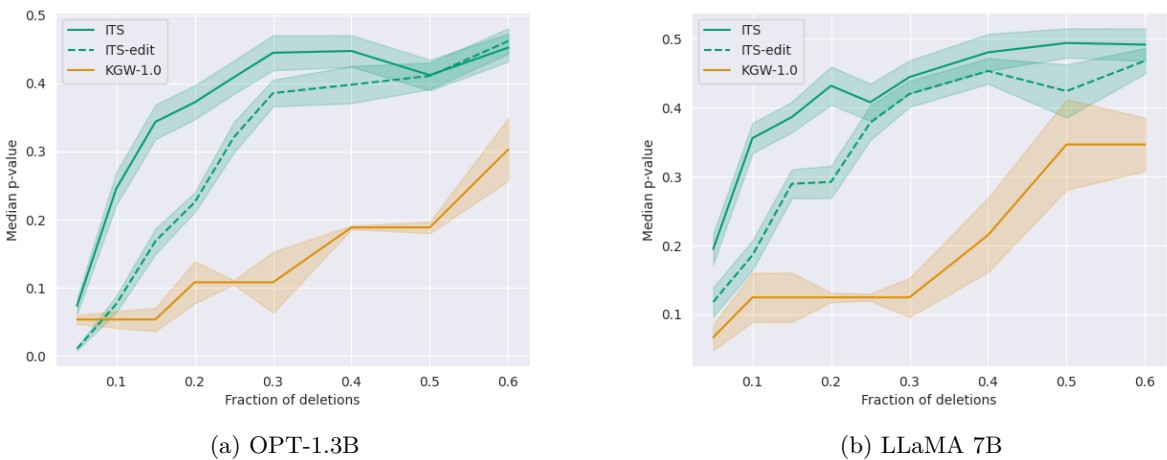

(a) OPT-1.3B               (b) LLaMA 7B

Figure 29: Median $p$-value of watermarked text relative to the fraction of deletion errors, for OPT-1.3B and LLaMA 7B models with $m = 35$.

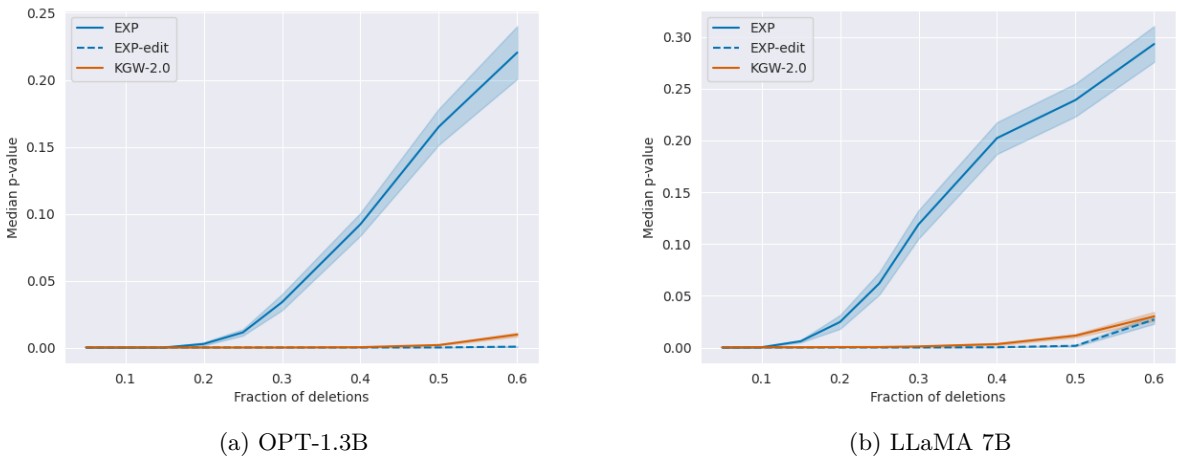

(a) OPT-1.3B               (b) LLaMA 7B

Figure 30: Median $p$-value of watermarked text relative to the fraction of deletion errors, for OPT-1.3B and LLaMA 7B models with $m = 70$.

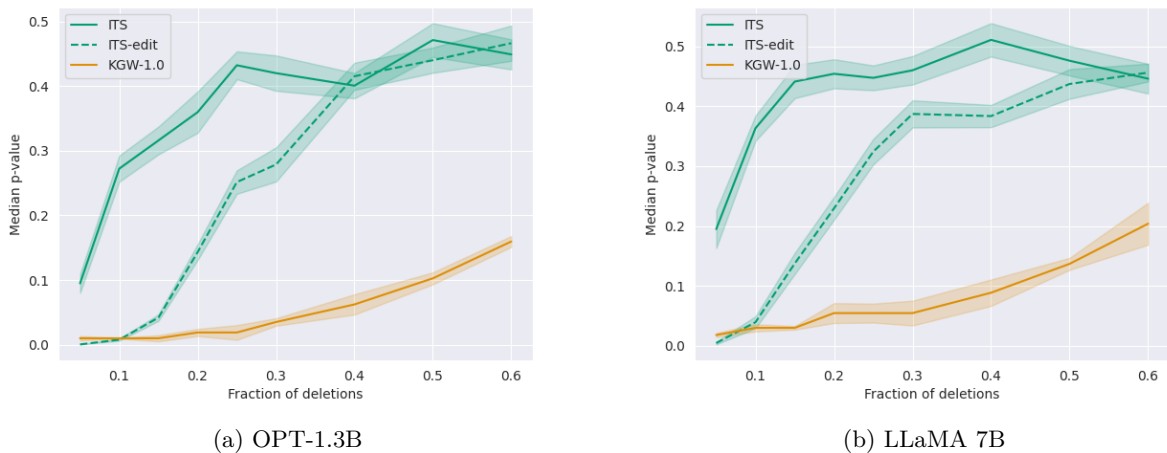

(a) OPT-1.3B  (b) LLaMA 7B

Figure 31: Median $p$-value of watermarked text relative to the fraction of deletion errors, for OPT-1.3B and LLaMA 7B models with $m = 70$.

### D.5.4 Experiment 6

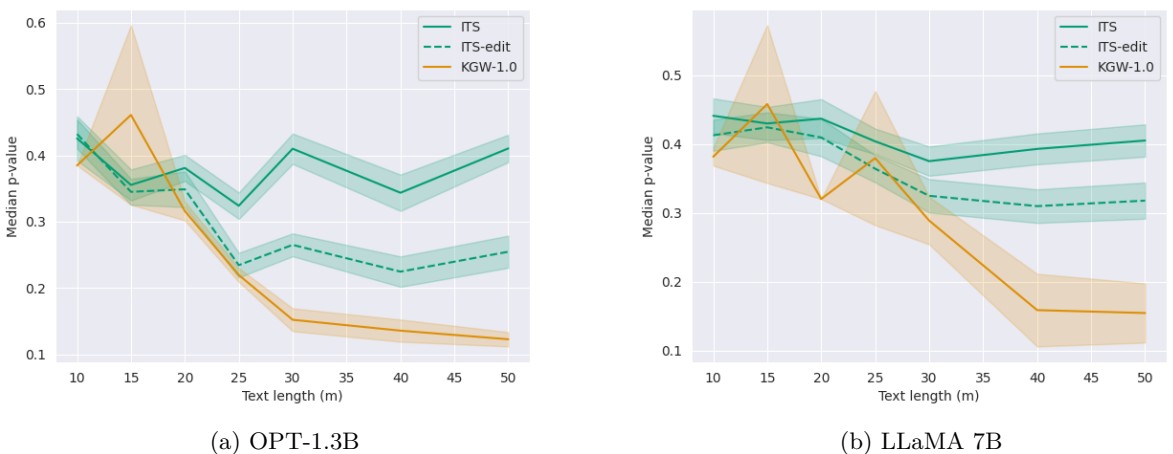

(a) OPT-1.3B  (b) LLaMA 7B

Figure 32: Median $p$-value of watermarked text relative to the fraction of insertion errors, after roundtrip translation via French, for OPT-1.3B and LLaMA 7B models with $m = 35$.

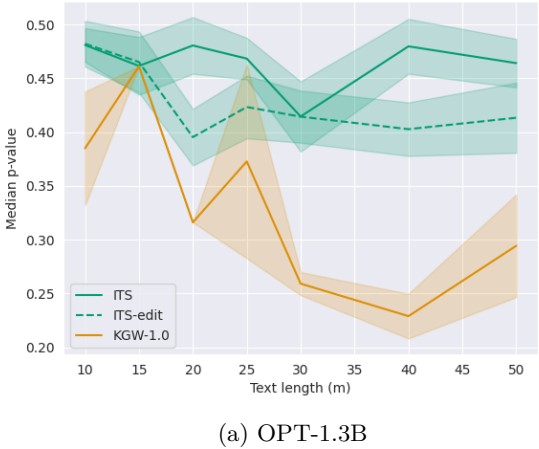

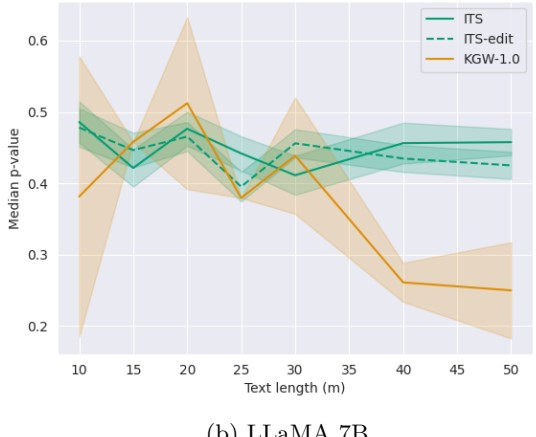

(a) OPT-1.3B

(b) LLaMA 7B

Figure 33: Median $p$-value of watermarked text relative to the text length, after roundtrip translation via Russian, for OPT-1.3B and LLaMA 7B models with $m = 35$.

### D.5.5 Instruction following case study

We give three examples of instructions for which hashing produces qualitatively worse responses than regular samples from the language model:

1. "Give me 20 ideas for the title of a paper on watermarking language models."

2. "Give me 20 ideas for startup names."

3. "Give me a list of 20 movies."

We format each of the instructions as described by Taori et al. (2023) before calling the model.

We compare samples from our EXP watermark strategy,[18] which are equivalent to regular samples from the language model, to samples from KGW-2.0 and the hashing-based version of EXP we describe in the main text (i.e., the watermark of Aaronson (2023)), i.e., EXP-hash. For both EXP and KGW-2.0, we generate the samples using five different random seeds (the hash function in KGW-2.0 is fixed in the implementation of Kirchenbauer et al. (2023)), whereas in the case of EXP-hash we use five different hash functions (namely, we let the previous $k$ tokens $\{y_i\}_{i=1}^k$ hash to $j + \sum_{i=1}^k y_i$ for $j \in \{0, \dots, 4\}$). We label each sample using the seed/hash we used to generate it. We include samples from two versions of EXP-hash: one where we hash the previous tokens ($k = 1$) and another where we hash the previous four tokens ($k = 4$). For KGW-2.0, we only hash the previous token since the public implementation of Kirchenbauer et al. (2023) does not include the option to hash more tokens.

We find that EXP-hash with $k = 1$ often produces qualitatively worse responses that degenerate into repetition. With $k = 4$, the repetition is substantially less noticeable, though occasionally it still manifests. In contrast, even when we only hash the previous token, the repetition of KGW-2.0 is not nearly as noticeable as in EXP-hash. We speculate this is due to stochasticity of KGW-2.0 (i.e., KGW-2.0 biases the distribution over the next token to a subset of tokens but still ultimately samples from this distribution randomly). Of course, this stochasticity comes at a price: KGW-2.0 was generally less powerful compared to the EXP and EXP-edit strategies in our other experiments.

We include sample sheets for all methods for the first instruction below. To avoid excessive clutter, we defer the sample sheets for the remaining two instructions to our code release.

---

[18]Recall both EXP and EXP-edit use the same `generate` method.

| Method | Seed | Response |
|---|---|---|
| EXP-hash $(k=1)$ | 0 | 1. Watermarking Deep Learning Models: A Review 2. Understanding and Evaluating Watermarking Techniques for Neural Networks 3. Towards Unobtrusive and Invisible Watermarking for Neural Networks 4. Customizable Watermarking for Neural Networks 5. Reverse Engineering Watermarked Neural Networks 6. Understanding and Exploiting Watermarking Attacks on Neural Networks 7. Combatting Watermarking Attacks on Neural Networks 8. Customizable Watermarking for Neural Networks 9. Customizable Watermarking for Neural Networks 10. Customizable Watermarking for Neural Networks 11. Customizable Watermarking for Neural Networks 12. Customizable Watermarking for Neural Networks 13. Customizable Watermarking for Neural Networks 14. Customizable Watermarking for Neural Networks 15. Customizable Watermarking for Neural Networks 16. Customizable Watermarking for Neural Networks 17. Customizable Watermarking for Neural Networks 18. Customizable Watermarking for Neural Networks 19. Customizable Watermarking for Neural Networks |
| EXP-hash $(k=1)$ | 1 | 1. Exploiting Watermarks in Language Modeling 2. Watermarking Language Models: State of the Art 3. Keys to the Kingdom: Refining Watermarking Language Models 4. Exploiting Watermarks in Natural Language Processing 5. Keys to the Kingdom: Refining Watermarking Language Models 6. Keys to the Kingdom: Refining Watermarking Language Models 7. Keys to the Kingdom: Refining Watermarking Language Models 8. Keys to the Kingdom: Refining Watermarking Language Models 9. Keys to the Kingdom: Refining Watermarking Language Models 10. Keys to the Kingdom: Refining Watermarking Language Models 11. Keys to the Kingdom: Refining Watermarking Language Models 12. Keys to the Kingdom: Refining Watermarking Language Models 13. Keys to the Kingdom: Refining Watermarking Language Models 14. Keys to the Kingdom: Refining Watermarking Language Models 15. Keys to the Kingdom: Refining Watermarking Language Models 16. Keys to the Kingdom: Refining Watermarking Language Models 17. Keys to the Kingdom: Refining Watermarking Language Models 18. Keys to the Kingdom: Refining Watermarking Language Models 19. Keys to the Kingdom: Refining Watermarking Language Models 20. Keys to the Kingdom: Refining Watermarking Language Models |
| EXP-hash $(k=1)$ | 2 | 1. "Making Watermarks: Creative Approaches to Watermarking Language Models" 2. "Making Watermarks: Creative Approaches to Watermarking Language Models" 3. "Making Watermarks: Creative Approaches to Watermarking Language Models" 4. "Making Watermarks: Creative Approaches to Watermarking Language Models" 5. "Making Watermarks: Creative Approaches to Watermarking Language Models" 6. "Making Watermarks: Creative Approaches to Watermarking Language Models" 7. "Making Watermarks: Creative Approaches to Watermarking Language Models" 8. "Making Watermarks: Creative Approaches to Watermarking Language Models" 9. "Making Watermarks: Creative Approaches to Watermarking Language Models" 10. "Making Watermarks: Creative Approaches to Watermarking Language Models" 11. "Making Watermarks: Creative Approaches to Watermarking Language Models" 12. "Making Watermarks: Creative Approaches to Watermarking Language Models" 13. "Making Watermarks: Creative Approaches to Watermarking Language Models" 14. "Making Watermarks: Creative Approaches to Watermarking Language Models" 15. "Making Watermarks: Creative Approaches to Watermarking Language Models" 16. "Making Watermarks: Creative Approaches to Watermarking Language Models" 17. "Making Watermarks: Creative Approaches to Watermarking Language Models" 18. "Making Watermarks: Creative Approaches to Watermarking Language Models" 19. "Making Watermarks: Creative Approaches to Watermarking Language Models" 20. "Making Watermarks: Creative Approaches to Watermarking Language Models" |

| EXP-hash ($k=1$) | 3 | 1. Watermarking Language Models: A Review 2. Watermarking Language Models for Copyright Protection 3. Watermarking Language Models for Forensic Analysis 4. Watermarking Language Models for Audio Signals 5. Watermarking Language Models for Video Signals 6. Watermarking Language Models for Documents 7. Watermarking Language Models for Software 8. Watermarking Language Models for Big Data 9. Watermarking Language Models for Natural Language Processing 10. Watermarking Language Models for Machine Learning 11. Watermarking Language Models for Deep Learning 12. Watermarking Language Models for Speech Recognition 13. Watermarking Language Models for Audio Synthesis 14. Watermarking Language Models for Computer Vision 15. Watermarking Language Models for Robotics 16. Watermarking Language Models for Autonomous Driving 17. Watermarking Language Models for Medical Image Processing 18. Watermarking Language Models for Medical Diagnostics 19. Watermarking Language Models for Text Translation |
|---|---|---|
| EXP-hash ($k=1$) | 4 | 1. Image based watermarking for language models 2. Steganographic watermarking for language models 3. Timestamps for language models 4. Watermarking for language model improvisation 5. Secure watermarking for language model sharing 6. Image based watermarking for language model improvisation 7. Timestamps for language model improvisation 8. Watermarking for language model evolution 9. Secure watermarking for language model evolution 10. Image based watermarking for language model evolution 11. Timestamps for language model evolution 12. Watermarking for language model generation 13. Image based watermarking for language model generation 14. Timestamps for language model generation 15. Watermarking for language model understanding 16. Image based watermarking for language model understanding 17. Timestamps for language model understanding 18. Watermarking for language model improvisation and understanding 19. Image based watermarking for language model improvisation and understanding |

Table 1: Responses from EXP-hash with $k=1$ to the prompt: "Give me 20 ideas for the title of a paper on watermarking language models."

| Method | Seed | Response |
|---|---|---|
| EXP-hash ($k=4$) | 0 | 1. "Safeguarding Language Models: Exploring Watermarking for Vertext and Image Models" 2. "Watermarking Language Models: Protecting Your Creations" 3. "The Future of Watermarking Language Models: Ethical and Practical Considerations" 4. "Securing Language Models: Exploring Emerging Technologies" 5. "Watermarking Language Models: Balancing Creativity and Security" 6. "Watermarking Language Models: Balancing Security and Accessibility" 7. "Watermarking Language Models: Defending your Creativity" 8. "Watermarking Language Models: Defending your Creativity with Emerging Technologies" 9. "Protecting Language Models: Exploring Emerging Technologies" 10. "Securing Language Models: Balancing Security and Accessibility" 11. "The Future of Watermarking Language Models: Ethical and Practical Considerations" 12. "Watermarking Language Models: Balancing Security and Accessibility" 13. "Watermarking Language Models: Balancing Security and Accessibility with Emerging Technologies" 14. "Watermarking Language Models: Defending your Creativity with Emerging Technologies" 15. "Watermarking Language Models: Defending your Creativity with Emerging Technologies" 16. "Securing Language Models: Balancing Security and Accessibility with Emerging Technologies" 17. "Exploring Watermarking for Vertext and Image Models" 18. "Watermarking Language Models: Balancing Security and Accessibility with Emerging Technologies" 19. "Defending your Creativity with Emerging Technologies" |

| | | |
|---|---|---|
| EXP-hash ($k = 4$) | 1 | 1. Towards a New Era of Transparent Language Models 2. A Review of the State of Watermarking Language Models 3. The Benefits of Embedding Watermarks in Language Models 4. Protecting Language Models with Multiscale Watermarks 5. Impact of Watermarking on the Performance of Language Models 6. A Survey on Watermarking for Language Models 7. Practical Perspectives on Watermarking for Language Models 8. A Comprehensive Study on Designing Watermarks for Language Models 9. Overview of Techniques for Adding Watermarks to Language Models 10. Exploring the Possibilities of Watermarking for Language Models 11. How to Incorporate Watermarks in Your Language Model 12. The Science behind Watermarking for Language Models 13. AI for Insertion of Watermarks in Language Models 14. The Role of Machine Learning in Watermarking for Language Models 15. Future Trends in Watermarking for Language Models 16. A Review on Watermarking for Language Models 17. Applications of Watermarking in Language Modeling 18. A Comprehensive Study on Designing Robust Watermarks for Language Models 19. A Novel Approach to Incorporate Watermarks in Your Language Model. |
| EXP-hash ($k = 4$) | 2 | 1. Securing Your Language Model 2. Stamping Out Unauthorized Use 3. Coloring Outside the Lines: Creative Watermarks 4. Avoiding Watermarks: Best Practices 5. Authentication Made Easy with Watermarks 6. Defending Your Language Model 7. Unique Identifiers: Adding Value to your Model 8. Connected Learning: Leveraging Watermarks 9. The Problem with Open Access 10. How to Effectively Mark a Language Model 11. Making a Splash with Creative Watermarks 12. Understanding the Benefits of Watermarking 13. Utilizing Watermarks for Better Attribution 14. Stewarding Your Language Model 15. The Role of Technology in Watermarking 16. Beyond the Horizon: Adaptive Watermarking 17. The Art of Discretion in Watermarking 18. Harnessing the Power of Invisible Watermarks 19. Practical Considerations for Watermarking |
| EXP-hash ($k = 4$) | 3 | 1. Elucidating Watermarking Strategies for Language Models 2. Innovative Techniques for Watermarking Language Models 3. Intelligent and Innovative Watermarking Strategies for Language Models 4. Overview of Watermarking Strategies for Language Models 5. Emerging Technologies in Watermarking for Language Models 6. Practical Perspectives on Watermarking Language Models 7. Reducing Watermarking Strategies for Language Models 8. Robust Techniques for Watermarking Language Models 9. Security Factors in Watermarking Language Models 10. Understanding Watermarking Strategies for Language Models 11. Evaluation of Watermarking Strategies for Language Models 12. Exploring Watermarking Strategies for Language Models 13. Innovations in Watermarking Language Models 14. Intelligent Techniques for Watermarking Language Models 15. Methods for Watermarking Language Models 16. Overlapping Security Factors in Watermarking Language Models 17. Understanding Watermarking Strategies for Language Models 18. Emerging Technologies in Watermarking Language Models 19. Exploring Watermarking Strategies for Language Models 20. Intelligent and Innovative Watermarking Strategies for Language Models |
| EXP-hash ($k = 4$) | 4 | 1. Securing Language Models 2. Extra Secure Language Models 3. Defending Language Models 4. Protecting Language Models 5. Cryptographic Language Models 6. Unique Language Models 7. Robust Language Models 8. Marking Language Models 9. Preventing Language Models 10. Hiding Language Models 11. Invisible Language Models 12. Steganographic Language Models 13. Watermarking Language Models 14. Preserving Language Models 15. Uniquely Identifiable Language Models 16. Characterizing Language Models 17. Distinguishable Language Models 18. Language Model Defence 19. Ultra Defence Language Models |

Table 2: Responses from EXP-hash with $k = 4$ to the prompt: "Give me 20 ideas for the title of a paper on watermarking language models."

| Method | Seed | Response |
|---|---|---|

| | | |
|---|---|---|
| KGW-2.0 | 0 | 1. Steganography in Language Models: A Review 2. Combating Model Piracy with Watermarking 3. Techniques for Traceability and Watermarking 4. Salient Features for Traceability and Watermarking 5. Reflections on Model Security and Watermarking 6. Insights on Model Security and Watermarking 7. Protecting Models with Watermarking 8. Techniques for Esoteric Traceability and Watermarking 9. Ins and Outs of Steganography in Language Models 10. Technologies for Efficient Watermarking 11. Vision Beyond Model Piracy: Watermarking Perspectives 12. Impact of Model Security on Watermarking 13. Emerging Trends in Watermarking of Language Models 14. Future of Watermarking Techniques for Language Models 15. Drivers for Success in Watermarking Language Models 16. Robustness of Models against Watermarking 17. Taking Security beyond Watermarking 18. Leveraging Model History and Watermarking 19. Techniques for Secure Watermarking 20. Comprehensive Overview of Model Security and Watermarking. |
| KGW-2.0 | 1 | 1. Embedding Digital Signatures: Towards Traceable and Transparent Language Models 2. Stamping Out Deception: Recording Attribution in Language Models 3. Defining Your Digital Dashboard: Watermarking for Traceability 4. Scaling Trust in Language Models: Watermarking for Traceability and Transparency 5. Taking an East Asian Approach to Transparency in Language Models 6. Riding the Big Five: Towards Transparency in Language Models 7. Measuring the "Ps" of Language Models: Perceptual and Practical Transparency 8. Marking a Mark: Comprehensive Examination of Attribution in Language Models 9. Visualizing Deception: Comprehensive Examination of Deceptive Language Models 10. Peak Performance: Evaluating Language Models for Transparency 11. Closing the Loop: Examining the Long-term Impacts of Language Models 12. Diversifying Perspective: Impacts of Modelling Language 13. Disclosing the Hidden Layers: Understanding Transparency in Language Models 14. Auditable: Making Language Models Accountable 15. Labelling Human Language: Exploring the Capabilities of Language Models 16. Setting the Record: Estimating the Real-world Impacts of Language Models 17. Language Models: Towards Traceable, Transparent and Accountable Systems 18. Think Outside the Box: Exploring Future Trends in Language Models 19. Embedding Trust: Towards Traceable and Transparent Language Models. |
| KGW-2.0 | 2 | 1. Defending Latent Space with Secure Watermarks 2. Emergent Waveforms: A Watermarking Approach 3. Embedding Digital Identities in Neural Networks 4. Marking up Models: A Survey on Watermarking 5. Leveraging Linked Data for Watermarking 6. Stamping Out Superfluous Models 7. Inspecting and Marking Models 8. Rethinking Digital Identity with Watermarks 9. Marking Up Models: A Usage-Driven Approach 10. Bridge the Distance with Digital Watermarks 11. Introducing Transparency in Model Formation 12. Unlocking Models with Secure Watermarks 13. Giving Visible Identity to Models 14. Unveiling Dark Knowledge with Watermarks 15. Linking Models: A Visual Approach 16. Visualizing Dark Knowledge through Watermarks 17. Leveraging Physical World for Model Tracking 18. Augmented Reality for Model Tracking 19. Securing Deep Models with Watermarks |
| KGW-2.0 | 3 | 1. Providing Type-Safe Watermarks 2. Constructing Robust Watermarks for Multimedia Applications 3. Adding Security Features to Text Vectors 4. Stretching Visible Watermarks 5. Embedding Watermarks in Natural Language Models 6. Training Recurrent Neural Networks with Watermarks 7. Detection of Unsupervised Watermarks in Text Sequences 8. Interactive Visualization of Watermarks 9. Observe Hidden Watermarks with Machine Learning 10. Extracting Information from Unseen Watermarks 11. Building Robust Watermarks for the Blockchain 12. Adding Dynamic Watermarks to Video Streams 13. Storing Watermarks in Individual Dataset Elements 14. Extracting Pertinent Information from Unwanted Watermarks 15. Decoding Hidden Digital Watermarks 16. Introducing Watermarks to Secure Semi-honest Adversaries 17. Genuine Components with Integrated Watermarks 18. Intelligent Techniques for Steganography 19. Protecting Quality with Data-Driven Watermarks |

| | | |
|---|---|---|
| KGW-2.0 | 4 | 1. Embedding Watermarks in Language Models 2. Secure Language Models using Watermarks 3. Steganographic watermarking of natural language models 4. Exploring Watermarking Techniques for Natural Language Models 5. Analyzing Natural Language Models using Watermarking Techniques 6. Detecting Watermarks in Natural Language Models 7. Supervised Learning of Watermarks for Natural Language Models 8. Unsupervised Learning of Watermarks for Natural Language Models 9. Multiple watermarks for secure language models 10. Generative watermarking techniques for natural language models 11. Evaluation of watermarks for secure natural language models 12. Efficacy of watermarks in preventing model poisoning 13. Robustness of watermarks for natural language models 14. Comparison of watermarking techniques for natural language models 15. Linking of natural language models with watermarks 16. Extraction of hidden information from natural language models 17. Universality of watermarks for different languages 18. Learning watermarked natural language models 19. Reduced Dimensionality Modeling using Watermarks |

Table 3: Responses from KGW-2.0 to the prompt: "Give me 20 ideas for the title of a paper on watermarking language models."

| Method | Seed | Response |
|---|---|---|
| EXP | 0 | Here is the list of 20 ideas for the title of a paper on watermarking language models: 1. "Understanding Watermarking Techniques for Language Models" 2. "A Comprehensive Study on Watermarking for Language Models" 3. "Watermarking Techniques for Optimizing Language Models" 4. "A Survey of Watermarking Techniques for Language Models" 5. "Exploring the Potential of Watermarking for Language Models" 6. "Implementing Watermarking for Language Model Optimization" 7. "Watermarking Strategies for Enhancing Language Models" 8. "Investigating the Possibilities of Watermarking for Language Models" 9. "Advanced Watermarking Approaches for Language Models" 10. "Exploring the Use of Watermarking for Language Model Validation" 11. "Understanding the Benefits of Watermarking for Language Model Ensemble" 12. "Leveraging Watermarking Techniques for Enhanced Language Model Performance" 13. "Enhancing Language Models with Watermarking" 14. "Evaluating the Impact of Watermarking Techniques on Language Models" 15. "Analyzing the Feasibility of Watermarking for Language Model Comparison" 16. "Exploring the Possibilities of Watermarking for Optimizing Language Models" 17. "Exploiting Watermarking to Enhance Language Model Accuracy" 18. "Advantages of Using Watermarking for Validating Language Models" 19. "The Promise of Watermarking for Evaluating Language Model Performance" |
| EXP | 1 | Title ideas for a paper on watermarking language models: 1. Hidden Messages: Protecting Language Models with Watermarking 2. Securing Language Models with Watermarking 3. Defending Language Models against Tampering 4. Watermarking Language Models for Better Protection 5. Verifying the Integrity of Language Models 6. Utilizing Watermarking to Boost Language Model Security 7. Ensuring the Authenticity of Language Models 8. Safekeeping Language Models with Watermarking 9. Defending Against Model Tampering with Watermarking 10. A Review of Model Security Techniques: Watermarking 11. Utilizing Watermarks to Protect Language Models 12. Protective Techniques for Language Models: Watermarking 13. Implementing Watermarks to Boost Language Model Security 14. Preventing Model Tampering with Watermarking 15. Methods for Verifying Language Model Integrity 16. Analyzing the Security of Language Models 17. Techniques for Securing Language Models 18. Investigating Model Security with Watermarking 19. Extending Language Model Security with Watermarking |

| | | |
|---|---|---|
| EXP | 2 | 1. "Deep Watermarks: Towards Pervasive Protection for Language Models" 2. "Adding Transparency to Language Models: A Watermarking Approach" 3. "Watermarking Language Models for Traceability and Verification" 4. "A Comprehensive Review on Watermarking Methods for Language Models" 5. "The Power of Watermarking for Language Model Protection" 6. "Attributing Authorship in Language Models with Watermarking" 7. "Understanding the Promise of Watermarking for Language Model Security" 8. "Watermarking Language Models: A Survey" 9. "Watermarking Language Models for Better Security" 10. "Mitigating Plagiarism in Language Models with Watermarking" 11. "Watermarking Language Models: Exploring the Possibilities" 12. "A Generative Approach to Watermarking Language Models" 13. "Watermarking Strategies for Protecting Language Models" 14. "Watermark Detection for Language Model Security" 15. "Making Language Model Security Watertight with Watermarking" 16. "Leveraging Watermarking for Enhancing Language Model Security" 17. "Understanding the Role of Watermarking in Language Model Security" 18. "A Novel Approach to Watermarking Language Models" 19. "Exploring the Possibilities of Watermarking for Language Model Security" |
| EXP | 3 | 1. A Comprehensive Review of Watermarking Techniques for Language Models 2. A Survey of Watermarking Approaches for Language Modeling 3. A Normative Analysis of Watermarking for Language Modeling 4. Investigating Watermarking Techniques for Language Modeling 5. An Overview of Watermarking Methods for Language Modeling 6. Exploring Watermarking Solutions for Language Modeling 7. A Taxonomy of Watermarking Methods for Language Modeling 8. A Comparative Study of Watermarking Approaches for Language Modeling 9. Evaluating Watermarking Methods for Language Modeling 10. A Theoretical Analysis of Watermarking for Language Modeling 11. Investigating Watermarking Techniques for Natural Language Modeling 12. An Analytical Study of Watermarking for Language Modeling 13. Exploring Watermarking Methods for Natural Language Modeling 14. A Review of Watermarking Techniques for Natural Language Modeling 15. A Comparative Study of Watermarking Approaches for Natural Language Modeling 16. Investigating Watermarking Solutions for Natural Language Modeling 17. A Survey of Watermarking Techniques for Natural Language Modeling 18. Evaluating Watermarking Methods for Natural Language Modeling 19. A Review of Watermarking for Natural Language Modeling 20. Exploring Watermarking Solutions for Natural Language Modeling |
| EXP | 4 | 1. Towards a Unified Watermarking Mechanism for Natural Language Processing Models 2. A Review of Methods for Watermarking Natural Language Models 3. Extracting Invariant Features for Watermarking Language Models 4. The Use of Steganography for Watermarking Natural Language Models 5. Introducing Secure Watermarking Techniques for Natural Language Models 6. A Comprehensive Study on Watermarking Techniques for Natural Language Models 7. Toward Remarkably Visible Watermarks for Natural Language Models 8. Analyzing the Impact of Watermarking on Natural Language Models 9. A Practical Guide to Marking Language Models 10. Enhancing the Accuracy of Watermarking Natural Language Models 11. Evaluating Strategies for Watermarking Natural Language Models 12. A Comparison of Watermarking Approaches for Natural Language Models 13. Promising Solutions for Securely Watermarking Natural Language Models 14. Generative and Discriminative Approaches for Watermarking Natural Language Models 15. Exploring the Possibilities of Steganography for Natural Language Models 16. Understanding the Challenges of Watermarking Natural Language Models 17. Evaluating the Effectiveness of Watermarking Techniques for Natural Language Models 18. Enhancing the Transparency of Watermarking Techniques for Natural Language Models 19. Extending the Capabilities of Watermarking Techniques for Natural Language Models 20. Assessing the Sophistication of Watermarking Techniques for Natural Language Models |

Table 4: Responses from EXP to the prompt: "Give me 20 ideas for the title of a paper on watermarking language models."

