# OpenReview forum: "Robust Distortion-free Watermarks for Language Models"
_TMLR — Accepted by TMLR_

### Review · Reviewer_eJM9 · 2023-12-21

**Summary Of Contributions:**

The paper presents a methodology for embedding robust, distortion-free watermarks in text generated by autoregressive language models. This is achieved without altering the text distribution. The process involves generating watermarked text by mapping random numbers, computed using a watermark key, to samples from the language model. Detection is possible for anyone with the key. Two watermarking methods are explored: inverse transform sampling and exponential minimum sampling. The techniques are tested on various language models (OPT-1.3B, LLaMA-7B, Alpaca-7B) to validate their statistical power and robustness against paraphrasing attacks and text corruption. The study finds that the watermarks are detectable even with substantial token corruption and explores a case study on watermarking responses to user instructions.

**Audience:**

Yes

**Claims And Evidence:**

Yes

**Requested Changes:**

1. **Interplay with Language Model Updates**: How does the watermarking technique adapt to updates or modifications in the underlying language model, e.g., moving from LLaMA to LLaMA 2,3 etc.?


2. **Quantitative Analysis of Text Distortion**: Can you provide a quantitative metric to assess the degree of distortion (if any) introduced by the watermarking process in the text? This could help in objectively evaluating the "distortion-free" claim.


3. **Watermark Capacity and Limitations**: What is the maximum amount of information that can be embedded as a watermark without affecting the text quality?


4. **Compatibility with Different Text Genres**: How does the watermarking process perform across various text genres, such as technical documents, creative writing, or conversational text?


5. **Watermark Traceability in Distributed Systems**: How traceable are these watermarks in distributed systems, like when text is shared across multiple platforms?


6. **Algorithm Specificity in Inverse Transform Sampling**: Could you elaborate on how inverse transform sampling is adapted for watermarking in language models? Specifically, how does it maintain the fidelity of the original text distribution while embedding watermarks?

**Strengths And Weaknesses:**

**Strengths:**

1. **Innovative Approach**: The paper introduces novel techniques for embedding watermarks in machine-generated text. It's a pioneering step in content attribution, crucial for intellectual property protection in AI-generated content. The approach is unique in its aim to seamlessly integrate watermarks without altering the natural text distribution.

2. **Robustness Against Text Corruption**: The watermarks demonstrate remarkable resilience to various perturbations, such as text alterations or corruptions. This robustness is crucial for the practical applicability of watermarks in real-world scenarios, where text modifications are common.

3. **Preservation of Text Distribution**: A significant achievement of this research is maintaining the originality and coherence of the text. The watermarks are designed to be distortion-free, meaning they do not affect the readability, style, or informational content of the text. This aspect is critical for ensuring that the watermarked text remains useful and engaging for its intended audience.

**Weaknesses:**

1. **Potential for Misuse**: While the paper focuses on the technical aspects of watermarking, it opens up discussions on ethical implications. The technology, if misused, could lead to surveillance or privacy concerns. It's important to consider these aspects in the development and implementation of such technologies.

2. **Model Dependency**: The effectiveness of the watermarking techniques seems to vary with different language models. This suggests a potential limitation in their applicability, indicating that each model may require a customized approach for effective watermarking.

---

> ### Author Response · Authors · 2024-02-01
> **Response to Reviewer eJM9**
>
> > “Potential for Misuse: While the paper focuses on the technical aspects of watermarking, it opens up discussions on ethical implications. The technology, if misused, could lead to surveillance or privacy concerns. It's important to consider these aspects in the development and implementation of such technologies.”
>
> We agree that there is potential for watermarking to be misused for surveillance. Relying on watermarking as a means of individual attribution to certain users (as opposed to generic bot vs. human detection) is fraught for a number of reasons, including the fact that watermarks may be spoofable. We are adding a discussion of broader impacts to the conclusion of the revised paper, where among other things we will discuss the potential for misuse as well as the potential implications of government-mandated watermarking (in light of the recent White House executive order on AI, which explicitly mentioned watermarking).
>
> > “Model Dependency: The effectiveness of the watermarking techniques seems to vary with different language models. This suggests a potential limitation in their applicability, indicating that each model may require a customized approach for effective watermarking.”
>
> We conduct experiments with three different models (OPT-1.3B, LLaMA-7B, Alpaca-7B) and find that the entropy of each model’s conditional distribution over text given the prompt is the main factor which influences watermark detectability (see Figure 12). We expect this to generally remain true across different downstream applications and text genres.
>
> > “Watermark Capacity and Limitations: What is the maximum amount of information that can be embedded as a watermark without affecting the text quality?”
>
> We provide a lower bound in Lemma 2.2 which characterizes the limitations of not just our watermark but any distortion-free watermark: the detectability of any such watermark necessarily suffers on low entropy text.
>
> > “Algorithm Specificity in Inverse Transform Sampling: Could you elaborate on how inverse transform sampling is adapted for watermarking in language models? Specifically, how does it maintain the fidelity of the original text distribution while embedding watermarks?”
>
> We prove in Theorem 1 that our inverse transform sampling watermark is distortion-free (i.e., maintains perfect fidelity). We would be happy to add some further exposition given feedback on what would be most helpful.
>
> > “Watermark Traceability in Distributed Systems: How traceable are these watermarks in distributed systems, like when text is shared across multiple platforms?”
>
> Publishing watermarked text on multiple platforms should not affect the detectability of the text. That said, it’s possible we have misunderstood the question, and we would be happy to discuss it further.

---

> > ### Comment · Reviewer_eJM9 · 2024-02-08
> > **Thank You.**
> >
> > I would like to thank the authors for their thoughtful response. I don't have any further questions.

---

### Review · Reviewer_3pbi · 2023-12-25

**Summary Of Contributions:**

To detect if text is generated the watermark for text was proposed in the past with different algorithms. The current paper considers extension of research in this direction. Authors propose watermarks algorithm for text generated by an autoregressive language
model with assumption that that the owner of LM will share the key with which the text was generated. The algorithm consists of the generate algorithm on the LM owner side and detect algorithm on the user side. The latter is the proper statistics computation and testing it if the text was generated using the key or not. The main difference with prior works is two folds: 1) it is guaranteed the generated text will be from the same distribution as without watermarking (LM conditional distribution is preserved) and 2) proposed algorithms are shown to be robust to perturbations. Authors generate watermarked text by mapping a sequence of random numbers - which they compute using a randomized watermark key - to a sample from the language model. To detect watermarked text, any party who knows the key can
align the text to the random number sequence. Besides deep theoretical analysis and proofs on the asymptotic of the algorithms with respect to the key length and text length, authors perform empirical analysis with modified version of algorithms for practical applicability and show results for OPT-1.3B, LLaMa-7B and Alpaca-7B models with additional substitutions/insertions/deletions and paraphrasing (via back-translation) attacks.

**Audience:**

Yes

**Broader Impact Concerns:**

The paper is trying to resolve some ethics concerns related to generative models. Particularly, it is proposing the method of text watermarking with the hypothesis testing if the text is generated by a particular provider (who gave us the key) independent from the model type and prompt.

**Claims And Evidence:**

Yes

**Requested Changes:**

The paper is very well written though I have some suggestions and notes how to improve readability and clarity further:
- "preserve original text distribution" -- I would be more precise here even in the intro part of the paper, as in the end we preserve only conditional distribution of language model.
- improve notation:
  - $\Nu^{*}$ and in general notation of "*" is not introduced.
  - definition 1 - maybe include here conditional pdf of LM as an example? otherwise seems the definition is about general pdf of text
  - lemma 2.1 why $n$ should be bigger than $m$?  maybe introduce earlier how you generate the text given watermark sequence? that every token in watermark is used to generate next token of text? What happens if $n < m$ (I got that empirically it seems still work to detect if text is watermarked or not)?
  - algorithm 2 line 3: does $\xi^{(t)}\in \Xi^*$?
  - page 9 what does notation $xi_j$ and $xi_i$ in lemma 2.3 constitute?
  - figure 2 - use the same scale for both plots
  - plot together all lines from Fig. 3 and 4 for comparison. Maybe plot with different slices as edit / non-edit, as you want to show the linear / exp dependence with respect of some param growth.
  - Fig. 9 -- add explanation / footnote why the pattern of curve now is different with respect to length (it is decreasing p-value, not growing).
  - Fig. 10 -- why do we see U-shape curves now?
  - Fig. 12 (b) use the same bins ranges so that histograms are exactly comparable.
  - Appendices proofs - first state all lemmas/theorems for every proof, otherwise hard to come back to the main text every time.
  - Algorithm 6 line 6: what is $P_{xi}$? do we sample one token of $xi$ or sequence of them? Do we sample different permutations of $\xi$ sequence or not?
- "in which lie the elements of the watermark" -- I guess from English language the sentence is built incorrectly. Maybe "in which the elements of watermark lie"?
- page 5 "independent copy" -- what do you mean? "to break ties" in the footnote -- what does it mean exactly?
- What happens with watermark detection if we swap paragraphs in the generated text as the method of corruption? or swap sentences?
- Typo page 7: "Observe the watermark" -> "Observed the watermark"
- I got that theory is simpler to improve in one form, while empirically to be practical statistics form is different. But could you have any empirical result with original statistics and algorithms (for which the strong proof is given) to show that it is consistent with the theory?
- page 8 ITS is used before notation appeared.
- what are the probability of type 1/2 errors?
- what happens if block size $k$ is less than $m$?
- how do you propose to resolve the issue that different methods are robust with respect to different attacks? e.g. exp-edit is worse than others except attack with insertions/deletions.
- proofs questions:
  - lemma 2.2
    - (*) transition: what happens with the part when $y$ is not in $\nu_c$?
    - what is the notation $D_{TC}$? why do we have inequality for the every $A$ for $D_{TC}$ in this form?
    - the final inequality I don't get, it seems incorrect to me. Could you give more detailed transitions and explanations how you ended up with it?
  - lemma B.1
    - add $eta$ notation as I forgot it by that time of reading the paper :)
    - equation between (7) and (8): what it is $P$?
    - why width of $I(Y, \pi)$ is $\mu(Y)$? I didn't get entirely transitions in (8)
  - lemma 2.5
   - what is notation for $\tilde{\alpha}$?
  - the rest of proofs looks reasonable for me, though it is hard to read them as too many notations (probably unavoidable). Any simplifications (maybe put more details on every transition) and putting definitions of all vars at the beginning of each proof make it simpler to read and follow.
- "The retokenization is not always equal to the original" - why is this? Can you explain in more details?

**Strengths And Weaknesses:**

**Strengths**
- very well written paper
- new watermarks algorithms which preserve the original LM conditional distribution over next tokens
- independence on the model and prompt - thus more general watermarking algorithm
- improved robustness with respect to insertions / deletions / substitutions and paraphrasing compared to prior works
- more sensitive statistics compared to prior works
- deep theoretical analysis
- practical simplifications of the theoretical algorithms and demonstration of their work on real case models

**Weaknesses**
- restriction of the key: hack on permuting the key for generation is solving the issue, but still restrict the number of possible generations - deeper discussion of this would be nice to have in the paper
- some proofs in the paper are not entirely clear; notation can be improved; a bit heavy with the proofs without detailed derivation of the equations.
- mismatch between theory and practice for the algorithms - some fair check or detailed comparison between them is needed to confirm why it is still eligible to use modified version in practice (though it is cheaper to compute)
- sharable key between LM owner and user (general limitation which is fine to have for the paper)
- paper proposes 4 variants for the algorithm and it is not clear which we should use in practice as depending on the attack or case one is more preferable than others but not always the same one.

---

> ### Author Response · Authors · 2024-02-01
> **Response to Reviewer 3pbi**
>
> Thank you for all the suggested changes! We are currently updating the paper and will incorporate them in the revision we upload.
>
> Regarding discrepancies between our theory and experiments (e.g., the choice of test statistics), in our view Theorems 1 and 2 are the two critically important theoretical results since they are necessary to guarantee that our watermarks are in fact distortion-free (the distortion-free property must hold by definition across all prompts/language models, making it intractable to validate via experiments); thus, we ensure both theorems apply to the exact watermarks we employ in practice in our experiments. The principal utility of the remaining lemmas (e.g., the p-value upper bounds)—which apply to slight variations of the watermarks we employ in practice—lies in how they give us a theoretical framework to argue about trade-offs between various hyperparameter configurations of our watermarks (e.g., the block size k and watermark key length n); within reason, the choice of test statistic does not change the main takeaways from these lemmas (e.g., that p-values decay exponentially in the length of watermarked text and grow linearly in the length of the key sequence). Unlike distortion-freeness, the typical p-values of watermarked text are something that is easy to empirically measure (as we do in our experiments), so in our view it is less important that these lemmas apply to the exact watermarks we deploy in practice. In fact, the test statistics we employ in our experiments perform *even better* in practice than what we use to state the lemmas; the reason for using different test statistics in the lemmas is simply to make the statement of the results cleaner.
>
> > “restriction of the key: hack on permuting the key for generation is solving the issue, but still restrict the number of possible generations - deeper discussion of this would be nice to have in the paper”
>
> As you point out, the length of the watermark key sequence n is an upper bound on the number of unique generations per prompt. Fortunately, our theory and experiments both imply asymptotically that we can reliably detect watermarked text consisting of m tokens so long as $n = O(m^c)$ for any constant $c$. Though the runtime of our detection algorithm scales linearly in n (in our view, this computational overhead is the main limitation of our watermark in comparison to the hashing-based watermarks we cite in the paper), detection is easily parallelizable (in particular, we can align different parts of the watermark key sequence to the putative text in parallel). We discuss these trade-offs in Section 4, as well as guidance for setting n in practice.
>
> > "​​paper proposes 4 variants for the algorithm and it is not clear which we should use in practice as depending on the attack or case one is more preferable than others but not always the same one."
>
> We recommend using the EXP-edit watermark in Section 4 since this watermark was generally the most robust across all attacks we tried implementing.
>
> > “"The retokenization is not always equal to the original" - why is this? Can you explain in more details?”
>
> The reason that the retokenized text in our watermark detection experiments is not always equal to the original sequence of tokens generated by the language model is because byte-pair tokenizations of text (used by both the OPT and LLaMA) are not unique. Byte-pair tokenization augments a base vocabulary (e.g., characters) with extra tokens to represent common substrings. This allows the tokenizer to encode text using shorter sequences of tokens, at the cost of uniqueness: you can always tokenize text using the base vocabulary, and depending how you parse an input string, there may be many other shorter tokenizations corresponding to the same input.

---

> > ### Comment · Reviewer_3pbi · 2024-02-08
> > **Reply to authors**
> >
> > Dear authors,
> >
> > Thanks for all clarifications and details you gave to my and  other reviewers questions. I don't have any other questions or concerns, but I am waiting for the updated and final manuscript draft for the final decision.

---

> > > ### Author Response · Authors · 2024-02-09
> > > **ETA on updated manuscript**
> > >
> > > We will post the updated manuscript in a few days. Thanks!

---

> ### Author Response · Authors · 2024-02-14
> **Updated manuscript**
>
> We have updated the manuscript to incorporate the suggestions. See below for a description of the changes (organized by suggestion). Thanks again for the helpful feedback!
>
> > "preserve original text distribution" -- I would be more precise here even in the intro part of the paper, as in the end we preserve only conditional distribution of language model.
>
> changed to "original language model's sampling distribution"
>
> > $\Nu^{}$ and in general notation of "" is not introduced.
>
> We use $\nu$ to refer to the watermark key distribution. We have updated the algorithm descriptions to clarify the definition of $\nu$
>
> > definition 1 - maybe include here conditional pdf of LM as an example? otherwise seems the definition is about general pdf of text
>
> Definition 1 is valid for general text distributions (regardless of whether or not they are modeled by an LM), so we left it as is
>
> > lemma 2.1 why $n$ should be bigger than $m$? maybe introduce earlier how you generate the text given watermark sequence? that every token in watermark is used to generate next token of text? What happens if $n < m$? (I got that empirically it seems still work to detect if text is watermarked or not)?
>
> We updated the prose immediately preceding Lemma 2.1 to explain why it is important that $n$ is bigger than $m$ (basically, we cannot reuse an element of the watermark key sequence)
>
> > algorithm 2 line 3: does $\xi^{(t)} \in \Xi^*$?
>
> Yes, it does. The changes we in response to the earlier comment re: $\nu$ should also clarify this point.
>
> > page 9 what does notation $\xi_i$ and $\xi_j$ and in lemma 2.3 constitute?
>
> We are indexing $\xi$ so that $\xi_i$ and $\xi_j$ respectively denote the i^th and j^th elements of the sequence.
>
> > figure 2 - use the same scale for both plots
>
> The y-axis of both plots is now set to [0.0,1.0]
>
> > plot together all lines from Fig. 3 and 4 for comparison. Maybe plot with different slices as edit / non-edit, as you want to show the linear / exp dependence with respect of some param growth.
>
> We tried plotting all the lines together, but it was a bit too cluttered; instead, we combined the two figures into a single 2x2 figure block with one caption.
>
> > Fig. 9 -- add explanation / footnote why the pattern of curve now is different with respect to length (it is decreasing p-value, not growing).
> > Fig. 10 -- why do we see U-shape curves now?
>
> In Fig. 9 (now Fig. 8), the p-values decrease with text length, which is what we generally expect to happen. As for Fig. 10 (now Fig. 9), the U-shaped curve is likely due to the fact that there is more flexibility in translating longer texts, making the roundtrip translation attack is more effective. We've added this discussion to the paper.
>
> > Fig. 12 (b) use the same bins ranges so that histograms are exactly comparable.
>
> Done.
>
> > Appendices proofs - first state all lemmas/theorems for every proof, otherwise hard to come back to the main text every time.
>
> We make sure to state and prove all supporting lemmas before using them. As for the main lemmas/theorems that appear in the main text, we double checked that the corresponding sections of the appendix link back to the main text. Thus, we decided not to restate any theorems to reduce clutter.
>
> > Algorithm 6 line 6: what is $P_\xi$? do we sample one token of $\xi$ or sequence of them? Do we sample different permutations of sequence or not?
>
> This was a typo. It should be $\nu$ not $P_\xi$. Hopefully this clarifies the issue (we also made sure to explicitly define $\nu$ in the algorithm, as per our response to a previous comment).
>
> > "in which lie the elements of the watermark" -- I guess from English language the sentence is built incorrectly. Maybe "in which the elements of watermark lie"?
>
> Both sentences are correct, but per the suggestion we changed to "in which the elements of the watermark lie" to improve readability.
>
> > page 5 "independent copy" -- what do you mean? "to break ties" in the footnote -- what does it mean exactly?
>
> We changed "independent copy" to "independent, identically distributed copy" to be more explicit, and added clarification in the footnote over what we mean by a "tie" (we mean that the original test statistic is equal to one of the resampled statistics)
>
> > What happens with watermark detection if we swap paragraphs in the generated text as the method of corruption? or swap sentences?
>
> The watermark will remain detectable (this is essentially a version of a cropping attack, which we discuss in the paper).
>
> > Typo page 7: "Observe the watermark" -> "Observed the watermark"
>
> Changed to just "The watermark..."
>
> > page 8 ITS is used before notation appeared.
>
> Changed to "inverse transform sampling".
>
> > what are the probability of type 1/2 errors?
>
> False positive/negatives (we added this clarification to the paper as a parenthetical).
>
> > what happens if block size $k$ is less than $m$?
>
> The p-values will scale as \exp(-k). We recommend setting $k < m$ if one expects the text to be cropped.

---

> > ### Author Response · Authors · 2024-02-14
> > **Updated manuscript, continued**
> >
> > > how do you propose to resolve the issue that different methods are robust with respect to different attacks? e.g. exp-edit is worse than others except attack with insertions/deletions.
> >
> > EXP-edit is only marginally less robust to substitutions than EXP and considerably more robust to insertions/deletions. Thus, we recommend deploying EXP-edit as is.
> >
> > > lemma 2.2
> >
> > $D_{TV}$ is an abbreviation of total variation distance. We made sure to clarify this explicitly in the proof of Lemma 2.2, which hopefully resolves the other comments as well (e.g., the inequality for every $A$ is a well-known fact about total variation distance). As for the case when $y$ is not in $\nu_c$, we can lower-bound the contribution of this case to the expectation by zero, leading to the final result.
> >
> > > lemma B.1
> >
> > We restated the definition of $\eta$. As for $P$, that was a typo: it should be $\mu$. After fixing this typo, it should now hopefully be clear why the width of the interval is $\mu(Y)$.
> >
> > > lemma 2.5
> >
> > $\tilde{\alpha}$ is defined as part of the lemma statement. Let us know if there's any more clarification we can add that would be helpful.
> >
> > > "The retokenization is not always equal to the original" - why is this? Can you explain in more details?
> >
> > We added the explanation in our original response to the paper.

---

### Review · Reviewer_XRf9 · 2024-01-24

**Summary Of Contributions:**

*** Requested short comparison to the work of Christ et al. (2023) ***

A previous work of Christ et al. (2023) presents "undetectable" watermarks for language models. Their work is cited and briefly discussed in this submission. I will go into a few more details on the differences and similarities between the two.

The main strength of Christ et al. over the current submission is that their definition of "undetectability" is much stronger than the current submission's definition of "distortion-free".
In this submission, "distortion-free" means that for a single query (i.e., a single prompt given to the model), the distribution of responses will be the same as if the original model was given the same query.
This definition is natural, but doesn't take into account multiple queries: it could be that a set of several responses to several queries is now correlated to each other in some manner. (For example, the model could become fully deterministic and still be distortion-free over the choice of random key).
In Christ et al., the definition of "undetectable" is stronger: they require that even if polynomially many, possibly adaptive, queries are made to the model, their responses would still be indistinguishable from those that would have been given by the original model.

The authors of this work do notice the issue of correlated responses and address it with a heuristic: they essentially have several different "keys" and run the scheme with a random one each query to "obfuscate" the correlations. Of course, the number of those "keys" is small as this requires enumerating over all of those "keys" in the detection algorithm. Thus this isn't really preventing correlations but just makes them slightly more difficult to notice (and indeed there is no formal guarantee about correlations between different queries in this submission, unlike in the other work).
Besides this new heuristic, the rest of the protocol is very similar to both the work of Christ et al. and another previous work of Aaronson (also cited in this paper).
Section 2.1 in this submission, for example, exactly describes an algorithm presented in Christ et al., without stating anywhere it is not a new contribution.
The entire "generate" method in fact is not new and is a variation of Christ et al. and/or Aaronson.

On the other hand, the strengths of the current paper over the aforementioned previous works are twofold:
First, this submission provides empirical implementation and evaluations, which both Christ et al. and Aaronson failed to do.
Second, this submission focuses more on robustness of the watermarks to edits in comparison to the previous works, which is an important property for watermarks that is "orthogonal" to undetectability/distortion-freeness. The authors study the "robustness" of their watermark both theoretically and empirically.

**Audience:**

Yes

**Broader Impact Concerns:**

-

**Claims And Evidence:**

No

**Requested Changes:**

A better comparison to the papers mentioned above is needed. In particular, comparing the definitions of robustness-free/undetectability, comparing the guarantees of robustness, and comparing the methodology to both Aaronson and Christ et al.
Techniques and algorithms are not appropriately attributed and cited in several places.

**Strengths And Weaknesses:**

^

---

> ### Author Response · Authors · 2024-02-01
> **Response to Reviewer XRf9**
>
> Regarding comparison to Christ et al., the definition of “undetectability” is not stronger than our notion of “distortion-free”: our definition implies exact equality in distribution to the original unwatermarked language model whereas undetectability implies approximate equality. It is worth emphasizing that neither our watermarks nor the watermark of Christ et al. satisfies exact equality in distribution for multiple (i.e., $> 2$) queries: just as the results of multiple calls to shift-generate will be correlated due to some probability of reusing a part of the watermark key sequence, multiple calls to Christ et al.’s watermark generation method will also be correlated due to some probability of a hash collision. In fact, Christ et al.’s watermark is not distortion-free for even a single query, again due to possible hash collisions (recall our watermarks are distortion-free up to a certain maximum generation budget corresponding to the length of the watermark key sequence, which is a hyperparameter we control). We will clarify how the notion of distortion-free in our paper relates to undetectability.
>
> Meanwhile, both ours and Christ et al.’s watermarks satisfy Christ et al.’s definition of undetectability. In particular, supposing we wish to detect the watermark from m tokens, there exist versions of our watermarks such that the following all hold asymptotically: 1) given m watermarked tokens, our detection method will produce p-values near zero; 2) the outputs of poly(m) adaptive queries to our generation method will be approximately indistinguishable in distribution from unwatermarked outputs; 3) the detection algorithm runs in poly(m) time. Concretely, we can set the length of the watermark key sequence $n = m^{c_0}$ for any constant $c_0$ while asymptotically ensuring negligible, i.e., o(1), p-values (since our p-values decay exponentially with m and increase only linearly with n). Moreover, the probability of reusing an element of the watermark key sequence will be negligible so long as the total number of (adaptively chosen) queries is $O(m^{c_1}$) for some constant $c_1 \approx c_0/2$. Finally, the runtime of our detection algorithm scales linearly in n, where recall we have chosen n = poly(m). If it would be valuable, we are happy to formalize this argument as a theorem in our paper.
>
> > “Section 2.1 in this submission, for example, exactly describes an algorithm presented in Christ et al., without stating anywhere it is not a new contribution. The entire "generate" method in fact is not new and is a variation of Christ et al. and/or Aaronson.”
>
> The main technical novelty of our watermarking protocols lies in how we avoid hashing previously generated tokens to watermark the next token, which distinguishes our generate method from those in prior work (Kirchenbauer et al., Aaronson, and Christ et al.) and crucially enables our watermarks to be both distortion-free *and* robust (unlike prior work). Also, the inverse transform sampling procedure we develop in Section 2.1 is different from the sampling procedure in Christ et al. In particular, Christ et al. construct a reduction from text generation over general vocabularies to text generation over a binary vocabulary and then sample from the binary vocabulary. In contrast, we directly use inverse transform sampling to sample from the original vocabulary. Inverse transform sampling itself has long been a well-known technique; the main technical innovation of our ITS watermark lies not in the development of inverse transform sampling but rather in how we randomize the ordering of the vocabulary in order to correlate the permuted token indices with the watermark key sequence during detection, without knowing the original token probabilities.

---

> > ### Comment · Reviewer_XRf9 · 2024-02-01
> >
> > This comparison simply isn't correct... I'll go into more technical details this time to be clearer, hopefully this will clear up the misunderstanding.
> >
> > 1. Let's denote the size of the key by lambda.
> > In your scheme, as you repeated in your response here, if you generate more than ~sqrt(lambda) tokens you'll start seeing correlations.
> > That means, that even a single query with a long enough response will not be "distortion-free", and most definitely ~sqrt(lambda) responses, even if the responses are short, will be correlated.
> > Your detection algorithm runs in >>lambda time, and hence sqrt(lambda) must be small: it is much smaller than the detection time (of a single response) which should be feasible. Hence, your watermark changes the distribution even when only a relatively small number of generated tokens are seen.
> > In Christ et al., on the other hand, it is proven that if you see less than exponential(lambda) responses then the probability of any algorithm to distinguish the original from their modified distribution is at most exp(-lambda).
> > Theirs is the standard definition of "indistinguishable" in cryptography research and the best possible dependence on the key length, as it basically means that you need to enumerate over all possible keys of the same length to detect the difference (which you can indeed do: if you'll try all keys and find the correct one you can always verify there is a watermark).
> > Crucially, exp(lambda) (i.e., time to enumerate over all keys of the same length) is usually assumed to be infeasible, while poly(lambda) (i.e., the time you spend to store the key or generate/detect a watermark in your scheme) is most definitely feasible. Hence, in their scheme the watermark is undetectable in any "feasible" running time, and yours is detectable even in the time it takes to detect the watermark in a single response.
> >
> > 2. In Example 1 in your Section 2.1 (bottom of page 6) you also consider only a binary alphabet and give an algorithm that is identical to the one in Section 4.2 of Christ et al., you do it without any citation. Please read the two sections side-by-side...

---

> ### Author Response · Authors · 2024-02-03
> **Follow up comment**
>
> 1. We define the watermark key as a sequence of n random variables since our focus is not on the distinction between randomness and pseudorandomness; however, we could alternatively generate a pseudorandom sequence of length n = poly($\lambda$) from an initial “key” of length $\lambda$, in which case all of the claims in our original response should essentially hold by replacing m with $\lambda$: i.e., neg($\lambda$) error (assuming sufficient entropy), poly($\lambda$) adaptive queries, and poly($\lambda$) detection time. In any case, we acknowledge in Section 4 that the main limitation of our watermarks in comparison to hashing-based watermarks such as Kirchenbauer et al., Aaronson, and Christ et al. is that the computational overhead of detection scales with n (i.e., the total number of distortion-free watermarked tokens we can generate). We posit that robustness is worth the additional overhead, especially seeing as detection easily parallelizes (in the same sense as a database lookup).
> 2. Apologies, we thought the original comment was referring to the ITS watermark in Section 2.3 (we conflated Section 2.3 with Section 2.1 in our original response). Example 1 in Section 2.1 is intended to serve as a warm-up toy example to build intuition for the watermarks to come in Sections 2.3 and 2.4. Note that Example 1 is different from the watermark of Christ et al. due to the fact that we are explicitly watermarking strings with binary elements (as a toy example), whereas Christ et al. first reduce from general vocabularies to the binary case as part of their actual (non-toy) watermark. For this reason, it seems unfair to Christ et al. to equate our toy example with their watermark, but we are happy to add a note explaining that the intuition of the token sampling procedure in our toy example is the same as in Christ et al. after their reduction from general to binary vocabularies. Note also that unlike like Christ et al. we are not hashing previous tokens.

---

### Author Response · Authors · 2024-02-01
**Response to reviews**

We thank all the reviewers for their thoughtful and detailed feedback. We will incorporate these suggestions into the next version of the paper. We have responded to each reviewer individually in the comments.

---

### Decision · Action_Editor_TwPs · 2024-04-24

**Recommendation:** Accept with minor revision

**Comment:**

We apologize to the authors for the extended delay.  Please see the comments above about claims and evidence for the justification and for the expectations for the revision.

**Audience:**

Watermarking is an important and timely topic for large language models so the topic would absolutely be interesting to a significant fraction of the TMLR audience.  The specific results about robustness of watermarks would also be interesting to nearly anyone interested in the topic.

**Claims And Evidence:**

The formal claims in the paper are supported by proofs.  However, there is an informal claim in the paper that the watermark "does not change the model's distribution" that we feel requires some clarification.  In particular, the watermark in this paper only preserves the distribution of a single prompt, while prior work preserves the joint distribution of several prompts.  The explanation in the submission for how to handle multiple prompts is unconvincing because it relies on the attacker having less computational resources than the watermarker.  Please see the comments by the reviewers about the difference between the notion of "undetectability" from Christ et al. and the notion of "distortion-free" from the submission.

In the spirit of TMLR's requirement of "claims supported by evidence," we feel it's important that the introduction clairfies precisely what the claims are and how they differ from the most directly comparable work.  However, since this issue concerns the informal introduction, we feel it can be corrected with a suitable minor revision.

---

> ### Author Response · Authors · 2024-05-24
>
> Thank you for your feedback. We have uploaded our camera ready submission.
>
> Summary of changes:
> We have added more detail to the comparison of our definition of distortion-freeness to Christ et al.’s definition of undetectability in the related work section. We have also included a more detailed discussion of the distortion of our watermarks in the multi-query setting at the end of Section 2 as well as the trade-offs between our watermarks vs. hashing-based watermarks in Section 4.
>
> Responses to action editor recommendation:
> > The explanation in the submission for how to handle multiple prompts is unconvincing because it relies on the attacker having less computational resources than the watermarker.
>
> We previously acknowledge that a limitation of our watermarks versus hashing-based watermarks is that the runtime of detection depends on the length of the watermark key sequence. We now discuss this trade-off in more concrete detail in Sections 2.2 and 4.
>
> > We feel it's important that the introduction clairfies precisely what the claims are and how they differ from the most directly comparable work
>
> In our initial draft, we precisely define what we mean by preserving the text distribution (i.e., single query distortion-freeness) in the introduction (see the third to last paragraph on page 2). We have since added more detail to the related work section. We have also added a discussion of the degree of distortion of our watermark in the multi-query setting, both in Sections 2.2 and 4.